# Towards Reliable Link Prediction with Robust Graph Information Bottleneck

## Abstract

Link prediction on graphs has achieved great success with the rise of deep graph learning. However, the potential robustness under the edge noise is less investigated. We reveal that the inherent edge noise that naturally perturbs both input *topology* and target *label* leads to severe performance degradation and representation collapse. In this work, we propose an information-theory-guided principle, Robust Graph Information Bottleneck (RGIB), to extract reliable supervision signals and avoid representation collapse. Different from the general information bottleneck, RGIB decouples and balances the mutual dependence among graph topology, target labels, and representation, building new learning objectives toward robust representation. We also provide two instantiations, RGIB-SSL and RGIB-REP, which benefit from different methodologies, i.e., self-supervised learning and data reparametrization, for implicit and explicit data denoising, respectively. Extensive experiments on 6 benchmarks of various scenarios verify the effectiveness of the proposed RGIB.

## 1 Introduction

As a fundamental problem in graph learning, link prediction (Liben-Nowell & Kleinberg, 2007) has attracted growing interest in real-world applications like drug discovery (Ioannidis et al., 2020), knowledge graph completion (Bordes et al., 2013), and question answering (Huang et al., 2019). Recent advances from heuristic designs (Katz, 1953; Page et al., 1999) to graph neural networks (GNNs) (Kipf & Welling, 2016a; Gilmer et al., 2017; Kipf & Welling, 2016b; Zhang & Chen, 2018; Zhu et al., 2021) have achieved superior performances. Nevertheless, the poor robustness in imperfect scenarios with the inherent edge noise is still a practical bottleneck to the current deep graph models (Gallagher et al., 2008; Ferrara et al., 2016; Wu et al., 2022a; Dai et al., 2022).

Early explorations improve the robustness of GNNs for node classification under label noise (Dai et al., 2021; Li et al., 2021) through the smoothing effect of neighboring nodes. Other methods achieve a similar goal via randomly removing edges (Rong et al., 2020) or actively selecting the informative nodes or edges and pruning the task-irrelevant ones (Zheng et al., 2020; Luo et al., 2021). However, when applying these noise-robust methods to the link prediction with noise, only marginal improvements are achieved (see Section 5). The attribution is that the edge noise can naturally deteriorate both the input topology and the target labels (Figure 1(a)). Previous works that consider the noise either in input space or label space cannot effectively deal with such a coupled scenario. Therefore, it raises a new challenge to understand and tackle the edge noise for robust link prediction.

In this paper, we dive into the inherent edge noise and empirically show the significantly degraded performances it leads to (Section 3.1). Then, we reveal the negative effect of the edge noise through carefully inspecting the distribution of learned representations, and discover that graph representation is severely collapsed, which is reflected by much lower alignment and poorer uniformity (Section 3.2). To solve this challenging problem, we propose the Robust Graph Information Bottleneck (RGIB) principle based on the basic GIB for adversarial robustness (Wu et al., 2020) (Section 4.1). Conceptually, the RGIB principle is with new learning objectives that decouple the mutual information (MI) among noisy inputs $\tilde{A}$, noisy labels $\tilde{Y}$, and the representation $\boldsymbol{H}$. As illustrated in Figure 1(b), RGIB generalizes the basic GIB to learn a robust representation that is resistant to the edge noise.

Technically, we provide two instantiations of RGIB based on different methodologies, i.e., RGIB-SSL and RGIB-REP: (1) the former utilizes contrastive pairs with automatically augmented views

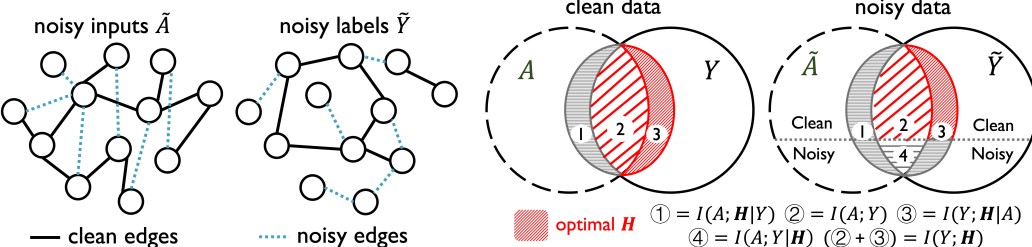

(a) Illustration of inherent edge noise. The GNN takes the graph topology $A$ as inputs, and predicts the logits of unseen edges with labels $Y$. The noisy $\tilde{A}$ and $\tilde{Y}$ are added with random edges for simulating the inherent edge noise as in Def. 3.1.

(b) The basic GIB (left) and the proposed RGIB (right). $I(\cdot;\cdot)$ here indicates the mutual information. To solve the intrinsic deficiency of basic GIB in tackling the edge noise, the RGIB learns the graph represenation $H$ via a further balance of informative signals ①, ③, ④ regarding the $H$.

Figure 1: Link prediction with inherent edge noise (a) and the proposed RGIB principle (b).

to form the informative regularization in a self-supervised learning manner (Section 4.2); and (2) the latter explicitly purifies the graph topology and supervision targets with the reparameterization mechanism (Section 4.3). Both instantiations are equipped with adaptive designs, aiming to effectively estimate and balance the corresponding informative terms in a tractable manner. For example, the hybrid augmentation algorithm and self-adversarial alignment loss for RGIB-SSL, and the relaxed information constraints on topology space as well as label space for RGIB-REP. Empirically, we show that these two instantiations work effectively under extensive noisy scenarios and can be seamlessly integrated with various existing GNNs (Section 5). Our main contributions are summarized as follows.

- To our best knowledge, we are the *first* to study the robustness problem of link prediction under the inherent edge noise. We reveal that the inherent noise can bring a severe representation collapse and performance degradation, and such negative impacts are general to common datasets and GNNs.

- We propose a general learning framework, RGIB, with refined representation learning objectives to promote the robustness of GNNs. Two instantiations, RGIB-SSL and RGIB-REP, are proposed upon different methodologies that are equipped with adaptive designs and theoretical guarantees.

- Without modifying the GNN architectures, the RGIB achieves state-of-the-art results on 3 GNNs and 6 datasets under various noisy scenarios, obtaining up to $12.9\%$ AUC promotion. The distribution of learned representations is notably recovered and more robust to the inherent noise.

## 2 PRELIMINARIES

**Notation.** We denote $\mathcal{V} = \{v_i\}_{i=1}^N$ as the set of nodes and $\mathcal{E} = \{e_{ij}\}_{ij=1}^M$ as the set of edges. With adjacent matrix $A$ and node features $X$, an undirected graph is denoted as $\mathcal{G} = (A, X)$, where $A_{ij} = 1$ means there is an edge $e_{ij}$ between $v_i$ and $v_j$. $X_{[i,:]} \in \mathbb{R}^D$ is the $D$-dimension node feature of $v_i$. Link prediction is to indicate the existence of query edges with labels $Y$ that are not observed in $A$.

**GNNs for Link Prediction.** We follow the common link prediction framework, i.e., graph auto-encoders (Kipf & Welling, 2016b), where the GNN architecture can be GCN (Kipf & Welling, 2016a), GAT (Veličković et al., 2018), or SAGE (Hamilton et al., 2017). Given a $L$-layer GNN, the graph representations $H \in \mathbb{R}^{|\mathcal{V}| \times D'}$ for each node $v_i \in \mathcal{V}$ are obtained by a $L$-layer message propagation as the encoding process. For decoding, logits $\phi_{e_{ij}}$ of each query edge $e_{ij}$ are computed with a readout function, e.g., the dot product $\phi_{e_{ij}} = h_i^\top h_j$. Finally, the optimization objective is to minimize the binary classification loss, i.e., $\min \mathcal{L}_{cls} = \sum_{e_{ij} \in \mathcal{E}^{train}} -y_{ij}\log\big(\sigma(\phi_{e_{ij}})\big) - (1-y_{ij})\log\big(1-\sigma(\phi_{e_{ij}})\big)$, where $\sigma(\cdot)$ is the sigmoid function, and $y_{ij} = 1$ for positive edges while $y_{ij} = 0$ for negative ones.

**Topological denoising approaches.** A natural way to tackle the input edge noise is to directly clean the noisy graph. Sampling-based methods, such as DropEdge (Rong et al., 2020), NeuralSparse (Zheng et al., 2020), and PTDNet (Luo et al., 2021), are proposed to remove the task-irrelevant edges. Besides, as GNNs can be easily fooled by adversarial network with only a few perturbed edges (Chen et al., 2018; Zhu et al., 2019; Entezari et al., 2020), defending methods like GCN-jaccard (Wu et al., 2019) and GIB (Wu et al., 2020) are designed for pruning adversarial edges.

**Label-noise-resistant techniques.** For tackling the general problem of noisy label, Co-teaching (Han et al., 2018) lets two neural networks teach each other with small-loss samples based on the memorization effect (Arpit et al., 2017). Besides, peer loss function (Liu & Guo, 2020) pairs independent peer examples for supervision and works within the standard empirical risk minimization framework. As for the graph domain, label propagation techniques proposed by pioneer works (Dai et al., 2021; Li et al., 2021) propagate the reliable signals from clean nodes to noisy ones, which are nonetheless entangled with the node annotations and node classification task that cannot be directly applied here.

## 3 AN EMPIRICAL STUDY OF THE INHERENT EDGE NOISE

In this section, we attempt to figure out how GNNs behave when learning with the edge noise and what are the latent mechanisms behind it. We first present an empirical study in Section 3.1 and then investigate the negative impact of noise through the lens of representation distribution in Section 3.2.

### 3.1 HOW DO GNNS PERFORM UNDER THE INHERENT EDGE NOISE?

Since existing benchmarks are usually well-annotated and clean, there is a need to simulate the inherent edge noise properly to investigate the impact of noise. Note the data split manner adopt by most relevant works (Kipf & Welling, 2016b; Zhang & Chen, 2018; Zhu et al., 2021) randomly divides partial edges as observations and the others as prediction targets. The inherent edge noise, if exists, should be *false positive* samples and uniformly distributed to both input observations and output labels. Thus, the training data can be with noisy adjacence $\tilde{A}$ and noisy labels $\tilde{Y}$, i.e., the input noise and label noise. We elaborate the formal simulation of such an *additive* edge noise as follows.

**Definition 3.1** (Additive edge noise). *Given a clean training data, i.e., observed graph $\mathcal{G} = (A, X)$ and labels $Y \in \{0, 1\}$ of query edges, the noisy adjacence $\tilde{A}$ is generated by only adding edges to the original adjacent matrix $A$ while keeping the node features $X$ unchanged. The noisy labels $\tilde{Y}$ are generated by only adding false-positive edges to the labels $Y$. Specifically, given a noise ratio $\varepsilon_a$, the added noisy edges $A'$ ($\tilde{A} = A + A'$) are randomly generated with the zero elements in $A$ as candidates. It satisfies that $A' \odot A = O$ and $\varepsilon_a = |nonzero(\tilde{A})| - |nonzero(A)| / |nonzero(A)|$. Similarly, noisy labels are generated and added to original labels, where $\varepsilon_y = |nonzero(\tilde{Y})| - |nonzero(Y)| / |nonzero(Y)|$.*

With the simulated noise, an empirical study is then performed with various GNNs and datasets. As shown in Figure 2, *the edge noise causes a significant drop in performance, and a larger noise ratio generally leads to greater degradation.* It means that these common GNNs normally trained with the stochastic gradient descent are vulnerable to the inherent edge noise, yielding a severe robustness problem. [1] However, none of the existing defending methods are designed for such a coupled noise, which is practical for real-world graph data that can be extremely noisy (Gallagher et al., 2008; Wu et al., 2022a). Thus, it is urgent to understand the noise effect and devise the corresponding robust method.

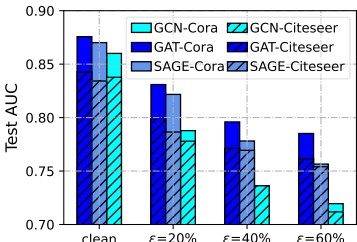

Figure 2: Performances (test AUC) with simulated edge noise. For fair evaluations, results are obtained by repeating 10 times with 4-layer GNNs on each dataset.

### 3.2 UNDERSTANDING THE IMPACT OF NOISE VIA REPRESENTATION DISTRIBUTION

Denote a GNN as $f_{\boldsymbol{w}}(\cdot)$ with learnable weights $\boldsymbol{w}$. Node representations are extracted by a forward pass as $\boldsymbol{H} = f_{\boldsymbol{w}}(A, X)$ while the backward propagation with stochastic gradient descent optimizes the GNN as by contrast When encountering noise within $\tilde{A}$ and $\tilde{Y}$, the representation $\boldsymbol{H}$ can be directly influenced, since the training neglects the adverse effect of data corruption. Besides, the GNN readouts the edge logit $\phi_{e_{ij}}$ based on top-layer node representations $\boldsymbol{h}_i$ and $\boldsymbol{h}_j$, which are possibly degraded or even collapsed under the edge noise (Graf et al., 2021; Nguyen et al., 2022).

For quantitive and qualitative analysis of $\boldsymbol{H}$ under edge noise, we introduce two concepts (Wang & Isola, 2020) here, i.e., *alignment* and *uniformity*. Specifically, alignment is computed as the

---

[1]For generality, we provide the full empirical study in Appendix D.

distance of representations between two randomly augmented graphs. It quantifies the stability of GNN when encountering edge noise in the testing phase. A higher alignment means the GNN is more resistant and invariant to input perturbations. Uniformity, on the other hand, qualitatively measures the denoising effects of GNN when learning with edge noise in the training phase. A greater uniformity implies that the learned representations of various samples are more uniformly distributed on the unit hypersphere, preserving as much information about the original data as possible.

As shown in Table 1 and Figure 3, *a poorer alignment and a worse uniformity are brought by a severer edge noise.* The learned GNN $f_{\boldsymbol{w}}(\cdot)$ is more sensitive to input perturbations as the alignment values are sharply increased, and the learned edge representations tend to be less uniformly distributed and gradually collapse to individual points as the noise ratio increases. Thereby, it is discovered that the graph representation is severely collapsed under the inherent edge noise, resulting in the non-discriminative property. This is usually undesirable and potentially risky for downstream applications.

Table 1: Mean values of alignment on Cora dataset, that are calculated as the L2 distance of representations of two randomly perturbed graphs i.e., $\frac{1}{N}\sum_{i=1}^{N}||\boldsymbol{H}_1^i-\boldsymbol{H}_2^i||_2$. where $\boldsymbol{H}_1^i = f_{\boldsymbol{w}}(A_1^i, X)$ and $\boldsymbol{H}_2^i = f_{\boldsymbol{w}}(A_2^i, X)$.

| dataset | Cora | Citeseer |
|---------|------|----------|
| clean | .616 | .445 |
| $\varepsilon=20\%$ | .687 | .586 |
| $\varepsilon=40\%$ | .695 | .689 |
| $\varepsilon=60\%$ | .732 | .696 |

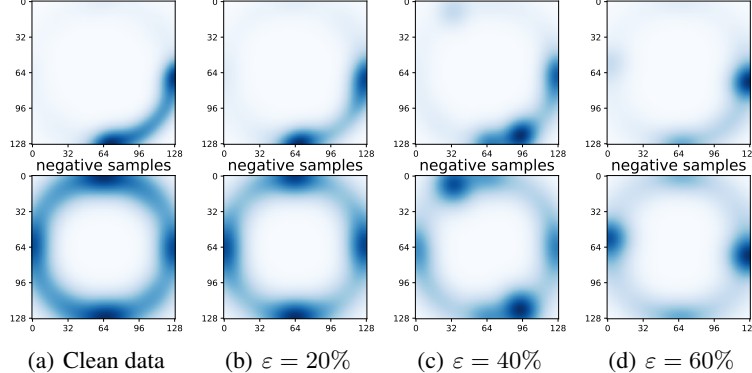

(a) Clean data    (b) $\varepsilon = 20\%$    (c) $\varepsilon = 40\%$    (d) $\varepsilon = 60\%$

Figure 3: Uniformity distribution on Cora dataset. Here, we map the representations of query edges in test set to unit circle of $\mathbb{R}^2$ followed by the Gaussian kernel density estimation, as in (Wang & Isola, 2020).

## 4  ON TACKLING THE REPRESENTATION COLLAPSE VIA ROBUST GIB

As aforementioned, the inherent edge noise brings unique challenges to link-level graph learning. Without prior knowledge like the noise ratio, without the assistance of auxiliary datasets, and even without modifying GNN architectures, how can learned representations be resistant to the inherent noise? Here, we formally build a method to address this problem from the perspective of graph information bottleneck (Section 4.1) and design its two practical instantiations (Section 4.2 and 4.3).

### 4.1  THE PRINCIPLE OF ROBUST GIB

Recall in Section 3.2, the graph representation $\boldsymbol{H}$ is severely degraded due to incorporating noisy signals in $\tilde{A}$ and $\tilde{Y}$. To robustify $\boldsymbol{H}$, one can naturally utilize the information constraint based on the basic graph information bottleneck (GIB) principle (Wu et al., 2020; Yu et al., 2020), i.e., solving

$$\min \text{GIB} \triangleq -I(\boldsymbol{H}; \tilde{Y}), \ s.t. \ I(\boldsymbol{H}; \tilde{A}) < \gamma. \tag{1}$$

where the hyper-parameter $\gamma$ constrains the MI $I(\boldsymbol{H}; \tilde{A})$ to avoid $\boldsymbol{H}$ from capturing excess task-irrelevant features from $\tilde{A}$. The basic GIB (Eqn. 1) can effectively withstand the input perturbation (Wu et al., 2020). However, it is intrinsically susceptive to label noise since it entirely preserves the label supervision with maximizing $I(\boldsymbol{H}; \tilde{Y})$. Empirical results in Section 5 show that it becomes ineffective when learning with inherent edge noise simulated as Definition 3.1.

The basic GIB decreases $I(\boldsymbol{H}; \tilde{A}|\tilde{Y})$ by directly constraining $I(\boldsymbol{H}; \tilde{A})$ to handle the input noise, as illustrated in Figure 1(b). Symmetrically, the label noise can be hidden in $I(\boldsymbol{H}; \tilde{Y}|\tilde{A})$, but trivially constraining $I(\boldsymbol{H}; \tilde{Y})$ to regularize $I(\boldsymbol{H}; \tilde{Y}|\tilde{A})$ is not ideal, since it will conflict with Eqn. 1 and also cannot tackle the noise within $I(\tilde{A}; \tilde{Y})$. Thus, it is crucial to further decouple the dependences among $\tilde{A}$, $\tilde{Y}$, and $\boldsymbol{H}$, while the noise can exist in areas of $I(\boldsymbol{H}; \tilde{Y}|\tilde{A})$, $I(\boldsymbol{H}; \tilde{A}|\tilde{Y})$, and $I(\tilde{A}; \tilde{Y}|\boldsymbol{H})$.

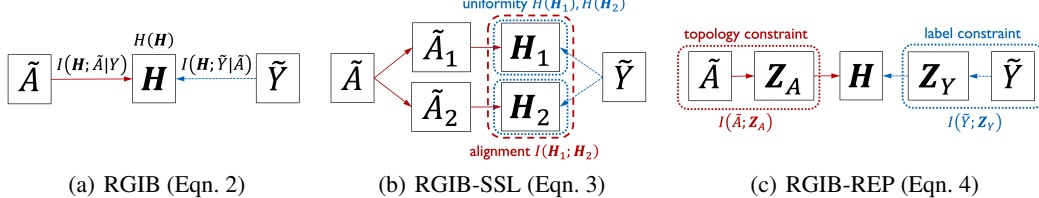

Figure 4: Digrams of the proposed RGIB and its two instantiations (best viewed in color).

Analytically, $I(\tilde{A}; \tilde{Y}|\boldsymbol{H}) = I(\tilde{A}; \tilde{Y}) + I(\boldsymbol{H}; \tilde{Y}|\tilde{A}) + I(\boldsymbol{H}; \tilde{A}|\tilde{Y}) - H(\boldsymbol{H}) + H(\boldsymbol{H}|\tilde{A}, \tilde{Y})$, where $I(\tilde{A}; \tilde{Y})$ is a constant and redundancy $H(\boldsymbol{H}|\tilde{A}, \tilde{Y})$ can be easily minimized. Thus, the $I(\tilde{A}; \tilde{Y}|\boldsymbol{H})$ can be approximated by the other three terms, i.e., $H(\boldsymbol{H})$, $I(\boldsymbol{H}; \tilde{Y}|\tilde{A})$ and $I(\boldsymbol{H}; \tilde{A}|\tilde{Y})$. Since the two later terms are also with noise, a balance of these three informative terms can be a solution to the problem.

Based on the above analysis, we propose the *Robust Graph Information Bottleneck (RGIB)*, a new learning objective to balance informative signals regarding $\boldsymbol{H}$, as illustrated in Figure 4(a), i.e.,

$$\min \text{RGIB} \triangleq -I(\boldsymbol{H}; \tilde{Y}), \ s.t. \ \gamma_H^- < H(\boldsymbol{H}) < \gamma_H^+, \ I(\boldsymbol{H}; \tilde{Y}|\tilde{A}) < \gamma_Y, \ I(\boldsymbol{H}; \tilde{A}|\tilde{Y}) < \gamma_A. \quad (2)$$

where constraints on $H(\boldsymbol{H})$ encourage a diverse $\boldsymbol{H}$ to prevent representation collapse ($> \gamma_H^-$) and also limit its capacity ($< \gamma_H^+$) to avoid over-fitting. Another two symmetric terms, $I(\boldsymbol{H}; \tilde{Y}|\tilde{A})$ and $I(\boldsymbol{H}; \tilde{A}|\tilde{Y})$, mutually regularize posteriors to mitigate the negative impact of inherent noise on $\boldsymbol{H}$.

Note that MI terms like $I(\boldsymbol{H}; \tilde{A}|\tilde{Y})$ are usually intractable. Therefore, we introduce two practical implementations of RGIB, i.e., RGIB-SSL and RGIB-REP, based on different methodologies. The former explicitly optimizes the representation $\boldsymbol{H}$ with the self-supervised regularization, while the latter implicitly optimizes $\boldsymbol{H}$ by purifying the noisy $\tilde{A}$ and $\tilde{Y}$ with the reparameterization mechanism.

## 4.2 INSTANTIATING ROBUST GIB VIA SELF-SUPERVISED LEARNING

Recall that the graph representation is deteriorated with the *supervised* learning paradigm. Naturally, we modify it into a *self-supervised* counterpart by explicitly regularizing the representation $\boldsymbol{H}$ (shown in Figure 4(b)) to avoid collapse and to implicitly capture reliable relations among noisy edges, i.e.,

$$\min \text{RGIB-SSL} \triangleq -\underbrace{\lambda_s(I(\boldsymbol{H}_1; \tilde{Y}) + I(\boldsymbol{H}_2; \tilde{Y}))}_{\text{supervision}} - \underbrace{\lambda_u(H(\boldsymbol{H}_1) + H(\boldsymbol{H}_2))}_{\text{uniformity}} - \underbrace{\lambda_a I(\boldsymbol{H}_1; \boldsymbol{H}_2)}_{\text{alignment}}. \quad (3)$$

where margins $\lambda_s, \lambda_u, \lambda_a$ balance one supervised and two self-supervised regularization terms. When $\lambda_s \equiv 1$ and $\lambda_u \equiv 0$, the RGIB-SSL can be degenerated to the basic GIB.

Note that contrastive learning is a prevalent technique in the self-supervised area to learn robust representation (Chen et al., 2020), where contrasting pair of samples with data augmentation plays an essential role. In practice, we follow such a manner and calculate the supervision term by $\mathbb{E}[I(\boldsymbol{H}; \tilde{Y}|\tilde{A})] \leq \mathbb{E}[I(\boldsymbol{H}; \tilde{Y})] = \mathbb{E}_{\tilde{A}_s \sim \mathbb{P}(\tilde{A})}[I(\boldsymbol{H}_s; \tilde{Y})] \approx \frac{1}{2}(I(\boldsymbol{H}_1; \tilde{Y}) + I(\boldsymbol{H}_2; \tilde{Y})) = \frac{1}{2}(\mathcal{L}_{cls}(\boldsymbol{H}_1; \tilde{Y}) + \mathcal{L}_{cls}(\boldsymbol{H}_2; \tilde{Y}))$, where $\tilde{A}_1, \tilde{A}_2$ are sampled from $\mathbb{P}(\tilde{A})$. Similarly, we have $\mathbb{E}[H(\boldsymbol{H})] \approx \frac{1}{2}(H(\boldsymbol{H}_1) + H(\boldsymbol{H}_2))$, and a higher uniformity leads to a high entropy $H(\boldsymbol{H})$ as proved in Proposition 4.1. As for $I(\boldsymbol{H}; \tilde{A}|\tilde{Y})$, we approximate it by the alignment term $I(\boldsymbol{H}_1; \boldsymbol{H}_2)$ with the derived upper bound in Proposition 4.2.

**Proposition 4.1.** *A higher entropy $H(\boldsymbol{H})$ leads to a higher uniformity. Proof. See Appendix C.1.*

**Proposition 4.2.** *The alignment $I(\boldsymbol{H}_1; \boldsymbol{H}_2)$ is the upper bound of $I(\boldsymbol{H}; \tilde{A}|\tilde{Y})$. Since $I(\boldsymbol{H}; \tilde{A}|\tilde{Y}) \leq I(\boldsymbol{H}; \tilde{A}) \leq \frac{1}{2}(I(\boldsymbol{H}_1; \boldsymbol{H}_2) + I(\tilde{A}_1; \tilde{A}_2)) = \frac{1}{2}(I(\boldsymbol{H}_1; \boldsymbol{H}_2) + c)$, a constrained alignment estimated by MI $I(\boldsymbol{H}_1; \boldsymbol{H}_2)$ is capable to bound a lower $I(\boldsymbol{H}; \tilde{A}|\tilde{Y})$ and $I(\boldsymbol{H}; \tilde{A})$. Proof. See Appendix C.2.*

However, directly applying existing contrastive methods like (Chen et al., 2020; Khosla et al., 2020) can be suboptimal, since they are not originally designed for graph data and neglect the internal correlation of topology $\tilde{A}$ and target $\tilde{Y}$. Two following designs are proposed to avoid trivial solutions.

**Hybrid graph augmentation.** To encourage more diverse views with lower $I(A_1; A_2)$ and to avoid manual selection of augmentation operations, we propose a hybrid augmentation method with four augmentation operations as predefined candidates and ranges of their corresponding hyper-parameters.

In each training iteration, two augmentation operators, $\mathcal{T}^1(\cdot)$ and $\mathcal{T}^2(\cdot)$, and their hyper-parameters $\theta_1$ and $\theta_2$ are *automatically* sampled from the search space. Then, two augmented graphs are obtained by applying the two operators on the original graph $\mathcal{G}$. Namely, $\mathcal{G}^1 = \mathcal{T}^1(\mathcal{G}|\theta_1)$ and $\mathcal{G}^2 = \mathcal{T}^2(\mathcal{G}|\theta_2)$.

**Self-adversarial alignment & uniformity loss.** With representations $\boldsymbol{H}_1$ and $\boldsymbol{H}_2$ from two augmented views, we build the alignment objective by minimizing the representation similarity of the positive pairs $(\boldsymbol{h}_{ij}^1, \boldsymbol{h}_{ij}^2)$ and maximizing that of the randomly-sampled negative pairs $(\boldsymbol{h}_{ij}^1, \boldsymbol{h}_{mn}^2)$, $e_{ij} \neq e_{mn}$. The proposed self-adversarial alignment loss is $\mathcal{R}_{align} = \sum_{i=1}^{N} \mathcal{R}_i^{pos} + \mathcal{R}_i^{neg}$, where $\mathcal{R}_i^{pos} = p^{pos}(\boldsymbol{h}_{ij}^1, \boldsymbol{h}_{ij}^2) \cdot \left\| \boldsymbol{h}_{ij}^1 - \boldsymbol{h}_{ij}^2 \right\|_2^2$ and $\mathcal{R}_i^{neg} = p^{neg}(\boldsymbol{h}_{ij}^1, \boldsymbol{h}_{mn}^2) \cdot (\gamma - \left\| \boldsymbol{h}_{ij}^1 - \boldsymbol{h}_{mn}^2 \right\|_2^2)$. [2] Importantly, softmax functions $p^{pos}(\cdot)$ and $p^{neg}(\cdot)$ aim to mitigate the inefficiency problem (Sun et al., 2019) that aligned pairs are not informative. Besides, with Gaussian potential kernel, the uniformity loss is as $\mathcal{R}_{unif} = \sum_{ij,mn}^{K} e^{-\left\| \boldsymbol{h}_{ij}^1 - \boldsymbol{h}_{mn}^1 \right\|_2^2} + e^{-\left\| \boldsymbol{h}_{ij}^2 - \boldsymbol{h}_{mn}^2 \right\|_2^2}$, where $e_{ij}, e_{mn}$ are randomly sampled.

**Optimization.** As Eqn. 3, the overall loss function of RGIB-SSL is $\mathcal{L} = \lambda_s \mathcal{L}_{cls} + \lambda_a \mathcal{R}_{align} + \lambda_u \mathcal{R}_{unif}$.

**Remark 4.1.** *The inherent noise leads to class collapse, i.e., samples from the same class have the same representation. It comes from trivially minimizing the noisy supervision $I(\boldsymbol{H}; \tilde{Y})$ and results in degraded representations shown in Section 3.2. Fortunately, the alignment and uniformity terms regularizing representations can alleviate such noise effects and avoid collapse (see Section 5).*

## 4.3 Instantiating Robust GIB via Data Reparameterization

Another realization is by reparameterizing the graph data on both topology space and label space jointly to preserve clean information and discard noise. We propose RGIB-REP that explicitly models the reliability of $\tilde{A}$ and $\tilde{Y}$ via latent variables $\boldsymbol{Z}$ to learn a noise-resistant $\boldsymbol{H}$ (as Figure 4(c)), namely,

$$\min \text{RGIB-REP} \triangleq -\underbrace{\lambda_s I(\boldsymbol{H}; \boldsymbol{Z}_Y)}_{\text{supervision}} + \underbrace{\lambda_A I(\boldsymbol{Z}_A; \tilde{A})}_{\text{topology constraint}} + \underbrace{\lambda_Y I(\boldsymbol{Z}_Y; \tilde{Y})}_{\text{label constraint}}. \quad (4)$$

where latent variables $\boldsymbol{Z}_Y$ and $\boldsymbol{Z}_A$ are clean signals extracted from noisy $\tilde{Y}$ and $\tilde{A}$. Their complementary parts $\boldsymbol{Z}_{Y'}$ and $\boldsymbol{Z}_{A'}$ are considered as noise, satisfying $\tilde{Y} = \boldsymbol{Z}_Y + \boldsymbol{Z}_{Y'}$ and $\tilde{A} = \boldsymbol{Z}_A + \boldsymbol{Z}_{A'}$. When $\boldsymbol{Z}_Y \equiv \tilde{Y}$ and $\boldsymbol{Z}_A \equiv \tilde{A}$, the RGIB-REP can be degenerated to the basic GIB. Here, the $I(\boldsymbol{H}; \boldsymbol{Z}_Y)$ measures the supervised signals with selected samples $\boldsymbol{Z}_Y$, where the classifier takes $\boldsymbol{Z}_A$ (a subgraph of $\tilde{A}$) as input instead of the original $\tilde{A}$, i.e., $\boldsymbol{H} = f_{\boldsymbol{w}}(\boldsymbol{Z}_A, X)$. Constraints $I(\boldsymbol{Z}_A; \tilde{A})$ and $I(\boldsymbol{Z}_Y; \tilde{Y})$ aim to select the cleanest and most task-relevant information from $\tilde{A}$ and $\tilde{Y}$.

**Instantiation.** For deriving a tractable objective regarding $\boldsymbol{Z}_A$ and $\boldsymbol{Z}_Y$, a parameterized sampler $f_{\boldsymbol{\phi}}(\cdot)$ sharing the same architecture and weights as $f_{\boldsymbol{w}}(\cdot)$ is adopted here. $f_{\boldsymbol{\phi}}(\cdot)$ generates the probabilistic distribution of edges that include both $\tilde{A}$ and $\tilde{Y}$ by $\boldsymbol{P} = \sigma(\boldsymbol{H}_{\boldsymbol{\phi}} \boldsymbol{H}_{\boldsymbol{\phi}}^\top) \in (0, 1)^{|\mathcal{V}| \times |\mathcal{V}|}$, where hidden representations $\boldsymbol{H}_{\boldsymbol{\phi}} = f_{\boldsymbol{\phi}}(\tilde{A}, X)$. Then, the Bernoulli sampling is used to obtain edges of high confidences, i.e., $\boldsymbol{Z}_A = \text{SAMP}(\boldsymbol{P}|\tilde{A})$, $\boldsymbol{Z}_Y = \text{SAMP}(\boldsymbol{P}|\tilde{Y})$ where $|\boldsymbol{Z}_A| \leq |\tilde{A}|$ and $|\boldsymbol{Z}_Y| \leq |\tilde{Y}|$.

**Optimization.** A relaxation is then conducted on the three MI terms in Eqn. 4. With derived bounds in Proposition 4.3 (i.e., $\mathcal{R}_A$ and $\mathcal{R}_Y$) and Proposition 4.4 (i.e., $\mathcal{L}_{cls}$), the final objective is formed as $\mathcal{L} = \lambda_s \mathcal{L}_{cls} + \lambda_A \mathcal{R}_A + \lambda_Y \mathcal{R}_Y$, and the corresponding guarantee is discussed in Theorem 4.5.

**Proposition 4.3.** *Given the edge number $n$ of $\tilde{A}$, the marginal distribution of $\boldsymbol{Z}_A$ is $\mathbb{Q}(\boldsymbol{Z}_A) = \sum_n \mathbb{P}_{\boldsymbol{\phi}}(\boldsymbol{P}|n)\mathbb{P}(\tilde{A} = n) = \mathbb{P}(n) \prod_{\tilde{A}_{ij}=1}^{n} \boldsymbol{P}_{ij}$. Then, we derive the upper bound $I(\boldsymbol{Z}_A; \tilde{A}) \leq \mathbb{E}[KL(\mathbb{P}_{\boldsymbol{\phi}}(\boldsymbol{Z}_A|A)||\mathbb{Q}(\boldsymbol{Z}_A))] = \sum_{e_{ij} \in \tilde{A}} \boldsymbol{P}_{ij} \log \frac{\boldsymbol{P}_{ij}}{\tau} + (1 - \boldsymbol{P}_{ij}) \log \frac{1 - \boldsymbol{P}_{ij}}{1 - \tau} = \mathcal{R}_A$, where $\tau$ is a constant. Similarly, the label constraint is bounded as $I(\boldsymbol{Z}_Y; \tilde{Y}) \leq \mathcal{R}_Y$, Proof. See Appendix C.3.*

**Proposition 4.4.** *Since $I(\boldsymbol{H}; \boldsymbol{Z}_Y) \geq \mathbb{E}_{\boldsymbol{Z}_Y, \boldsymbol{Z}_A}[\log \mathbb{P}_{\boldsymbol{w}}(\boldsymbol{Z}_Y|\boldsymbol{Z}_A)] \approx -\mathcal{L}_{cls}(f_{\boldsymbol{w}}(\boldsymbol{Z}_A), \boldsymbol{Z}_Y)$, the supervision term in Eqn. 4 can be empirically reduced to the classification loss. Proof. See Appendix C.4.*

**Theorem 4.5.** *Assume the noisy training data $D_{train} = (\tilde{A}, X, \tilde{Y})$ contains a potentially clean subset $D_{sub} = (\boldsymbol{Z}_A^*, X, \boldsymbol{Z}_Y^*)$ and $\boldsymbol{Z}_Y^* \approx Y$, based on which a trained GNN predictor $f_{\boldsymbol{w}}(\cdot)$ satisfies $f_{\boldsymbol{w}}(\boldsymbol{Z}_A^*, X) = \boldsymbol{Z}_Y^* + \epsilon$. The error $\epsilon$ is independent from $D_{sub}$ and $\epsilon \to 0$. Then, for any $\lambda_s, \lambda_A, \lambda_Y \in [0, 1]$, $\boldsymbol{Z}_A = \boldsymbol{Z}_A^*$ and $\boldsymbol{Z}_Y = \boldsymbol{Z}_Y^*$ minimizes the RGIB-REP (Eqn. 4). Proof. See Appendix C.5.*

---

[2]Specifically, $p^{pos}(\boldsymbol{h}_{ij}^1, \boldsymbol{h}_{ij}^2) = \frac{\exp(\left\| \boldsymbol{h}_{ij}^1 - \boldsymbol{h}_{ij}^2 \right\|_2^2)}{\sum_{i=1}^{N} \exp(\left\| \boldsymbol{h}_{ij}^1 - \boldsymbol{h}_{ij}^2 \right\|_2^2)}$, and $p^{neg}(\boldsymbol{h}_{ij}^1, \boldsymbol{h}_{mn}^2) = \frac{\exp(\alpha - \left\| \boldsymbol{h}_{ij}^1 - \boldsymbol{h}_{mn}^2 \right\|_2^2)}{\sum_{i=1}^{N} \exp(\alpha - \left\| \boldsymbol{h}_{ij}^1 - \boldsymbol{h}_{mn}^2 \right\|_2^2)}$.

Table 2: Method comparison with GCN under inherent noise, i.e., both the input and label noise exist. The **boldface** numbers mean the best results, while the underlines indicate the second-best results.

| method | Cora | | | Citeseer | | | Pubmed | | | Facebook | | | Chameleon | | | Squirrel | | |
|---|---|---|---|---|---|---|---|---|---|---|---|---|---|---|---|---|---|---|
| | 20% | 40% | 60% | 20% | 40% | 60% | 20% | 40% | 60% | 20% | 40% | 60% | 20% | 40% | 60% | 20% | 40% | 60% |
| Standard | .8111 | .7419 | .6970 | .7864 | .7380 | .7085 | .8870 | .8748 | .8641 | .9829 | .9520 | .9438 | .9616 | .9496 | .9274 | .9432 | .9406 | .9386 |
| DropEdge | .8017 | .7423 | .7303 | .7635 | .7393 | .7094 | .8711 | .8482 | .8354 | .9811 | .9682 | .9473 | .9568 | .9548 | .9407 | .9439 | .9377 | .9365 |
| NeuralSparse | .8190 | .7318 | .7293 | .7765 | .7397 | .7148 | .8908 | .8733 | .8630 | .9825 | .9638 | .9456 | .9599 | .9497 | .9402 | .9494 | .9309 | .9297 |
| PTDNet | .8047 | .7559 | .7388 | .7795 | .7423 | .7283 | .8872 | .8733 | .8623 | .9725 | .9674 | .9485 | .9607 | .9514 | .9424 | .9485 | .9326 | .9304 |
| Co-teaching | .8197 | .7479 | .7030 | .7533 | .7238 | .7131 | .8943 | .8760 | .8638 | .9820 | .9526 | .9480 | .9595 | .9516 | .9483 | .9461 | .9352 | .9374 |
| Peer loss | .8185 | .7468 | .7018 | .7423 | .7345 | .7104 | .8961 | .8815 | .8566 | .9807 | .9536 | .9430 | .9543 | .9533 | .9267 | .9457 | .9345 | .9286 |
| Jaccard | .8143 | .7498 | .7024 | .7473 | .7324 | .7107 | .8872 | .8803 | .8512 | .9794 | .9579 | .9428 | .9503 | .9538 | .9344 | .9443 | .9327 | .9244 |
| GIB | .8198 | .7485 | .7148 | .7509 | .7388 | .7121 | .8899 | .8729 | .8544 | .9773 | .9608 | .9417 | .9554 | .9561 | .9321 | .9472 | .9329 | .9302 |
| SupCon | .8240 | .7819 | .7490 | .7554 | .7458 | .7299 | .8853 | .8718 | .8525 | .9588 | .9508 | .9297 | .9561 | .9531 | .9467 | .9473 | .9348 | .9301 |
| GRACE | .7872 | .6940 | .6929 | .7632 | .7242 | .6844 | .8922 | .8749 | .8588 | .8899 | .8865 | .8315 | .8978 | .8987 | .8949 | .9394 | .9380 | .9363 |
| **RGIB-REP** | .8313 | .7966 | .7591 | .7875 | .7519 | .7312 | .9017 | .8834 | .8652 | **.9832** | **.9770** | .9519 | **.9723** | **.9621** | **.9519** | **.9509** | **.9455** | **.9434** |
| **RGIB-SSL** | **.8930** | **.8554** | **.8339** | **.8694** | **.8427** | **.8137** | **.9225** | **.8918** | **.8697** | .9829 | .9711 | **.9643** | .9655 | .9592 | .9500 | .9499 | .9426 | .9425 |

## 5 EXPERIMENTS

**Setup.** In this section, we empirically verify the effectiveness of the proposed RGIB framework. 6 popular datasets and 3 GNNs are taken in the experiments. The inherent edge noise is generated based on Definition. 3.1 after the commonly used data split where 85% edges are randomly selected for training, 5% as the validation set, and 10% for testing. The AUC is used as the evaluation metric as in (Zhang & Chen, 2018; Zhu et al., 2021). The software framework is the Pytorch (Paszke et al., 2017), while the hardware platform is a single NVIDIA RTX 3090 GPU. We repeat all the experiments 10 times to obtain evaluation results with mean values and standard deviations. [3]

**Baselines.** As existing robust methods separately deal with input noise or label noise, both kinds of methods should be considered as baselines. For input noise, three sampling-based approaches are used for comparison, i.e., DropEdge (Rong et al., 2020), NeuralSparse (Zheng et al., 2020), and PTDNet (Luo et al., 2021). Jaccard (Wu et al., 2019), GIB (Wu et al., 2020), VIB (Sun et al., 2022), and PRI (Yu et al., 2022), which are designed for pruning adversarial edges, are also included. Two generic methods are selected for label noise, i.e., Co-teaching (Han et al., 2018) and Peer loss (Liu & Guo, 2020). In addition, two contrastive learning methods are taken into comparison with adaptation to the link prediction task, including SupCon (Khosla et al., 2020) utilizing the full labels for supervision and GRACE (Zhu et al., 2020) that are optimized in a self-supervised manner without labels. All the above baseline methods are evaluated w.r.t. their original implementations

### 5.1 MAIN RESULTS

**Performance comparison.** As shown in Table 2, the RGIB achieves the best results in all 6 datasets under the inherent edge noise with various noise ratios, especially on the more challenging datasets, i.e., Cora and Citeseer, where a 12.9% AUC promotion can be gained compared with the second-best methods. When it comes to decoupled noise settings shown in Table 3, RGIB also surpasses all the baselines ad hoc for input noise or label noise by a large margin.

**Remark 5.1.** *The two instantiations of RGIB can be generalized to different scenarios with their own priority w.r.t. intrinsic graph properties that can be complementary to each other with flexible options in practice. Basically, the RGIB-SSL is more adaptive to sparser graphs, e.g., Cora and Citeseer, where the inherent edge noise presents a considerable challenge and results in greater performance degradation. The RGIB-REP can be more suitable for denser graphs, e.g., Facebook and Chameleon. Meanwhile, two instantiations also work effectively on heterogeneous graphs with low homophily.*

**The learned representation distribution.** Next, we justify that the proposed method can effectively alleviate the representation collapse. Compared with the standard training, both RGIB-REP and RGIB-SSL bring significant improvements for the alignment with much lower values, as in Table 4. At the same time, the uniformity of learned representation is also enhanced: it can be seen from Figure 5 that the various query edges tend to be more uniformly distributed on the unit circle. As RGIB-SSL explicitly constrains the representation, its recovery power on representation distribution is naturally stronger than RGIB-REP, resulting in comparably better alignment and uniformity measures.

---

[3]Full evaluation results can be found in Appendix F.

Table 3: Method comparison with GCN under decoupled input noise (upper) or label noise (lower).

| input noise | Cora | | | Citeseer | | | Pubmed | | | Facebook | | | Chameleon | | | Squirrel | | |
|---|---|---|---|---|---|---|---|---|---|---|---|---|---|---|---|---|---|---|
| | 20% | 40% | 60% | 20% | 40% | 60% | 20% | 40% | 60% | 20% | 40% | 60% | 20% | 40% | 60% | 20% | 40% | 60% |
| Standard | .8027 | .7856 | .7490 | .8054 | .7708 | .7583 | .8854 | .8759 | .8651 | .9819 | .9668 | .9622 | .9608 | .9433 | .9368 | .9416 | .9395 | .9411 |
| DropEdge | .8338 | .7826 | .7454 | .8025 | .7730 | .7473 | .8682 | .8456 | .8376 | .9803 | .9685 | .9531 | .9567 | .9433 | .9432 | .9426 | .9376 | .9358 |
| NeuralSparse | .8534 | .7794 | .7637 | .8093 | .7809 | .7468 | .8931 | .8720 | .8649 | .9712 | .9691 | .9583 | .9609 | .9540 | .9348 | .9469 | .9403 | .9417 |
| PTDNet | .8433 | .8214 | .7770 | .8119 | .7811 | .7638 | .8903 | .8776 | .8609 | .9725 | .9668 | .9493 | .9610 | .9457 | .9360 | .9469 | .9400 | .9379 |
| Co-teaching | .8045 | .7871 | .7530 | .8059 | .7753 | .7668 | .8931 | .8792 | .8606 | .9712 | .9707 | .9714 | .9524 | .9446 | .9447 | .9462 | .9425 | .9306 |
| Peer loss | .8051 | .7866 | .7517 | .8106 | .7767 | .7653 | .8917 | .8811 | .8643 | .9758 | .9703 | .9622 | .9558 | .9482 | .9412 | .9362 | .9386 | .9336 |
| Jaccard | .8200 | .7838 | .7617 | .8176 | .7776 | .7725 | .8987 | .8764 | .8639 | .9784 | .9702 | .9638 | .9507 | .9436 | .9364 | .9388 | .9345 | .9240 |
| GIB | .8002 | .8099 | .7741 | .8070 | .7717 | .7798 | .8932 | .8808 | .8618 | .9796 | .9647 | .9650 | .9605 | .9521 | .9416 | .9390 | .9406 | .9397 |
| SupCon | .8349 | .8301 | .8025 | .8076 | .7767 | .7655 | .8867 | .8739 | .8558 | .9647 | .9517 | .9401 | .9606 | .9536 | .9468 | .9372 | .9343 | .9305 |
| GRACE | .7877 | .7107 | .6975 | .7615 | .7151 | .6830 | .8810 | .8795 | .8593 | .9015 | .8833 | .8395 | .8994 | .9007 | .8964 | .9392 | .9378 | .9363 |
| **RGIB-REP** | .8624 | .8313 | .8158 | .8299 | .7996 | .7771 | .9008 | .8822 | .8687 | **.9833** | **.9723** | **.9682** | .9705 | .9604 | .9480 | .9495 | .9432 | .9405 |
| **RGIB-SSL** | **.9024** | **.8577** | **.8421** | **.8747** | **.8461** | **.8245** | **.9126** | **.8889** | **.8693** | .9821 | .9707 | .9668 | .9658 | .9570 | **.9486** | .9479 | .9429 | **.9429** |

| label noise | Cora | | | Citeseer | | | Pubmed | | | Facebook | | | Chameleon | | | Squirrel | | |
|---|---|---|---|---|---|---|---|---|---|---|---|---|---|---|---|---|---|---|
| | 20% | 40% | 60% | 20% | 40% | 60% | 20% | 40% | 60% | 20% | 40% | 60% | 20% | 40% | 60% | 20% | 40% | 60% |
| Standard | .8281 | .8054 | .8060 | .7965 | .7850 | .7659 | .9030 | .9039 | .9070 | .9882 | .9880 | .9886 | .9686 | .9580 | .9362 | .9720 | .9720 | .9710 |
| DropEdge | .8363 | .8273 | .8148 | .7937 | .7853 | .7632 | .9313 | .9201 | .9240 | .9673 | .9771 | .9776 | .9580 | .9579 | .9578 | .9608 | .9603 | .9698 |
| NeuralSparse | .8524 | .8246 | .8211 | .7968 | .7921 | .7752 | .9272 | .9136 | .9089 | .9781 | .9781 | .9784 | .9583 | .9583 | .9571 | .9633 | .9626 | .9625 |
| PTDNet | .8460 | .8214 | .8138 | .7968 | .7765 | .7622 | .9219 | .9099 | .9093 | .9879 | .9880 | .9783 | .9585 | .9576 | .9665 | .9633 | .9623 | .9626 |
| Co-teaching | .8446 | .8209 | .8157 | .7974 | .7877 | .7913 | .9315 | .9291 | .9319 | .9762 | .9797 | .9638 | .9642 | .9650 | .9533 | .9675 | .9641 | .9655 |
| Peer loss | .8325 | .8036 | .8069 | .7991 | .7990 | .7751 | .9126 | .9101 | .9210 | .9769 | .9750 | .9734 | .9621 | .9501 | .9569 | .9636 | .9694 | .9696 |
| Jaccard | .8289 | .8064 | .8148 | .8061 | .7887 | .7689 | .9098 | .9135 | .9096 | .9702 | .9725 | .9758 | .9603 | .9659 | .9557 | .9529 | .9512 | .9501 |
| GIB | .8337 | .8137 | .8157 | .7986 | .7852 | .7649 | .9037 | .9114 | .9064 | .9742 | .9703 | .9771 | .9651 | .9582 | .9489 | .9641 | .9628 | .9601 |
| SupCon | .8491 | .8275 | .8256 | .8024 | .7983 | .7807 | .9131 | .9108 | .9162 | .9647 | .9567 | .9553 | .9584 | .9580 | .9477 | .9516 | .9595 | .9511 |
| GRACE | .8531 | .8237 | .8193 | .7909 | .7630 | .7737 | .9234 | .9252 | .9255 | .8913 | .8972 | .8887 | .9053 | .9074 | .9075 | .9171 | .9174 | .9166 |
| **RGIB-REP** | .8554 | .8318 | .8297 | .8083 | .7846 | .7945 | .9357 | .9343 | .9332 | **.9884** | **.9883** | **.9889** | **.9785** | **.9797** | **.9785** | **.9735** | **.9733** | **.9737** |
| **RGIB-SSL** | **.9314** | **.9224** | **.9241** | **.9204** | **.9218** | **.9250** | **.9594** | **.9604** | **.9613** | .9857 | .9881 | .9857 | .9730 | .9752 | .9744 | .9727 | .9729 | .9726 |

Table 5: Comparison on different schedulers. Here, SSL/REP are short for RGIB-SSL/RGIB-REP. Experiments are performed with a 4-layer GAT and $\epsilon = 40\%$ inherent edge noise.

| dataset | Cora | | Citeseer | | Pubmed | | Facebook | | Chameleon | | Squirrel | |
|---|---|---|---|---|---|---|---|---|---|---|---|---|
| method | SSL | REP | SSL | REP | SSL | REP | SSL | REP | SSL | REP | SSL | REP |
| $constant$ | .8398 | **.7927** | **.8227** | **.7742** | .8596 | **.8416** | **.9694** | **.9778** | .9384 | **.9498** | .9293 | **.9320** |
| $linear(\cdot)$ | .8427 | .7653 | .8167 | .7559 | **.8645** | .8239 | .9669 | .9529 | .9434 | .9369 | .9316 | .9265 |
| $sin(\cdot)$ | **.8436** | .7924 | .8132 | .7680 | .8637 | .8275 | .9685 | .9594 | **.9447** | .9434 | **.9325** | .9282 |
| $cos(\cdot)$ | .8334 | .7833 | .8088 | .7647 | .8579 | .8372 | .9477 | .9629 | .9338 | .9444 | .9251 | .9303 |
| $exp(\cdot)$ | .8381 | .7815 | .8085 | .7569 | .8617 | .8177 | .9471 | .9613 | .9402 | .9418 | .9316 | .9299 |

Table 4: Alignment comparison. Here, std. is short for *standard training*, and SSL/REP are short for RGIB-SSL/RGIB-REP, respectively.

| dataset | Cora | | | Citeseer | | |
|---|---|---|---|---|---|---|
| method | std. | REP | SSL | std. | REP | SSL |
| clean | .616 | .524 | **.475** | .445 | .439 | **.418** |
| $\varepsilon=20\%$ | .687 | .642 | **.543** | .586 | .533 | **.505** |
| $\varepsilon=40\%$ | .695 | .679 | **.578** | .689 | .623 | **.533** |
| $\varepsilon=60\%$ | .732 | .704 | **.615** | .696 | .647 | **.542** |

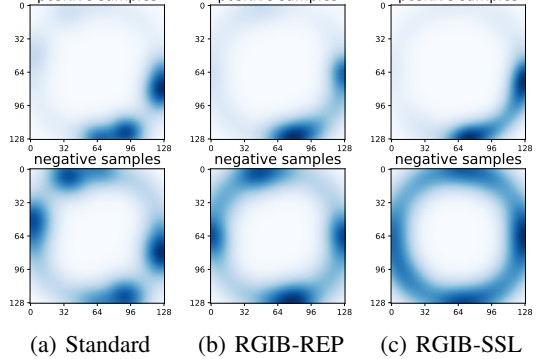

(a) Standard  (b) RGIB-REP  (c) RGIB-SSL

Figure 5: Uniformity distribution on Citeseer with $\varepsilon = 40\%$.

**Optimization schedulers.** To reduce the search cost of coefficients in objectives of RGIB-SSL and RGIB-REP, we setup a unified optimization framework that can be formed as $\mathcal{L} = \alpha\mathcal{L}_{cls} + (1-\alpha)\mathcal{R}_1 + (1-\alpha)\mathcal{R}_2$. Here, we attempt 5 different schedulers to tune the only hyper-parameter $\alpha \in [0, 1]$, including (1) constant, $\alpha \equiv c$; (2) linear, $\alpha_t = k \cdot t$, where $t$ is the normalized time step; (3) sine, $\alpha_t = \sin(t \cdot \pi/2)$; (4) cosine, $\alpha_t = \cos(t \cdot \pi/2)$; and (5) exponential, $\alpha_t = e^{k \cdot t}$. As empirical results summarized in Table 5, the selection of optimization schedulers greatly influences the final results. Although there is no gold scheduler that consistently performs the best, the *constant* and *sine* schedulers are generally better than others among the 5 above candidates. A further hyper-parameter study with grid search of the $\lambda$s in RGIB-SSL and RGIB-REP can be found in Appendix F.2.

Table 6: Ablation study for RGIB-SSL and RGIB-REP with a 4-layer SAGE. Here, $\epsilon = 60\%$ indicates the $60\%$ coupled inherent noise, while the $\epsilon_a/\epsilon_y$ represent ratios of decoupled input/label noise.

| variant | Cora | | | Chameleon | | |
|---|---|---|---|---|---|---|
| | $\epsilon = 60\%$ | $\epsilon_a = 60\%$ | $\epsilon_y = 60\%$ | $\epsilon = 60\%$ | $\epsilon_a = 60\%$ | $\epsilon_y = 60\%$ |
| RGIB-SSL (full) | .8596 | .8730 | .8994 | .9663 | .9758 | .9762 |
| - w/o hybrid augmentation | .8150 (5.1%↓) | .8604 (1.4%↓) | .8757 (2.6%↓) | .9528 (1.3%↓) | .9746 (0.1%↓) | .9695 (0.6%↓) |
| - w/o self-adversarial | .8410 (2.1%↓) | .8705 (0.2%↓) | .8927 (0.7%↓) | .9655 (0.1%↓) | .9732 (0.2%↓) | .9746 (0.1%↓) |
| - w/o supervision ($\lambda_s = 0$) | .7480 (12.9%↓) | .7810 (10.5%↓) | .7820 (13.0%↓) | .8626 (10.7%↓) | .8628 (11.5%↓) | .8512 (12.8%↓) |
| - w/o alignment ($\lambda_a = 0$) | .8194 (4.6%↓) | .8510 (2.5%↓) | .8461 (5.9%↓) | .9613 (0.5%↓) | .9749 (0.1%↓) | .9722 (0.4%↓) |
| - w/o uniformity ($\lambda_u = 0$) | .8355 (2.8%↓) | .8621 (1.2%↓) | .8878 (1.3%↓) | .9652 (0.1%↓) | .9710 (0.4%↓) | .9751 (0.1%↓) |
| RGIB-REP (full) | .7611 | .8487 | .8095 | .9567 | .9706 | .9676 |
| - w/o edge selection ($Z_A \equiv \tilde{A}$) | .7515 (1.2%↓) | .8199 (3.3%↓) | .7890 (2.5%↓) | .9554(0.1%↓) | .9704 (0.1%↓) | .9661 (0.1%↓) |
| - w/o label selection ($Z_Y \equiv \tilde{Y}$) | .7533 (1.0%↓) | .8373 (1.3%↓) | .7847 (3.0%↓) | .9484(0.8%↓) | .9666 (0.4%↓) | .9594 (0.8%↓) |
| - w/o topology constraint ($\lambda_A = 0$) | .7355 (3.3%↓) | .7699 (9.2%↓) | .7969 (1.5%↓) | .9503(0.6%↓) | .9658 (0.4%↓) | .9635 (0.4%↓) |
| - w/o label constraint ($\lambda_Y = 0$) | .7381 (3.0%↓) | .8106 (4.4%↓) | .8032 (0.7%↓) | .9443(1.2%↓) | .9665 (0.4%↓) | .9669 (0.1%↓) |

## 5.2 ABLATION STUDY

In this part, we conduct a thorough ablation study for the RGIB framework. As shown in Table 6, each component contributes to the final performance. For RGIB-SSL, we have the following analysis:

- **Hybrid augmentation.** RGIB-SSL benefits from the hybrid augmentation algorithm that automatically generates graphs of high diversity for contrast. Compared with fixed augmentation, the hybrid augmentation brings consistent improvements with a $3.0\%$ average AUC promotion on Cora.

- **Self-adversarial alignment loss.** Randomly-sampled pairs are with hierarchical information to be learned from, and the proposed re-weighting technique further enhances high-quality pairs and decreases low-quality counterparts. It enables to discriminate the more informative contrasting pairs and thus refines the alignment signal for optimization, bringing to up $2.1\%$ AUC promotion.

- **Information constraints.** Label supervision contributes the most among the three informative terms, even with label noise. Degenerating RGIB-SSL to a pure self-supervised manner without supervision (i.e., $\lambda_s = 0$) leads to an average $11.9\%$ AUC drop. Meanwhile, we show that three regularization terms can be jointly optimized while another two terms are also of significant values.

As for RGIB-REP, its sample selection mechanism and corresponding constraints are also essential:

- **Edge / label selection.** Two sample selection methods are near equally important for learning with the coupled inherent noise that both informative sources are required to be purified. Besides, the edge selection is more important for tackling the decoupled input noise, as a greater drop will come when removed. Similarly, the label selection plays a dominant role in handling the label noise.

- **Topological / label constraint.** Table 6 also shows that the selection mechanism should be regularized by the related constraint; and otherwise, sub-optimal solutions will be achieved. Besides, the topological constraint is generally more sensitive than the label constraint for RGIB-REP.

## 6 CONCLUSION

In this work, we study the problem of link prediction with the inherent edge noise and reveal that the graph representation is severely collapsed under such a coupled noise. Based on the observation, we introduce the Robust Graph Information Bottleneck (RGIB) principle, aiming to extract reliable signals via decoupling and balancing the mutual information among inputs, labels, and representation to enhance the robustness and avoid collapse. Regarding the instantiation of RGIB, the self-supervised learning technique and data reparametrization mechanism are utilized to establish the RGIB-SSL and RGIB-REP, respectively. Empirical studies on 6 datasets and 3 GNNs verify the denoising effect of the proposed RGIB under different noisy scenarios. In future work, we will generalize RGIB to more scenarios in the link prediction domain, e.g., multi-hop reasoning on knowledge graphs with multi-relational edges that characterize more diverse and complex patterns.

REPRODUCIBILITY STATEMENT

To ensure the reproducibility of the empirical result in this work, we have stated all the details about our experiments in corresponding contents of Section 3 and Section 5. The implementation details are further elaborated in Appendix E. Besides, we will provide an anonymous Github repository for the source codes during the discussion phase for the reviewers of ICLR 2023.

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

# Appendix

## Table of Contents

## A    BROADER IMPACT

In this work, we propose the RGIB framework for denoising the inherent edge noise in graphs and for improving the robustness of link prediction. We conceptually derive the RGIB and empirically justify that it enables the various GNNs to learn the input-invariant and label-invariant graph representations, preventing representation collapse and obtaining superior performances against edge noise. Our objective is to protect and enhance the current graph models, and we do not think our work would have negative societal impacts.

A similar problem has been noticed in an early work (Gallagher et al., 2008), where robots tend to build connections with normal users to spread misinformation on social networks, yielding the degeneration of GNNs for robot detection. In addition, as pointed out by a recent survey (Wu et al., 2022b), the inherent noise can be produced in the data generation process that requires manually annotating the data. As introduced in (Wu et al., 2022b), the inherent noise here refers to irreducible noises in graph structures, attributes, and labels. In short, the studied edge noise in our work is in line with the generic concept of graph inherent noise in the literature, which is recognized as a common problem in practice but also an under-explored challenge in both academic and industrial scenes.

However, several existing benchmarks, e.g., Cora and Citeseer, are generally clean and without annotated noisy edges. Inevitably, there is a gap when one would like to study the influence of inherent noise on these common benchmarks. Thus, to fill this gap, it is necessary to simulate the inherent edge noise properly to investigate the impact of noise. If the inherent edge noise exists, it should be false positive samples, while the false negative samples are often intractable to be collected. Thus, we focus on the investigation of the false positive edges as the inherent noise and design the simulation approach in Definition 3.1 as it is more close to the real-world scenarios.

Besides, to our best knowledge, we are the first to study the robustness problem of link prediction under the inherent edge noise. One of our major contributions to this research problem is that we reveal that the inherent noise can bring a severe representation collapse and performance degradation, and such negative impacts are general to common datasets and GNNs.

What's more, it is also possible that new instantiations based on other kinds of methodology are inspired by the robust GIB principle. We believe that such a bidirectional information bottleneck that strictly treats the information source on both input side and label side is helpful in practice, especially for extremely noisy scenarios.

## B    NOTATIONS

We summarize the frequently used notations in Table 7.

Table 7: The most frequently used notations in this paper.

| notations | meanings |
|---|---|
| $\mathcal{V} = \{v_i\}_{i=1}^{N}$ | the set of nodes |
| $\mathcal{E} = \{e_{ij}\}_{ij=1}^{M}$ | the set of edges |
| $A \in \{0,1\}^{N \times N}$ | the adjacent matrix with binary elements |
| $X \in \mathbb{R}^{N \times D}$ | the node features |
| $\mathcal{G} = (A, X)$ | the input graph of a GNN |
| $\mathcal{E}_{query} = \{e_{ij}\}_{ij=1}^{K}$ | the query edges to be predicted |
| $Y$ | the labels of the query edges |
| $\boldsymbol{h}_{ij}$ | representation of a query edge $e_{ij}$ |
| $\boldsymbol{H}$ | representation of all the query edges |
| $I(X;Y)$ | the mutual information of X and Y |
| $I(X;Y|Z)$ | the conditional mutual information of $X$ and $Y$ when observing $Z$ |

## C   THEORETICAL JUSTIFICATION

### C.1   PROOF FOR PROPOSITION 4.1

*Proof.* As introduced in Section 4.2, the uniformity loss on two augmented graphs is formed as $\mathcal{R}_{unif} = \sum_{ij,mn}^{K} e^{-\|\boldsymbol{h}_{ij}^1 - \boldsymbol{h}_{mn}^1\|_2^2} + e^{-\|\boldsymbol{h}_{ij}^2 - \boldsymbol{h}_{mn}^2\|_2^2}$. For simplicity, we can reduce it to one graph, i.e., $\mathcal{R}_{unif} = \mathbb{E}_{ij,mn \sim p_{\mathcal{E}}} [e^{-\|\boldsymbol{h}_{ij} - \boldsymbol{h}_{mn}\|_2^2}]$. Next, we obtain its upper bound by the following derivation.

$$
\begin{aligned}
\log \mathcal{R}_{unif} &= \log \mathbb{E}_{ij,mn \sim p_{\mathcal{E}}} [e^{-\|\boldsymbol{h}_{ij} - \boldsymbol{h}_{mn}\|_2^2}] \\
&= \log \mathbb{E}_{ij,mn \sim p_{\mathcal{E}}} [e^{2t \cdot \boldsymbol{h}_{ij}^T \boldsymbol{h}_{mn} - 2t}] \\
&\leq \mathbb{E}_{ij \sim p_{\mathcal{E}}} \big[ \log \mathbb{E}_{mn \sim p_{\mathcal{E}}} [e^{2t \cdot \boldsymbol{h}_{ij}^T \boldsymbol{h}_{mn}}] \big] \\
&= \frac{1}{N} \sum_{ij}^{N} \big[ \log \sum_{mn}^{N} [e^{2t \cdot \boldsymbol{h}_{ij}^T \boldsymbol{h}_{mn}}] \big] \\
&= \frac{1}{N} \sum_{ij}^{N} \log \hat{p}_{\text{vMF-KDE}}(\boldsymbol{h}_{ij}) + \log Z_{\text{vMF}} \\
&\triangleq -\hat{H}(\boldsymbol{H}) + \log Z_{\text{vMF}}
\end{aligned}
\tag{5}
$$

Here, $\hat{p}_{\text{vMF-KDE}}(\cdot)$ is a von Mises-Fisher (vMF) kernel density estimation based on $N$ training samples, and $Z_{\text{vMF}}$ is the vMF normalization constant. $\hat{H}(\cdot)$ is the resubstitution entropy estimator of $\boldsymbol{H} = f_{\boldsymbol{w}}(\cdot)$ (Ahmad & Lin, 1976). As the uniformity loss $\mathcal{L}_{unif}$ can be approximated by entropy $\hat{H}(\boldsymbol{H})$, a higher entropy $H(\boldsymbol{H})$ indicates a lower uniformity loss, i.e., a higher uniformity. □

### C.2   PROOF FOR PROPOSITION 4.2

*Proof.* Recall the hybrid graph augmentation technique is adopted by RGIB-SSL for contrasting, we can approximate the expectation $\mathbb{E}[I(\boldsymbol{H}; \tilde{A}|\tilde{Y})]$ by $1/N \sum_{i=1}^{N} I(\boldsymbol{H}_i; \tilde{A}_i|\tilde{Y})$ with $N$ augmented graphs. Based on this, we drive the upper bound of $I(\boldsymbol{H}; \tilde{A}|\tilde{Y})$ as follows.

$$
\begin{aligned}
I(\boldsymbol{H}; \tilde{A}|\tilde{Y}) \leq I(\boldsymbol{H}; \tilde{A}) &= 1/N \sum_{i=1}^{N} \big( I(\boldsymbol{H}_i; \tilde{A}_i) \big) \\
&\approx 1/2 \big( I(\boldsymbol{H}_1; \tilde{A}_1) + I(\boldsymbol{H}_2; \tilde{A}_2) \big) \\
&= 1/2 \big( H(\boldsymbol{H}_1) + H(\tilde{A}_1) + H(\boldsymbol{H}_2) + H(\tilde{A}_2) - H(\boldsymbol{H}_1, \tilde{A}_1) - H(\boldsymbol{H}_2, \tilde{A}_2) \big) \\
&\leq 1/2 \big( H(\boldsymbol{H}_1) + H(\tilde{A}_1) + H(\boldsymbol{H}_2) + H(\tilde{A}_2) - H(\boldsymbol{H}_1, \boldsymbol{H}_2) - H(\tilde{A}_1, \tilde{A}_2) \big) \\
&= 1/2 \big( I(\boldsymbol{H}_1; \boldsymbol{H}_2) + I(\tilde{A}_1; \tilde{A}_2) \big) \\
&= 1/2 \big( I(\boldsymbol{H}_1; \boldsymbol{H}_2) + c \big)
\end{aligned}
\tag{6}
$$

Thus, a lower $I(\boldsymbol{H}_1; \boldsymbol{H}_2)$ can upper bounded as lower $I(\boldsymbol{H}; \tilde{A})$ and $I(\boldsymbol{H}; \tilde{A}|\tilde{Y})$. □

### C.3 PROOF FOR PROPOSITION 4.3

*Proof.* With the marginal distribution $\mathbb{Q}(\boldsymbol{Z}_A) = \sum_n \mathbb{P}_\phi(\boldsymbol{P}|n)\mathbb{P}(\tilde{A}=n) = \mathbb{P}(n)\prod_{\tilde{A}_{ij}=1}^n \boldsymbol{P}_{ij}$, we drive the upper bound of MI $I(\boldsymbol{Z}_A; \tilde{A})$ as:

$$
\begin{aligned}
I(\boldsymbol{Z}_A; \tilde{A}) =& \mathbb{E}_{\boldsymbol{Z}_A, \tilde{A}}[\log(\frac{\mathbb{P}(\boldsymbol{Z}_A|\tilde{A})}{\mathbb{P}(\boldsymbol{Z}_A)})] \\
=& \mathbb{E}_{\boldsymbol{Z}_A, \tilde{A}}[\log(\frac{\mathbb{P}_\phi(\boldsymbol{Z}_A|\tilde{A})}{\mathbb{Q}(\boldsymbol{Z}_A)})] - \mathrm{KL}(\mathbb{P}(\boldsymbol{Z}_A)||\mathbb{Q}(\boldsymbol{Z}_A)) \\
\leq& \mathbb{E}[\mathrm{KL}(\mathbb{P}_\phi(\boldsymbol{Z}_A|A)||\mathbb{Q}(\boldsymbol{Z}_A))] \\
=& \sum_{e_{ij} \in \tilde{A}} \boldsymbol{P}_{ij} \log \frac{\boldsymbol{P}_{ij}}{\tau} + (1 - \boldsymbol{P}_{ij}) \log \frac{1 - \boldsymbol{P}_{ij}}{1 - \tau} = \mathcal{R}_A
\end{aligned}
\tag{7}
$$

where the KL divergence on two given distribution $\mathbb{P}(x)$ and $\mathbb{Q}(x)$ is defined as $\mathrm{KL}(\mathbb{P}(x)||\mathbb{Q}(x)) = \sum_x \mathbb{P}(x) \log \mathbb{P}(x)/\mathbb{Q}(x)$. Thus, we obtain the upper bound of $I(\boldsymbol{Z}_A; \tilde{A})$ as Eqn. 7. Similarly, the label constraint is bounded as $I(\boldsymbol{Z}_Y; \tilde{Y}) \leq \sum_{e_{ij} \in \tilde{Y}} \boldsymbol{P}_{ij} \log \frac{\boldsymbol{P}_{ij}}{\tau} + (1 - \boldsymbol{P}_{ij}) \log \frac{1-\boldsymbol{P}_{ij}}{1-\tau} = \mathcal{R}_Y$. $\square$

### C.4 PROOF FOR PROPOSITION 4.4

*Proof.* We derive the lower bound of $I(\boldsymbol{H}; \boldsymbol{Z}_Y)$ as follows.

$$
\begin{aligned}
I(\boldsymbol{H}; \boldsymbol{Z}_Y) =& I(f_{\boldsymbol{w}}(\boldsymbol{Z}_A); \boldsymbol{Z}_Y) \\
\geq& \mathbb{E}_{\boldsymbol{Z}_Y, \boldsymbol{Z}_A}[\log \frac{\mathbb{P}_{\boldsymbol{w}}(\boldsymbol{Z}_Y|\boldsymbol{Z}_A)}{\mathbb{P}(\boldsymbol{Z}_Y)}] \\
=& \mathbb{E}_{\boldsymbol{Z}_Y, \boldsymbol{Z}_A}[\log \mathbb{P}_{\boldsymbol{w}}(\boldsymbol{Z}_Y|\boldsymbol{Z}_A) - \log(\mathbb{P}(\boldsymbol{Z}_Y))] \\
=& \mathbb{E}_{\boldsymbol{Z}_Y, \boldsymbol{Z}_A}[\log \mathbb{P}_{\boldsymbol{w}}(\boldsymbol{Z}_Y|\boldsymbol{Z}_A) + H(\boldsymbol{Z}_Y)] \\
\geq& \mathbb{E}_{\boldsymbol{Z}_Y, \boldsymbol{Z}_A}[\log \mathbb{P}_{\boldsymbol{w}}(\boldsymbol{Z}_Y|\boldsymbol{Z}_A)] \\
\approx& -\frac{1}{|\boldsymbol{Z}_Y|} \sum_{e_{ij} \in \boldsymbol{Z}_Y} \mathcal{L}_{cls}(f_{\boldsymbol{w}}(\boldsymbol{Z}_A), \boldsymbol{Z}_Y)
\end{aligned}
\tag{8}
$$

Thus, the MI $I(\boldsymbol{H}; \boldsymbol{Z}_Y)$ can be lower bounded by the classification loss, and $\min -\lambda_s I(\boldsymbol{H}; \boldsymbol{Z}_Y)$ in RGIB-REP (Eqn. 4) is upper bounded by $\min \lambda_s/|\boldsymbol{Z}_Y| \sum_{e_{ij} \in \boldsymbol{Z}_Y} \mathcal{L}_{cls}(f_{\boldsymbol{w}}(\boldsymbol{Z}_A), \boldsymbol{Z}_Y)$ as Eqn. 8. $\square$

### C.5 PROOF FOR THEOREM 4.5

*Proof.* With the relaxation via parametrization in C.4, we first relax the standard RGIB-REP to its upper bound as follows.

$$
\begin{aligned}
\min \mathrm{RGIB\text{-}REP} \triangleq& -\lambda_s I(\boldsymbol{H}; \boldsymbol{Z}_Y) + \lambda_A I(\boldsymbol{Z}_A; \tilde{A}) + \lambda_Y I(\boldsymbol{Z}_Y; \tilde{Y}) \\
\leq& -\lambda_s I(\boldsymbol{Z}_A; \boldsymbol{Z}_Y) + \lambda_A I(\boldsymbol{Z}_A; \tilde{A}) + \lambda_Y I(\boldsymbol{Z}_Y; \tilde{Y})
\end{aligned}
\tag{9}
$$

As $\boldsymbol{Z}_Y^* \approx Y$, Eqn. 9 can be reduced to $\min -\lambda_s I(\boldsymbol{Z}_A; Y) + \lambda_A I(\boldsymbol{Z}_A; \tilde{A}) + \lambda_Y H(Y)$, where $I(Y; \tilde{Y}) = H(Y)$ as $Y \subseteq \tilde{Y}$. Removing the final term with constant $H(Y)$, it can be further reduced to $\min -I(\boldsymbol{Z}_A; Y) + \lambda I(\boldsymbol{Z}_A; \tilde{A})$, where $\lambda = \lambda_A/\lambda_s$. Since to minimize the $-I(\boldsymbol{Z}_A; Y) + \lambda I(\boldsymbol{Z}_A; \tilde{A})$

is equal to maximize the $I(\boldsymbol{Z}_A; Y) - \lambda I(\boldsymbol{Z}_A; \tilde{A})$, next, we conduct the following derivation:

$$
\begin{aligned}
& \max I(\boldsymbol{Z}_A; Y) - \lambda I(\boldsymbol{Z}_A; \tilde{A}) \\
={} & \max \big( I(Y; \boldsymbol{Z}_A, \tilde{A}) - I(\tilde{A}; Y | \boldsymbol{Z}_A) \big) - \lambda I(\boldsymbol{Z}_A; \tilde{A}) \\
={} & \max I(Y; \boldsymbol{Z}_A, \tilde{A}) - I(\tilde{A}; Y | \boldsymbol{Z}_A) - \lambda \big( I(\boldsymbol{Z}_A; \tilde{A}, Y) - I(\tilde{A}; Y | \boldsymbol{Z}_A) \big) \\
={} & \max I(Y; \boldsymbol{Z}_A, \tilde{A}) - (1 - \lambda) I(\tilde{A}; Y | \boldsymbol{Z}_A) - \lambda I(\boldsymbol{Z}_A; \tilde{A}, Y) \\
={} & \max I(Y; \tilde{A}) - (1 - \lambda) I(\tilde{A}; Y | \boldsymbol{Z}_A) - \lambda I(\boldsymbol{Z}_A; \tilde{A}, Y) \\
={} & \max (1 - \lambda) I(\tilde{A}; Y) - (1 - \lambda) I(\tilde{A}; Y | \boldsymbol{Z}_A) - \lambda I(\boldsymbol{Z}_A; \tilde{A} | Y) \\
={} & \max (1 - \lambda) c - (1 - \lambda) I(\tilde{A}; Y | \boldsymbol{Z}_A) - \lambda I(\boldsymbol{Z}_A; \tilde{A} | Y) \\
={} & (1 - \lambda) c
\end{aligned}
\tag{10}
$$

Since MI $I(\tilde{A}; Y | \boldsymbol{Z}_A) \geq 0$ and $I(\boldsymbol{Z}_A; \tilde{A} | Y) \geq 0$ are always true, the optimal $\boldsymbol{Z}_A^*$ should makes $-(1 - \lambda) I(\tilde{A}; Y | \boldsymbol{Z}_A) - \lambda I(\boldsymbol{Z}_A; \tilde{A} | Y) = 0$ to reach the optimal case. Thus, it should satisfy $I(\tilde{A}; Y | \boldsymbol{Z}_A) = 0$ and $I(\boldsymbol{Z}_A; \tilde{A} | Y) = 0$ simultaneously. Therefore, $\boldsymbol{Z}_A = \boldsymbol{Z}_A^*$ maximizes $I(\boldsymbol{Z}_A; Y) - \lambda I(\boldsymbol{Z}_A; \tilde{A})$, which is equal to minimize $-I(\boldsymbol{Z}_A; Y) + \lambda I(\boldsymbol{Z}_A; \tilde{A})$, i.e., the RGIB-REP. Symmetrically, when $\boldsymbol{Z}_A \equiv A$, $\boldsymbol{Z}_Y = \boldsymbol{Z}_Y^*$ maximizes $I(A; \boldsymbol{Z}_Y) - \lambda I(\boldsymbol{Z}_Y; \tilde{Y})$ as $I(A; \boldsymbol{Z}_Y) - \lambda I(\boldsymbol{Z}_Y; \tilde{Y}) = (1 - \lambda) I(\tilde{Y}; A) - (1 - \lambda) I(\tilde{Y}; A | \boldsymbol{Z}_Y) - \lambda I(\boldsymbol{Z}_Y; \tilde{Y} | A)$. □

## D  A FURTHER EMPIRICAL STUDY OF NOISE EFFECTS

This section is the extension of Section 3 in the main content.

### D.1  FULL EVALUATION RESULTS

**Evaluation settings.** In this part, we provide a thorough empirical study traversing all combinations of following settings from 5 different aspects.

- 3 GNN architectures: GCN, GAT, SAGE.
- 3 numbers of layers: $2, 4, 6$.
- 4 noise types: clean, mixed noise, input noise, label noise.
- 3 noise ratios: $20\%, 40\%, 60\%$.
- 6 datasets: Cora, CiteSeer, PubMed, Facebook, Chameleon, Squirrel.

Then, we summarize the entire evaluation results of three kinds of GNNs as follows. As can be seen, all three common GNNs, including GCN, GAT, and SAGE, are vulnerable to the inherent edge noise.

- Table 13: full evaluation results with GCN.
- Table 14: full evaluation results with GAT.
- Table 15: full evaluation results with SAGE.

**Data statistics.** The statistics of the 6 datasets in our experiments are shown in Table 8. As can be seen, 4 homogeneous graphs (Cora, CiteSeer, PubMed, and Facebook) are with the much higher homophily values than the other 2 heterogeneous graphs (Chameleon and Squirrel).

Table 8: Dataset statistics.

| dataset | Cora | CiteSeer | PubMed | Facebook | Chameleon | Squirrel |
|---|---|---|---|---|---|---|
| # Nodes | 2,708 | 3,327 | 19,717 | 4.039 | 2,277 | 5,201 |
| # Edges | 5,278 | 4,676 | 88,648 | 88,234 | 31,421 | 198,493 |
| Homophily | 0.81 | 0.74 | 0.80 | 0.98 | 0.23 | 0.22 |

### D.2 FURTHER INVESTIGATION ON THE DECOUPLED NOISE

**Loss distribution** We visualize the loss distribution under different scenarios to further investigate the memorization effect of GNNs. As shown in Figure 6, two clusters are gradually separated apart with clean data, but such a learning process can be slowed down when training with the noisy-input data, which confuses the model and leads to overlapped distributions. As for the label noise, the model cannot distinguish the noisy samples with a clear decision boundary to separate the clean and noisy samples apart. Besides, it is found that the model can gradually memorize the noisy edges according to the decreasing trend of the corresponding losses, where the loss distribution of noisy edges is minimized and progressively moving towards that of the clean ones.

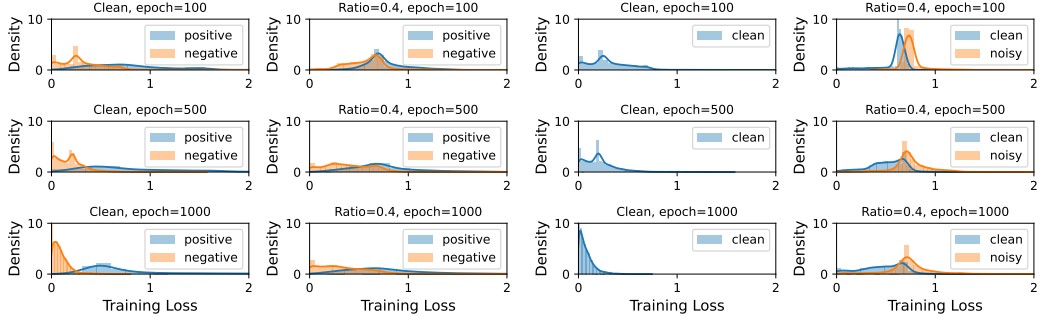

(a) Training loss of positive/nagative samples with clean data (left) and 40% input-noisy data (right).

(b) Training loss of clean/noisy samples with clean data (left) and 40% label-noisy data (right).

Figure 6: Loss distribution of the standard GCN with 40% input noise (a) and 40% label noise (b).

**Visualization of edge representations** As shown in Figure 7, the dimensionality reduction technique, T-SNE, is utilized here to measure the similarity of the learned edge representations $\boldsymbol{h}_{ij} = [\boldsymbol{h}_i, \boldsymbol{h}_j]$ with edges $e_{ij}$ in the test set $\mathcal{E}^{test}$. Compared with the representation distribution of clean data (Figure 7(a)), which present comparatively distinct cluster boundaries, both the input noise (Figure 7(b)) and the label noise (Figure 7(c)) are with overlapped clusters of the positive and negative samples. Namely, the GCN model is confused by the two patterns of noise, and fails to give a discriminate the true or false edges with their representations.

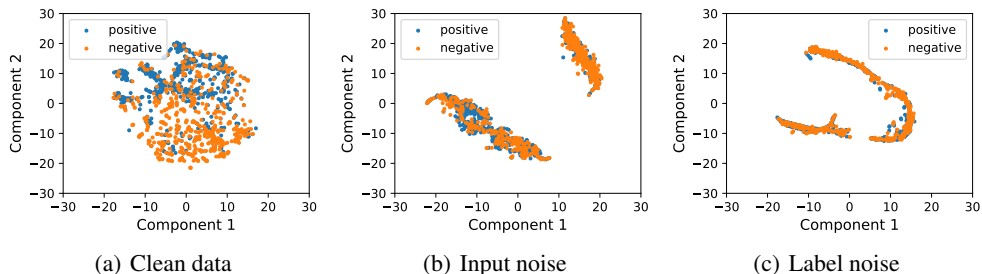

(a) Clean data                (b) Input noise                (c) Label noise

Figure 7: Visualization of the representations of predictive edges through T-SNE. The edge representations are learned on the Cora dataset by a 4-layer GCN with the standard supervised learning.

### D.3 DECOUPLING INPUT NOISE AND LABEL NOISE

For a further and deeper study of the coupled noise, we use the *edge homophily* metric to quantify the distribution of edges. Specifically, the homophily value $h_{ij}^{homo}$ of the edge $e_{ij}$ is computed as the cosine similarity of the node feature $x_i$ and $x_j$, i.e., $h_{ij}^{homo} = cos(x_i, x_j)$. As shown in figure 8, the distributions of edge homophily are nearly the same for label noise and structure noise, where the

envelope of the two distributions are almost overlapping. Here, we justify that the randomly split label noise and structure noise are indeed coupled together.

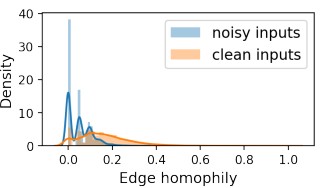 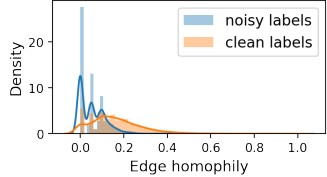 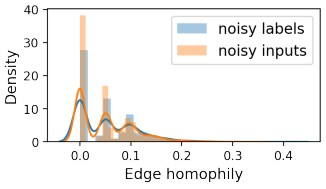

(a) Homophily distributions of the input edges

(b) Homophily distributions of the output edges

(c) Homophily distributions of the noisy edges

Figure 8: Distributions of edge homophily with random split manner.

Based on the measurement of edge homophily, we then decouple these two kinds of noise in simulation. Here, we consider two cases that separate these two kinds of noise apart, i.e., (case 1) input noise with high homophily and label noise with low homophily shown in Figure 9, and (case 2) input noise with low homophily and label noise with high homophily shown in Figure 10.

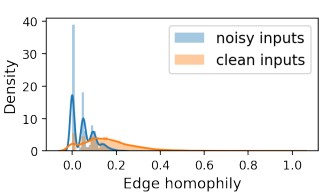 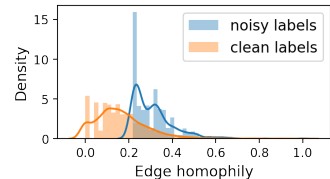 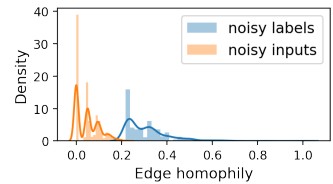

(a) Homophily distributions of the input edges

(b) Homophily distributions of the output edges

(c) Homophily distributions of the noisy edges

Figure 9: Distributions of edge homophily for decoupled noise (case 1).

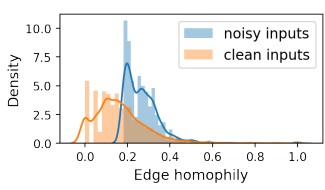 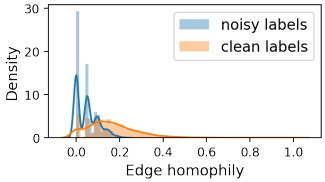 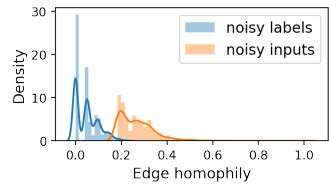

(a) Homophily distributions of the input edges

(b) Homophily distributions of the output edges

(c) Homophily distributions of the noisy edges

Figure 10: Distributions of edge homophily for decoupled noise (case 2).

### D.4 A FURTHER DISCUSSION ON THE INHERENT EDGE NOISE

Basically, the edge noise can exist in the form of *false positive* edges or *false negative* noise. Specifically, the false positive edges are treated as existing edges with label 1, but in fact, such edges do not exist. On the other hand, the false negative edges are treated as non-existing edges with label 0 that the predictive probabilities of such edges will be minimized.

Here, we would highlight that our work focuses on the *false positive* edges as it is more practical and common in real-world scenarios since the data annotating procedure can produce such a kind of noise (Wu et al., 2022b). Thus, if the inherent edge noise exists, it is more likely to be false positive samples, while the false negative samples are often intractable to be collected and annotated

in practice.

Actually, investigating the influences of the false negative samples is another line of research, such as (Yang et al., 2020; Kamigaito & Hayashi, 2022), which is orthogonal and complementary to our work. Learning from a clean graph can also encounter the problem of false negative samples, which is usually due to the random sampling of negative nodes/edges. Note that tackling the false negative samples is a well-known problem in the area of link prediction while handling the false positive samples is also valuable but still remains under-explored.

The input noise and label noise does look quite similar and coupled. From the perspective of data processing, the collected noisy edges in the training set can be randomly split into the observed graph (i.e., the input of GNN) or the predictive graph (i.e., the query edges for the GNN to predict). Considering the noisy edges might come from similar sources, e.g., biases from human annotation, the corresponding noise patterns can also be similar between these two kinds of noise that are naturally coupled together to some extent. However, we would claim and justify that the two kinds of noise are fundamentally different.

Although both noise can bring severe degradation on empirical performances, they actually act in different places from the model perspective. As the learnable weights are updated as $\mathbf{w} := \mathbf{w} - \eta \nabla_{\mathbf{w}} \mathcal{L}(\mathbf{H}, \tilde{\mathbf{Y}})$, the label noise $\tilde{Y}$ acting on the backward propagation can directly influence the model. By contrast, the input noise indirectly acts on the learned weights as it appears in the front end of the forward inference, i.e., $\mathbf{H} = \mathbf{f_w}(\mathbf{A}, \mathbf{X})$.

Empirically, as results shown in Table 3 and Table 4, the standard GNN (without any defenses) performs quite differently under the same proportion of input noise and label noise. Such a phenomenon can inspire one to understand the intrinsic denoising mechanism or memorization effects of the GNN, and we would leave that as further work.

More importantly, from the perspective of defending, it could be easy and trivial to defend if these two kinds of noise are the same. However, none of the existing robust methods can effectively defend such an inherently coupled noise. As can be seen from Table 2, only marginal improvements are achieved when applying the existing robust methods to the coupled noise. While in Table 3 and Table 4, these robust methods work effectively in handling the decoupled noise, i.e., only the structure noise or label noise exists. The reason is that, the properties of coupled noise are much more complex than the single decoupled noise. Both sides of the information sources, i.e., $\tilde{A}$ and $\tilde{Y}$, should be considered noisy, based on which the defending mechanism could be devised.

## E    IMPLEMENTATION DETAILS

### E.1    GNNS FOR LINK PREDICTION

We provide a detailed introduction forward propagation and backward update of GNNs in this part.

Formally, let $\ell = 1 \ldots L$ denote the layer index, $h_i^\ell$ is the representation of the node $i$, MESS$(\cdot)$ is a learnable mapping function to transformer the input feature, AGGREGATE$(\cdot)$ captures the 1-hop information from neighborhood $\mathcal{N}(v)$ in the graph, and COMBINE$(\cdot)$ is final combination between neighbour features and the node itself. Then, the $l$-layer operation of GNNs can be formulated as $\boldsymbol{m}_v^\ell = \text{AGGREGATE}^\ell(\{\text{MESS}(\boldsymbol{h}_u^{\ell-1}, \boldsymbol{h}_v^{\ell-1}, e_{uv}) : u \in \mathcal{N}(v)\})$, where the representation of node $v$ is $\boldsymbol{h}_v^\ell = \text{COMBINE}^\ell(\boldsymbol{h}_v^{\ell-1}, \boldsymbol{m}_v^\ell)$.

After $L$-layer propagation, the final node representations $\boldsymbol{h}_e^L$ of each $e \in V$ are obtained. Then, for each query edge $e_{ij} \in \mathcal{E}^{train}$ unseen from the input graph, the logit $\boldsymbol{\phi}_{e_{ij}}$ is computed with the node representations $\boldsymbol{h}_i^L$ and $\boldsymbol{h}_j^{L\ 4}$ with the readout function, i.e., $\boldsymbol{\phi}_{e_{ij}} = \text{READOUT}(\boldsymbol{h}_i^L, \boldsymbol{h}_j^L) \to \mathbb{R}$.

Finally, the optimization objective can be defined as minimizing the binary cross-entropy loss, i.e., $\min \mathcal{L}_{cls} = \sum_{e_{ij} \in \mathcal{E}^{train}} -y_{ij} \log(\sigma(\boldsymbol{\phi}_{e_{ij}})) - (1 - y_{ij}) \log(1 - \sigma(\boldsymbol{\phi}_{e_{ij}}))$ where $\sigma(\cdot)$ is the sigmoid function, and $y_{ij} = 1$ for positive edges while $y_{ij} = 0$ for negative ones.

In addtion, we summarize the detailed architectures of different GNNs in the following Table 9.

---

[4] To avoid abusing notations, we use the $\boldsymbol{h}_i$ to stand for the final representation $\boldsymbol{h}_i^L$ in later contents.

Table 9: Detailed architectures of different GNNs.

| GNN | MESS($\cdot$) & AGGREGATE($\cdot$) | COMBINE($\cdot$) | READOUT($\cdot$) |
|-----|-----|-----|-----|
| GCN | $\boldsymbol{m}_i^l = \boldsymbol{W}^l \sum_{j \in \mathcal{N}(i)} \frac{1}{\sqrt{\hat{d}_i \hat{d}_j}} \boldsymbol{h}_j^{l-1}$ | $\boldsymbol{h}_i^l = \sigma(\boldsymbol{m}_i^l + \boldsymbol{W}^l \frac{1}{\hat{d}_i} \boldsymbol{h}_i^{l-1})$ | $\phi_{e_{ij}} = \boldsymbol{h}_i^\top \boldsymbol{h}_j$ |
| GAT | $\boldsymbol{m}_i^l = \sum_{j \in \mathcal{N}(i)} \alpha_{ij} \boldsymbol{W}^l \boldsymbol{h}_j^{l-1}$ | $\boldsymbol{h}_i^l = \sigma(\boldsymbol{m}_i^l + \boldsymbol{W}^l \alpha_{ii} \boldsymbol{h}_i^{l-1})$ | $\phi_{e_{ij}} = \boldsymbol{h}_i^\top \boldsymbol{h}_j$ |
| SAGE | $\boldsymbol{m}_i^l = \boldsymbol{W}^l \frac{1}{|\mathcal{N}(i)|} \sum_{j \in \mathcal{N}(i)} \boldsymbol{h}_j^{l-1}$ | $\boldsymbol{h}_i^l = \sigma(\boldsymbol{m}_i^l + \boldsymbol{W}^l \boldsymbol{h}_i^{l-1})$ | $\phi_{e_{ij}} = \boldsymbol{h}_i^\top \boldsymbol{h}_j$ |

### E.2 IMPLEMENTATION DETAILS OF RGIB-SSL

As introduced in Section 4.2, the graph augmentation technique is adopted here to generate the perpetuated graphs of various views.

To avoid manually selecting and tuning the augmentation operations, we propose the hybrid graph augmentation method with the four augmentation operations as predefined candidates and the ranges of their corresponding hyper-parameters. The search space is summarized in Table 10, where the candidate operations cover most augmentation approaches except for those operations modifying the number of nodes that are unsuitable for the link prediction task.

In each training iteration, two augmentation operators $\mathcal{T}^1(\cdot)$ and $\mathcal{T}^2(\cdot)$ and their hyper-parameters $\theta_1$ and $\theta_2$ are randomly sampled from the search space as elaborated in Algorithm. 1. The two operators will be performed on the observed graph $\mathcal{G}$, obtaining the two augmented graphs, namely, $\mathcal{G}^1 = \mathcal{T}^1(\mathcal{G}|\theta_1)$ and $\mathcal{G}^2 = \mathcal{T}^2(\mathcal{G}|\theta_2)$. The edge representations are gained by $\boldsymbol{h}_{ij}^1 = f(\mathcal{G}^1, e_{ij}|\boldsymbol{w}) = \boldsymbol{h}_i^1 \odot \boldsymbol{h}_j^1$, where the node representations $\boldsymbol{h}_i^1$ and $\boldsymbol{h}_i^2$ are generated by the GNN model $f(\cdot|\boldsymbol{w})$ with learnable weights $\boldsymbol{w}$, and so it is for $\boldsymbol{h}_{ij}^2$ with $\mathcal{G}^2$.

Table 10: Search space $\mathcal{H}_T$ of the hybrid graph augmentation.

| Operator $\mathcal{T}_i(\cdot)$ | hyper-parameter $\theta_{T_i}$ | description |
|-----|-----|-----|
| edge removing | $\theta_{er} \in (0.0, 0.5)$ | Randomly remove the observed edges with prob. $\theta_{er}$. |
| feature masking | $\theta_{fm} \in (0.0, 0.3)$ | Masking the columns of node features with prob. $\theta_{fm}$. |
| feature dropping | $\theta_{fd} \in (0.0, 0.3)$ | Dropping the elements of node features with prob. $\theta_{fd}$. |
| identity | NA. | Do not modify anything. |

---

**Algorithm 1** Hybrid graph augmentation.

---

**Require:** number of augmentation operators $n$;
1: initialize $\emptyset \to \mathcal{T}(\cdot)$
2: **for** $i = 1 \ldots n$ **do**
3:     randomly select one augmentation operator from operator space $\mathcal{T}_i(\cdot) \in \mathcal{H}_T$;
4:     uniformly sample the corresponding hyper-parameter $\theta_i \sim U(\theta_{T_i})$;
5:     store the newly sampled operator and combine it with the existing ones $\mathcal{T}(\cdot) \cup \{\mathcal{T}_i(\cdot|\theta_i)\} \to \mathcal{T}(\cdot)$;
6: **end for**
7: **return** the hybrid augmentation operators $\mathcal{T}(\cdot)$

---

Then, we learn the self-supervised edge representations by maximizing the edge-level agreement between the same query edge of different augmented graphs (positive pairs) and minimizing the agreement among different edges (negative pairs) with their representations as shown in Figure 11. Note the $\boldsymbol{h}_{ij}$ here is the edge representation. Specifically, we minimize the representation similarity of the positive pairs $(\boldsymbol{h}_{ij}^1, \boldsymbol{h}_{ij}^2)$ and maximize the representation similarity of the randomly-sampled negative pairs $(\boldsymbol{h}_{ij}^1, \boldsymbol{h}_{mn}^2)$, where $e_{ij} \neq e_{mn}$.

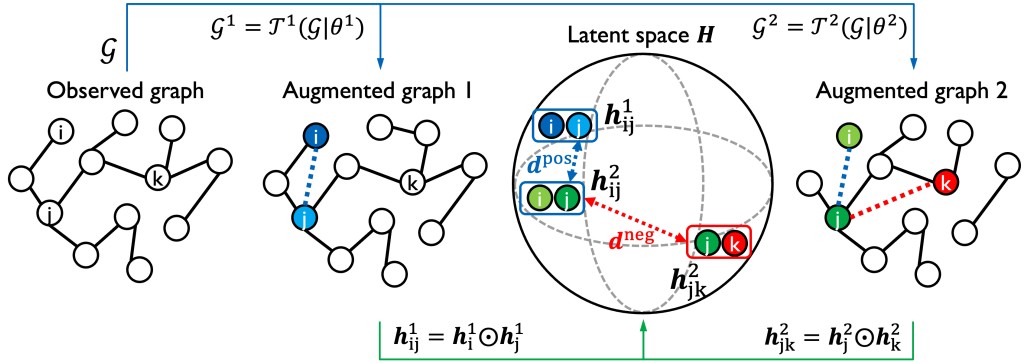

Figure 11: Illustration of the RGIB-SSL model.

### E.3 BASELINE IMPLEMENTATIONS

All baselines compared in this paper are based on their own original implementations. We list their source links here.

- DropEdge, `https://github.com/DropEdge/DropEdge`.
- NeuralSparse, `https://github.com/flyingdoog/PTDNet`.
- PTDNet, `https://github.com/flyingdoog/PTDNet`.
- GCN Jaccard, `https://github.com/DSE-MSU/DeepRobust`.
- GIB, `https://github.com/snap-stanford/GIB`.
- VIB, `https://github.com/RingBDStack/VIB-GSL`.
- PRI, `https://github.com/SJYuCNEL/PRI-Graphs`.
- Co-teaching, `https://github.com/bhanML/Co-teaching`.
- Peer loss functions,
  `https://github.com/weijiaheng/Multi-class-Peer-Loss-functions`.
- SupCon, `https://github.com/HobbitLong/SupContrast`.
- GRACE, `https://github.com/CRIPAC-DIG/GRACE`.

## F  FULL EMPIRICAL RESULTS

In this section, we elaborate the full empirical study on inherent edge noise with various robust methods and GNN architectures.

### F.1  ROBUST METHODS COMPARISON WITH CLEAN DATA

Here, we would like to figure out how the robust methods introduced in Section 5 behave when learning with clean data, i.e., no edge noise exists. As shown in Table 11, the proposed two instantiations of RGIB can also boost the predicting performance when learning on clean graphs, and outperforms other baselines in most cases.

### F.2  FURTHER ABLATION STUDIES ON THE TRADE-OFF PARAMETERS $\lambda$

We conduct an ablation study with the grid search of several hyper-parameters $\lambda$ in RGIB. For simplicity, we fix the weight of the supervision signal as one, i.e., $\lambda_s = 1$. Then, the objective of RGIB can be formed as $\mathcal{L} = \mathcal{L}_{cls} + \lambda_1 \mathcal{R}_1 + \lambda_2 \mathcal{R}_2$, where the information regularization terms $\mathcal{R}_1/\mathcal{R}_2$ are alignment and uniformity for RGIB-SSL, while topology constraint and label constraint for RGIB-REP, respectively. As the heatmaps illustrated in Figure 12 and Figure 13, the $\lambda_1, \lambda_2$ are

Table 11: Method comparison with a 4-layer GCN trained on the clean data. The **boldface** numbers mean the best results, while the underlines indicate the second-best results.

| method | Cora | Citeseer | Pubmed | Facebook | Chameleon | Squirrel |
|---|---|---|---|---|---|---|
| Standard | .8686 | .8317 | .9178 | .9870 | .9788 | **.9725** |
| DropEdge | .8684 | .8344 | .9344 | .9869 | .9700 | .9629 |
| NeuralSparse | .8715 | .8405 | .9366 | .9865 | **.9803** | .9635 |
| PTDNet | .8577 | .8398 | .9315 | .9868 | .9696 | .9640 |
| Co-teaching | .8684 | .8387 | .9192 | .9771 | .9698 | .9626 |
| Peer loss | .8313 | .7742 | .9085 | .8951 | .9374 | .9422 |
| Jaccard | .8413 | .8005 | .8831 | .9792 | .9703 | .9610 |
| GIB | .8582 | .8327 | .9019 | .9691 | .9628 | .9635 |
| SupCon | .8529 | .8003 | .9131 | .9692 | .9717 | .9619 |
| GRACE | .8329 | .8236 | .9358 | .8953 | .8999 | .9165 |
| **RGIB-REP** | .8758 | .8415 | .9408 | **.9875** | .9792 | .9680 |
| **RGIB-SSL** | **.9260** | **.9148** | **.9593** | .9845 | .9740 | .9646 |

better in certain ranges. Neither too large nor too small value is not guaranteed to find a good solution.

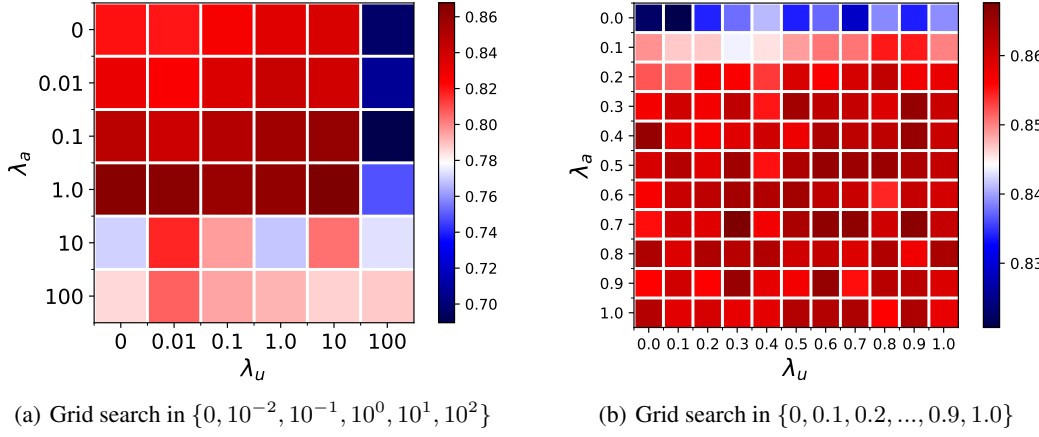

(a) Grid search in $\{0, 10^{-2}, 10^{-1}, 10^0, 10^1, 10^2\}$
(b) Grid search in $\{0, 0.1, 0.2, ..., 0.9, 1.0\}$

Figure 12: Grid search of hyper-parameters with RGIB-SSL on Cora dataset ($\epsilon = 40\%$).

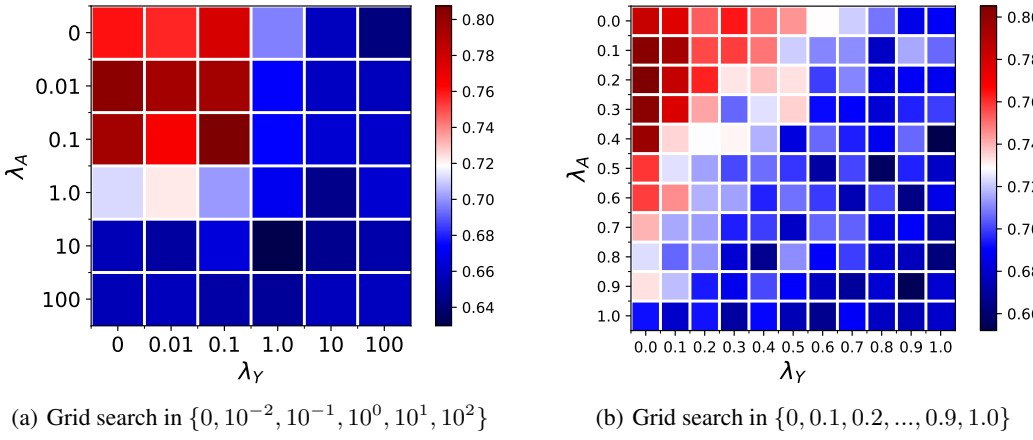

(a) Grid search in $\{0, 10^{-2}, 10^{-1}, 10^0, 10^1, 10^2\}$
(b) Grid search in $\{0, 0.1, 0.2, ..., 0.9, 1.0\}$

Figure 13: Grid search of hyper-parameters with RGIB-REP on Cora dataset ($\epsilon = 40\%$).

## F.3 FURTHER CASE STUDIES

**Alignment and uniformity of baseline methods.** The alignment of other methods is summarized in Table 12, while the uniformity is visualized in Figure 14 and Figure 15. We have the following three observations.

First, the sampling-based methods, e.g., DropEdge, PTDNet, and NeuralSparse, can also promote alignment and uniformity due to their sampling mechanisms to defend the structural perturbations. Second, the contrastive methods, e.g., SupCon and GRACE, are with much better alignment but much worse uniformity. The reason is that the learned representations are severely collapsed, which can be degenerated to single points seen from the uniformity plots but stay nearly unchanged when encountering structural perturbations. Third, the remaining methods are not observed with significant improvements in alignment or uniformity.

When connecting the above observations with their empirical performances, we can draw a conclusion. That is, both alignment and uniformity are important to evaluate the robust methods from the perspective of representation learning. Besides, such a conclusion is in line with the previous study (Wang & Isola, 2020).

Table 12: Alignment comparison.

| method | Cora | | | | Citeseer | | | |
|---|---|---|---|---|---|---|---|---|
| | clean | $\varepsilon = 20\%$ | $\varepsilon = 40\%$ | $\varepsilon = 60\%$ | clean | $\varepsilon = 20\%$ | $\varepsilon = 40\%$ | $\varepsilon = 60\%$ |
| Standard | .616 | .687 | .695 | .732 | .445 | .586 | .689 | .696 |
| DropEdge | .606 | .670 | .712 | .740 | .463 | .557 | .603 | .637 |
| NeuralSparse | .561 | .620 | .691 | .692 | .455 | .469 | .583 | .594 |
| PTDNet | .335 | .427 | .490 | .522 | .330 | .397 | .459 | .499 |
| Co-teaching | .570 | .691 | .670 | .693 | .464 | .581 | .683 | .706 |
| Peer loss | .424 | .690 | .722 | .629 | .529 | .576 | .580 | .598 |
| Jaccard | .608 | .627 | .658 | .703 | .471 | .596 | .674 | .703 |
| GIB | .592 | .635 | .652 | .692 | .439 | .524 | .591 | .623 |
| SupCon | .060 | .045 | .033 | .024 | .086 | .077 | .060 | .065 |
| GRACE | .466 | .548 | .582 | .556 | .454 | .551 | .589 | .609 |
| **RGIB-REP** | .524 | .642 | .679 | .704 | .439 | .533 | .623 | .647 |
| **RGIB-SSL** | .475 | .543 | .578 | .615 | .418 | .505 | .533 | .542 |

**Learning curves of RGIB.** We draw the learning curves of RGIB with constant schedulers in Figure 16, Figure 17, Figure 18, and Figure 19. We normalize the values of each plotted line to $(0, 1)$ for better visualization.

For RGIB-SSL, the uniformity term, i.e, $H(\mathbf{H})$, converges quickly and remains low after 200 epochs. Similarly, the alignment term, i.e, $I(\mathbf{H_1}; \mathbf{H_2})$ also converges in the early stages and keeps stable in the rest. At the same time, the supervised signal, i.e, $I(\mathbf{H}; \mathbf{Y})$ gradually and steadily decreases as the training time moves forward. The learning processes are generally stable across different datasets.

As for RGIB-REP, we observe that the topology $I(\mathbf{Z_A}; \tilde{\mathbf{A}})$ and label constraints $I(\mathbf{Z_Y}; \tilde{\mathbf{Y}})$ can indeed adapt to noisy scenarios with different noise ratios. As can be seen, these two regularizations converge more significantly when learning on a more noisy case. That is, when noise ratio $\epsilon$ increases from 0 to 60%, these two regularizations react adaptively to the noisy data $\tilde{A}, \tilde{Y}$. Such a phenomenon shows that RGIB-REP with these two information constraints works as an effective information bottleneck to filter out the noisy signals.

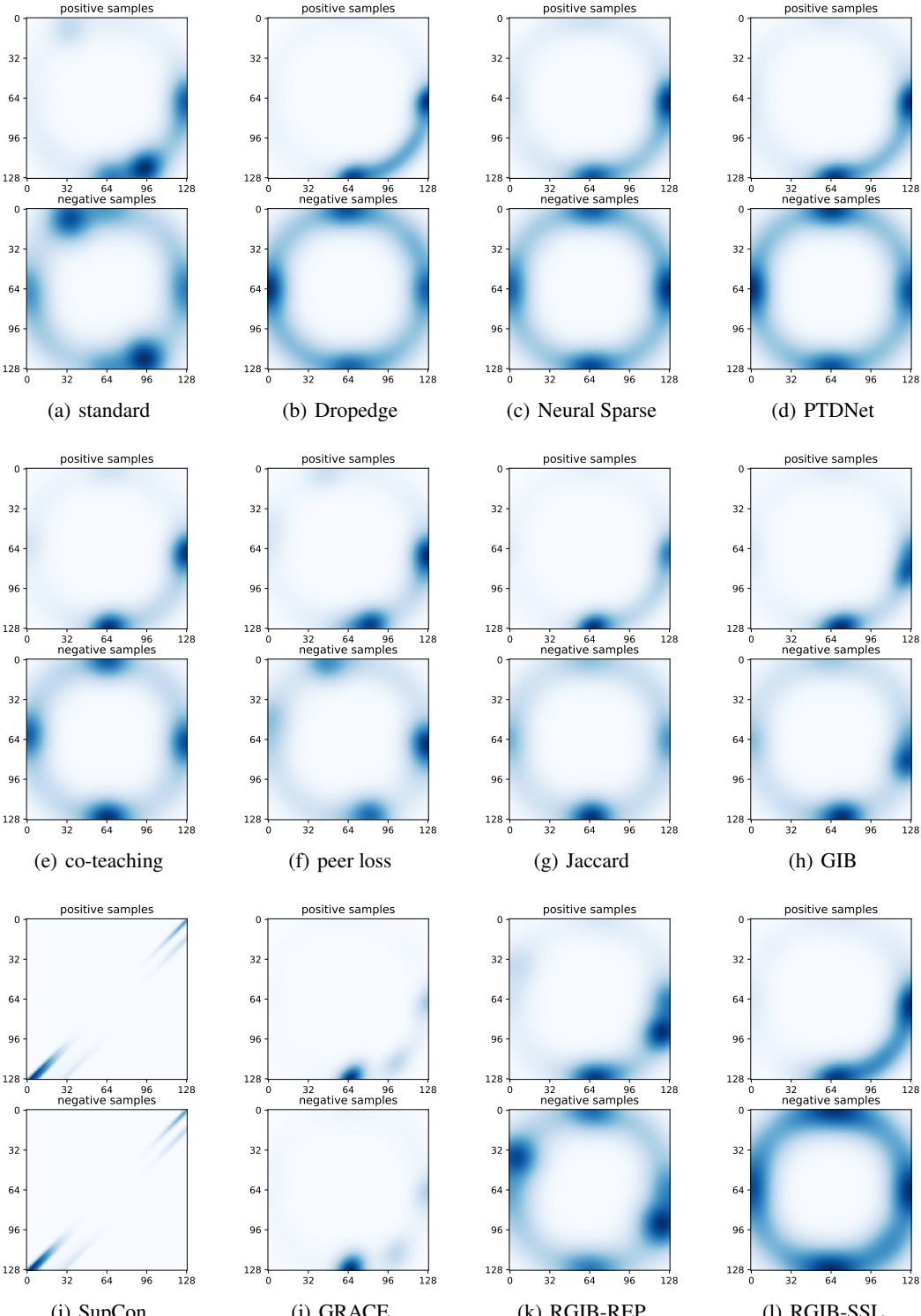

Figure 14: Uniformity on Cora with $\epsilon = 40\%$ inherent noise.

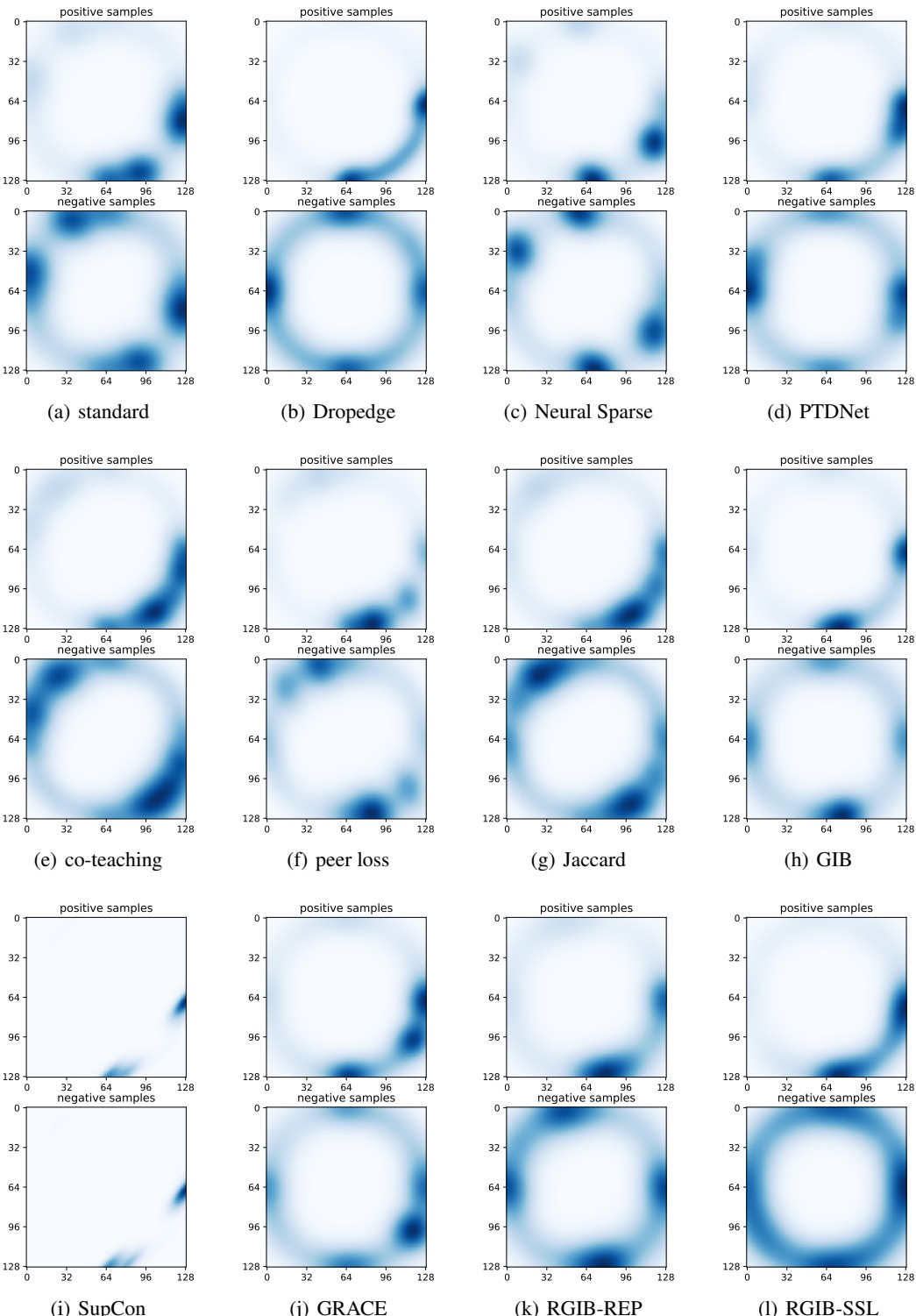

Figure 15: Uniformity on Citeseer with $\epsilon = 40\%$ inherent noise.

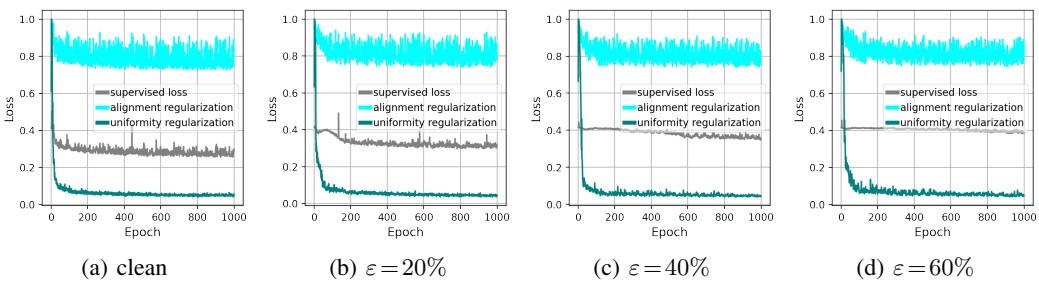

Figure 16: Learning curves of RGIB-SSL on Cora dataset.

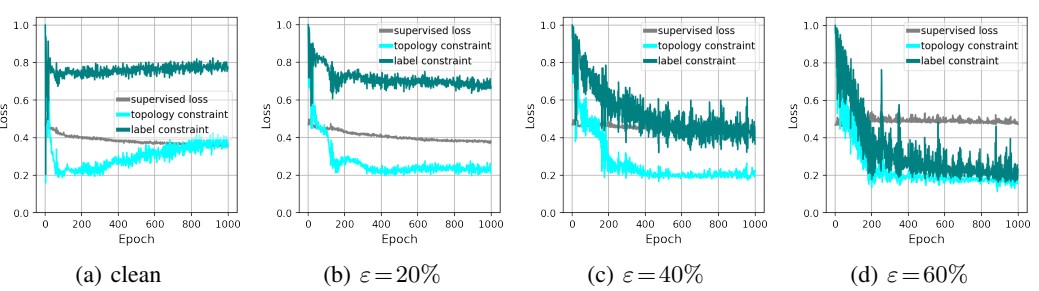

Figure 17: Learning curves of RGIB-REP on Cora dataset.

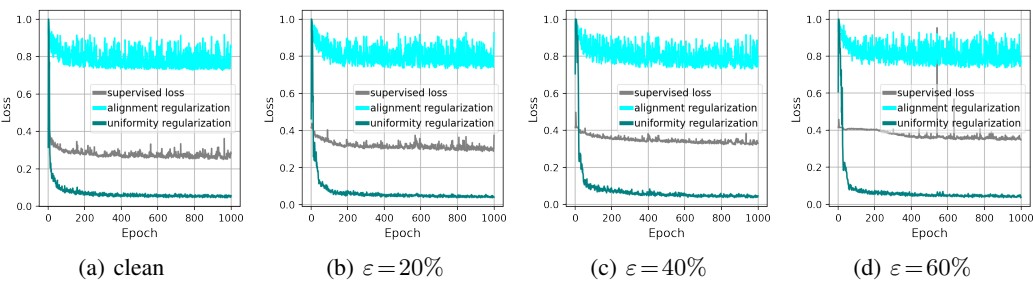

Figure 18: Learning curves of RGIB-SSL on Citeseer dataset.

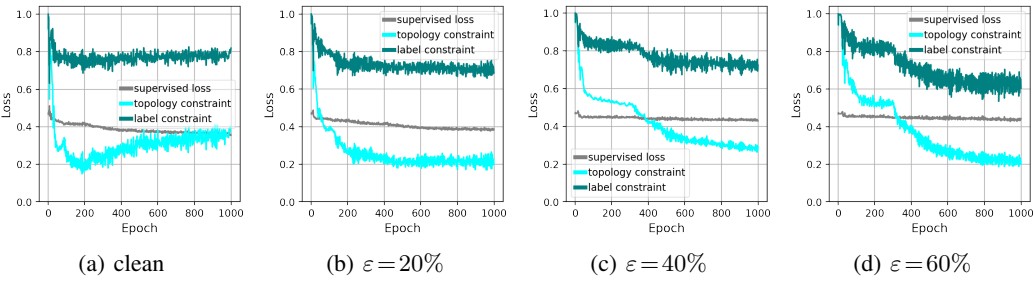

Figure 19: Learning curves of RGIB-REP on Citeseer dataset.

## F.4 Full baseline comparison with decoupled noise

The entire evaluation with 10 baselines and two proposed methods are conducted keeping the same settings as in D.1.

Results on each dataset are summarized as follows.

- Table 16, Table 17, Table 18 : full results of GCN/GAT/SAGE on Cora dataset.
- Table 19, Table 20, Table 21 : full results of GCN/GAT/SAGE on CiteSeer dataset.
- Table 22, Table 23, Table 24 : full results of GCN/GAT/SAGE on PubMed dataset.
- Table 25, Table 26, Table 27 : full results of GCN/GAT/SAGE on Facebook dataset.
- Table 28, Table 29, Table 30 : full results of GCN/GAT/SAGE on Chameleon dataset.
- Table 31, Table 32, Table 33 : full results of GCN/GAT/SAGE on Squirrel dataset.

Table 13: Full results of GCN with edge noise.

| layers | clean | mixed noise | | | input noise | | | label noise | | |
|---|---|---|---|---|---|---|---|---|---|---|
| | | 20% | 40% | 60% | 20% | 40% | 60% | 20% | 40% | 60% |
| | | | | | Cora | | | | | |
| 2 | .9183±.0071 | .8870±.0108 | .8430±.0145 | .7959±.0190 | .9091±.0085 | .8943±.0085 | .8905±.0092 | .9020±.0041 | .8793±.0099 | .8723±.0103 |
| 4 | .8686±.0102 | .8111±.0213 | .7419±.0325 | .6970±.0377 | .8027±.0580 | .7856±.0526 | .7490±.0498 | .8281±.0133 | .8054±.0213 | .8060±.0172 |
| 6 | .8256±.0494 | .7940±.0133 | .7429±.0226 | .7177±.0207 | .8035±.0370 | .7973±.0166 | .7546±.0270 | .8249±.0112 | .7925±.0137 | .7901±.0145 |
| | | | | | Citeseer | | | | | |
| 2 | .8968±.0092 | .8651±.0084 | .8355±.0154 | .8254±.0066 | .8767±.0097 | .8615±.0090 | .8585±.0085 | .8834±.0155 | .8649±.0104 | .8674±.0122 |
| 4 | .8317±.0217 | .7864±.0145 | .7380±.0201 | .7085±.0146 | .8054±.0240 | .7708±.0325 | .7583±.0258 | .7965±.0124 | .7850±.0174 | .7659±.0240 |
| 6 | .8161±.0163 | .7355±.0387 | .7110±.0222 | .7106±.0168 | .7720±.0161 | .7460±.0401 | .7212±.0470 | .7900±.0131 | .7741±.0157 | .7648±.0229 |
| | | | | | Pubmed | | | | | |
| 2 | .9737±.0011 | .9473±.0019 | .9271±.0027 | .9141±.0041 | .9590±.0022 | .9468±.0022 | .9337±.0016 | .9646±.0024 | .9637±.0021 | .9597±.0022 |
| 4 | .9178±.0084 | .8870±.0041 | .8748±.0031 | .8641±.0041 | .8854±.0051 | .8759±.0031 | .8651±.0040 | .9030±.0082 | .9039±.0029 | .9070±.0062 |
| 6 | .9081±.0056 | .8870±.0056 | .8731±.0032 | .8640±.0036 | .8855±.0025 | .8742±.0029 | .8652±.0041 | .9050±.0083 | .9112±.0059 | .9063±.0036 |
| | | | | | Facebook | | | | | |
| 2 | .9887±.0008 | .9880±.0007 | .9866±.0007 | .9843±.0010 | .9878±.0008 | .9852±.0006 | .9834±.0011 | .9892±.0006 | .9888±.0008 | .9888±.0007 |
| 4 | .9870±.0009 | .9829±.0020 | .9520±.0424 | .9438±.0402 | .9819±.0015 | .9668±.0147 | .9622±.0154 | .9882±.0007 | .9880±.0007 | .9886±.0006 |
| 6 | .9849±.0009 | .9798±.0013 | .9609±.0138 | .9368±.0179 | .9764±.0034 | .9502±.0096 | .9469±.0160 | .9863±.0013 | .9865±.0012 | .9876±.0010 |
| | | | | | Chameleon | | | | | |
| 2 | .9823±.0027 | .9753±.0025 | .9696±.0022 | .9657±.0029 | .9784±.0017 | .9762±.0016 | .9754±.0023 | .9775±.0018 | .9769±.0018 | .9755±.0036 |
| 4 | .9797±.0021 | .9616±.0033 | .9496±.0190 | .9274±.0276 | .9608±.0038 | .9433±.0261 | .9368±.0271 | .9686±.0020 | .9580±.0021 | .9362±.0035 |
| 6 | .9752±.0036 | .9662±.0042 | .9511±.0079 | .9286±.0067 | .9656±.0045 | .9450±.0177 | .9276±.0229 | .9752±.0027 | .9766±.0035 | .9745±.0040 |
| | | | | | Squirrel | | | | | |
| 2 | .9761±.0011 | .9621±.0018 | .9519±.0020 | .9444±.0024 | .9610±.0028 | .9490±.0031 | .9401±.0036 | .9744±.0013 | .9731±.0010 | .9722±.0011 |
| 4 | .9725±.0011 | .9432±.0036 | .9406±.0031 | .9386±.0025 | .9416±.0042 | .9395±.0011 | .9411±.0040 | .9720±.0016 | .9720±.0016 | .9710±.0021 |
| 6 | .9694±.0028 | .9484±.0049 | .9429±.0038 | .9408±.0039 | .9489±.0057 | .9408±.0021 | .9386±.0022 | .9688±.0028 | .9675±.0027 | .9656±.0034 |

Table 14: Full results of GAT with edge noise.

| layers | clean | mixed noise | | | input noise | | | label noise | | |
|---|---|---|---|---|---|---|---|---|---|---|
| | | 20% | 40% | 60% | 20% | 40% | 60% | 20% | 40% | 60% |
| Cora | | | | | | | | | | |
| 2 | .9076±.0070 | .8786±.0155 | .8489±.0131 | .8286±.0120 | .9014±.0070 | .8825±.0097 | .8880±.0077 | .8991±.0088 | .8841±.0097 | .8830±.0101 |
| 4 | .8783±.0103 | .8323±.0181 | .8005±.0212 | .7841±.0095 | .8616±.0107 | .8454±.0118 | .8377±.0146 | .8530±.0121 | .8357±.0082 | .8281±.0114 |
| 6 | .8650±.0157 | .8067±.0302 | .7514±.0389 | .7032±.0452 | .8414±.0127 | .7891±.0537 | .7678±.0641 | .8376±.0097 | .8154±.0069 | .8024±.0169 |
| Citeseer | | | | | | | | | | |
| 2 | .8911±.0090 | .8586±.0125 | .8338±.0127 | .8207±.0121 | .8689±.0096 | .8526±.0130 | .8512±.0174 | .8762±.0076 | .8650±.0102 | .8648±.0166 |
| 4 | .8386±.0138 | .8026±.0157 | .7775±.0248 | .7518±.0183 | .8191±.0092 | .8043±.0105 | .7912±.0073 | .8174±.0172 | .7998±.0143 | .7934±.0156 |
| 6 | .8299±.0098 | .7807±.0117 | .7373±.0270 | .7139±.0251 | .7970±.0134 | .7860±.0107 | .7741±.0126 | .7963±.0129 | .7883±.0162 | .7801±.0161 |
| Pubmed | | | | | | | | | | |
| 2 | .9406±.0032 | .9173±.0028 | .8984±.0030 | .8884±.0033 | .9255±.0024 | .9176±.0035 | .9102±.0030 | .9306±.0038 | .9271±.0030 | .9232±.0034 |
| 4 | .8960±.0068 | .8610±.0045 | .8434±.0042 | .8339±.0048 | .8668±.0040 | .8547±.0050 | .8476±.0037 | .8817±.0042 | .8772±.0040 | .8696±.0062 |
| 6 | .8631±.0072 | .8315±.0059 | .8116±.0073 | .8040±.0092 | .8374±.0036 | .8201±.0321 | .8067±.0306 | .8480±.0086 | .8414±.0116 | .8313±.0071 |
| Facebook | | | | | | | | | | |
| 2 | .9874±.0008 | .9869±.0005 | .9856±.0008 | .9836±.0007 | .9878±.0006 | .9861±.0006 | .9858±.0008 | .9872±.0006 | .9864±.0009 | .9857±.0008 |
| 4 | .9875±.0007 | .9857±.0015 | .9850±.0012 | .9820±.0014 | .9855±.0011 | .9827±.0019 | .9773±.0046 | .9873±.0010 | .9874±.0010 | .9874±.0005 |
| 6 | .9860±.0007 | .9805±.0025 | .9658±.0321 | .9577±.0314 | .9804±.0018 | .9738±.0044 | .9710±.0036 | .9854±.0015 | .9860±.0007 | .9867±.0012 |
| Chameleon | | | | | | | | | | |
| 2 | .9770±.0044 | .9725±.0027 | .9650±.0018 | .9625±.0018 | .9767±.0026 | .9747±.0020 | .9759±.0018 | .9746±.0023 | .9743±.0017 | .9711±.0041 |
| 4 | .9734±.0047 | .9721±.0035 | .9652±.0023 | .9605±.0031 | .9741±.0028 | .9686±.0030 | .9674±.0027 | .9740±.0037 | .9738±.0027 | .9712±.0047 |
| 6 | .9742±.0052 | .9659±.0029 | .9573±.0036 | .9482±.0054 | .9644±.0033 | .9543±.0075 | .9474±.0074 | .9722±.0043 | .9688±.0055 | .9698±.0065 |
| Squirrel | | | | | | | | | | |
| 2 | .9740±.0011 | .9680±.0007 | .9635±.0017 | .9588±.0025 | .9702±.0008 | .9690±.0010 | .9659±.0014 | .9719±.0018 | .9701±.0017 | .9686±.0012 |
| 4 | .9720±.0023 | .9581±.0046 | .9436±.0063 | .9335±.0062 | .9592±.0047 | .9455±.0075 | .9415±.0061 | .9682±.0030 | .9690±.0028 | .9686±.0021 |
| 6 | .9578±.0067 | .9507±.0050 | .9309±.0164 | .9254±.0089 | .9487±.0065 | .9419±.0041 | .9255±.0073 | .9585±.0097 | .9520±.0070 | .9507±.0162 |

Table 15: Full results of SAGE with edge noise.

| layers | clean | mixed noise | | | input noise | | | label noise | | |
|---|---|---|---|---|---|---|---|---|---|---|
| | | 20% | 40% | 60% | 20% | 40% | 60% | 20% | 40% | 60% |
| Cora | | | | | | | | | | |
| 2 | .9045±.0066 | .8733±.0101 | .8520±.0137 | .8469±.0056 | .9006±.0074 | .8857±.0093 | .8917±.0063 | .8868±.0070 | .8695±.0106 | .8564±.0091 |
| 4 | .8664±.0109 | .8225±.0079 | .7833±.0093 | .7595±.0259 | .8607±.0110 | .8437±.0099 | .8387±.0183 | .8309±.0090 | .8046±.0074 | .7920±.0191 |
| 6 | .8426±.0207 | .7787±.0423 | .7420±.0251 | .7180±.0248 | .8256±.0222 | .7947±.0561 | .8005±.0421 | .8158±.0168 | .7707±.0235 | .7660±.0134 |
| Citeseer | | | | | | | | | | |
| 2 | .8648±.0098 | .8438±.0143 | .8404±.0113 | .8279±.0098 | .8644±.0109 | .8608±.0064 | .8647±.0161 | .8560±.0120 | .8533±.0112 | .8412±.0122 |
| 4 | .8329±.0093 | .7914±.0101 | .7686±.0213 | .7539±.0149 | .8171±.0173 | .8226±.0141 | .8157±.0203 | .8068±.0116 | .7891±.0136 | .7705±.0172 |
| 6 | .8390±.0187 | .7708±.0168 | .7223±.0614 | .7204±.0236 | .8086±.0145 | .7997±.0183 | .7872±.0145 | .7903±.0219 | .7690±.0201 | .7564±.0096 |
| Pubmed | | | | | | | | | | |
| 2 | .8995±.0044 | .9136±.0032 | .9094±.0035 | .9035±.0040 | .9295±.0025 | .9378±.0022 | .9410±.0023 | .8817±.0034 | .8793±.0043 | .8757±.0041 |
| 4 | .8446±.0058 | .8627±.0056 | .8663±.0061 | .8619±.0073 | .8715±.0080 | .8901±.0082 | .9033±.0057 | .8386±.0085 | .8330±.0104 | .8268±.0092 |
| 6 | .8360±.0224 | .8314±.0081 | .8105±.0279 | .8333±.0089 | .8224±.0335 | .8538±.0105 | .8566±.0199 | .8242±.0104 | .8161±.0071 | .8090±.0129 |
| Facebook | | | | | | | | | | |
| 2 | .9882±.0008 | .9858±.0008 | .9827±.0008 | .9788±.0012 | .9881±.0008 | .9862±.0007 | .9862±.0008 | .9867±.0006 | .9840±.0006 | .9825±.0014 |
| 4 | .9859±.0013 | .9824±.0015 | .9783±.0025 | .9698±.0040 | .9849±.0015 | .9815±.0024 | .9815±.0025 | .9844±.0015 | .9817±.0007 | .9809±.0009 |
| 6 | .9828±.0024 | .9751±.0055 | .9603±.0365 | .9616±.0091 | .9715±.0077 | .9672±.0057 | .9517±.0228 | .9761±.0136 | .9788±.0045 | .9749±.0151 |
| Chameleon | | | | | | | | | | |
| 2 | .9786±.0030 | .9683±.0036 | .9605±.0038 | .9536±.0028 | .9754±.0019 | .9734±.0015 | .9762±.0018 | .9700±.0019 | .9670±.0026 | .9603±.0049 |
| 4 | .9743±.0025 | .9645±.0035 | .9580±.0035 | .9505±.0010 | .9721±.0017 | .9721±.0021 | .9721±.0016 | .9678±.0026 | .9641±.0028 | .9605±.0051 |
| 6 | .9729±.0022 | .9606±.0046 | .9541±.0058 | .9470±.0035 | .9679±.0031 | .9658±.0041 | .9640±.0059 | .9673±.0026 | .9641±.0038 | .9625±.0042 |
| Squirrel | | | | | | | | | | |
| 2 | .9745±.0015 | .9680±.0015 | .9626±.0015 | .9570±.0016 | .9737±.0010 | .9736±.0010 | .9721±.0011 | .9692±.0015 | .9643±.0012 | .9606±.0017 |
| 4 | .9689±.0052 | .9584±.0107 | .9577±.0076 | .9541±.0021 | .9637±.0092 | .9630±.0079 | .9607±.0090 | .9663±.0020 | .9612±.0049 | .9612±.0020 |
| 6 | .9682±.0045 | .9555±.0065 | .9528±.0038 | .9461±.0054 | .9592±.0053 | .9600±.0036 | .9551±.0042 | .9574±.0192 | .9540±.0207 | .9583±.0023 |

Table 16: Full results on Cora dataset with GCN.

| | | GCN | | | | | | | | |
|---|---|---|---|---|---|---|---|---|---|---|
| | | mixed noise | | | input noise | | | label noise | | |
| layers | method | 20% | 40% | 60% | 20% | 40% | 60% | 20% | 40% | 60% |
| L=2 | Standard | .8870±.0108 | .8430±.0145 | .7959±.0190 | .9091±.0085 | .8943±.0085 | .8905±.0092 | .9020±.0041 | .8793±.0099 | .8723±.0103 |
| | DropEdge | .8773±.0128 | .8305±.0175 | .8091±.0104 | .9016±.0086 | .8692±.0120 | .8635±.0099 | .9016±.0059 | .8791±.0089 | .8660±.0095 |
| | NeuralSparse | .8869±.0096 | .8497±.0103 | .8215±.0158 | .9118±.0063 | .8903±.0113 | .8849±.0086 | .8995±.0069 | .8797±.0073 | .8743±.0088 |
| | PTDNet | .8853±.0127 | .8451±.0168 | .8180±.0106 | .9095±.0108 | .8926±.0106 | .8825±.0096 | .9030±.0080 | .8801±.0067 | .8702±.0147 |
| | Co-teaching | .8857±.0202 | .8419±.0198 | .8026±.0237 | .9084±.0132 | .8959±.0107 | .8901±.0101 | .9021±.0203 | .8929±.0126 | .8699±.0115 |
| | Peer loss | .8867±.0115 | .8472±.0237 | .7970±.0182 | .9134±.0147 | .8993±.0171 | .8806±.0138 | .9015±.0021 | .8874±.0163 | .8813±.0291 |
| | Jaccard | .8912±.0190 | .8461±.0168 | .7964±.0182 | .9107±.0210 | .9016±.0270 | .8996±.0238 | .9094±.0107 | .8856±.0093 | .8721±.0135 |
| | GIB | .8857±.0296 | .8464±.0326 | .8037±.0309 | .9126±.0065 | .8973±.0141 | .8937±.0150 | .9083±.0136 | .8828±.0136 | .8807±.0146 |
| | SupCon | .8827±.0125 | .8451±.0157 | .8125±.0119 | .9006±.0077 | .8877±.0073 | .8846±.0124 | .8876±.0082 | .8668±.0088 | .8721±.0092 |
| | GRACE | .8588±.0158 | .8220±.0204 | .7987±.0076 | .8671±.0121 | .8102±.0126 | .8048±.0119 | .8990±.0092 | .8887±.0109 | .8896±.0074 |
| | **RGIB-REP** | .8915±.0091 | .8516±.0120 | .8358±.0077 | .9137±.0060 | .8958±.0048 | .8911±.0102 | .9046±.0107 | .8958±.0120 | .8935±.0087 |
| | **RGIB-SSL** | .9272±.0091 | .9001±.0145 | .8892±.0107 | .9342±.0049 | .9108±.0063 | .9014±.0107 | .9473±.0054 | .9427±.0051 | .9380±.0084 |
| L=4 | Standard | .8111±.0213 | .7419±.0325 | .6970±.0377 | .8027±.0580 | .7856±.0526 | .7490±.0498 | .8281±.0133 | .8054±.0213 | .8060±.0172 |
| | DropEdge | .8017±.0187 | .7423±.0335 | .7303±.0235 | .8338±.0132 | .7826±.0377 | .7454±.0425 | .8363±.0110 | .8273±.0106 | .8148±.0141 |
| | NeuralSparse | .8190±.0170 | .7318±.0379 | .7293±.0393 | .8534±.0250 | .7794±.0285 | .7637±.0529 | .8524±.0094 | .8246±.0111 | .8211±.0106 |
| | PTDNet | .8047±.0460 | .7559±.0246 | .7388±.0216 | .8433±.0443 | .8214±.0122 | .7770±.0624 | .8460±.0128 | .8214±.0166 | .8138±.0151 |
| | Co-teaching | .8197±.0236 | .7479±.0372 | .7030±.0475 | .8045±.0609 | .7871±.0564 | .7530±.0500 | .8446±.0219 | .8209±.0481 | .8157±.0246 |
| | Peer loss | .8185±.0226 | .7468±.0388 | .7018±.0473 | .8051±.0664 | .7866±.0623 | .7517±.0492 | .8325±.0201 | .8036±.0227 | .8069±.0193 |
| | Jaccard | .8143±.0218 | .7498±.0418 | .7024±.0403 | .8200±.0772 | .7838±.0558 | .7617±.0546 | .8289±.0177 | .8064±.0229 | .8148±.0227 |
| | GIB | .8198±.0331 | .7485±.0518 | .7148±.0455 | .8002±.0607 | .8099±.0566 | .7741±.0584 | .8337±.0133 | .8137±.0279 | .8157±.0270 |
| | SupCon | .8240±.0147 | .7819±.0261 | .7490±.0230 | .8349±.0124 | .8301±.0218 | .8025±.0210 | .8491±.0120 | .8275±.0115 | .8256±.0108 |
| | GRACE | .7872±.0207 | .6940±.0248 | .6929±.0140 | .7877±.0211 | .7107±.0318 | .6975±.0124 | .8531±.0166 | .8237±.0252 | .8193±.0246 |
| | **RGIB-REP** | .8313±.0098 | .7966±.0110 | .7591±.0142 | .8624±.0071 | .8313±.0136 | .8158±.0193 | .8554±.0149 | .8318±.0151 | .8297±.0150 |
| | **RGIB-SSL** | .8930±.0072 | .8554±.0167 | .8339±.0100 | .9024±.0097 | .8577±.0152 | .8421±.0156 | .9314±.0066 | .9224±.0100 | .9241±.0049 |
| L=6 | Standard | .7940±.0133 | .7429±.0226 | .7177±.0207 | .8035±.0370 | .7973±.0166 | .7546±.0270 | .8249±.0112 | .7925±.0137 | .7901±.0145 |
| | DropEdge | .7941±.0172 | .7353±.0130 | .6909±.0208 | .7900±.0560 | .7488±.0187 | .7170±.0187 | .8288±.0148 | .8187±.0076 | .7990±.0513 |
| | NeuralSparse | .7931±.0201 | .7324±.0434 | .7018±.0330 | .8082±.0418 | .7432±.0386 | .7393±.0243 | .8415±.0167 | .8160±.0181 | .8081±.0105 |
| | PTDNet | .8037±.0168 | .7601±.0225 | .7167±.0574 | .8129±.0158 | .7873±.0199 | .7570±.0511 | .8348±.0144 | .7933±.0538 | .7899±.0466 |
| | Co-teaching | .7930±.0132 | .7387±.0244 | .7151±.0262 | .8079±.0469 | .8058±.0170 | .7623±.0310 | .8298±.0365 | .8020±.0172 | .7976±.0375 |
| | Peer loss | .8015±.0211 | .7466±.0320 | .7196±.0286 | .8068±.0361 | .8031±.0169 | .7607±.0272 | .8409±.0146 | .7999±.0289 | .7918±.0240 |
| | Jaccard | .8002±.0159 | .7492±.0246 | .7211±.0250 | .8199±.0368 | .8041±.0204 | .7690±.0364 | .8277±.0138 | .8005±.0229 | .7904±.0230 |
| | GIB | .7961±.0163 | .7584±.0313 | .7201±.0300 | .8036±.0451 | .8067±.0331 | .7639±.0402 | .8347±.0111 | .8016±.0156 | .7947±.0147 |
| | SupCon | .8092±.0242 | .7365±.0227 | .6920±.0415 | .8021±.0251 | .7845±.0280 | .7434±.0257 | .8273±.0202 | .8181±.0234 | .8157±.0135 |
| | GRACE | .7576±.0148 | .7187±.0243 | .6860±.0213 | .7693±.0161 | .7171±.0223 | .6886±.0272 | .8209±.0347 | .8134±.0310 | .8102±.0234 |
| | **RGIB-REP** | .8103±.0137 | .7439±.0221 | .7040±.0192 | .8282±.0123 | .7857±.0142 | .7623±.0144 | .8365±.0163 | .8247±.0142 | .8240±.0119 |
| | **RGIB-SSL** | .8623±.0126 | .8080±.0240 | .7357±.0342 | .8632±.0187 | .7878±.0368 | .7310±.0483 | .9184±.0070 | .9120±.0108 | .9126±.0081 |

Table 17: Full results on Cora dataset with GAT.

| | | GAT | | | | | | | | |
|---|---|---|---|---|---|---|---|---|---|---|
| | | mixed noise | | | input noise | | | label noise | | |
| layers | method | 20% | 40% | 60% | 20% | 40% | 60% | 20% | 40% | 60% |
| L=2 | Standard | .8786±.0155 | .8489±.0131 | .8286±.0120 | .9014±.0070 | .8825±.0097 | .8880±.0077 | .8991±.0088 | .8841±.0097 | .8830±.0101 |
| | DropEdge | .8741±.0114 | .8279±.0172 | .8101±.0141 | .8930±.0056 | .8720±.0063 | .8586±.0110 | .8954±.0093 | .8788±.0083 | .8717±.0106 |
| | NeuralSparse | .8820±.0134 | .8447±.0151 | .8248±.0112 | .9051±.0075 | .8884±.0121 | .8828±.0067 | .8982±.0060 | .8855±.0110 | .8761±.0129 |
| | PTDNet | .8799±.0152 | .8487±.0101 | .8314±.0075 | .9023±.0105 | .8838±.0116 | .8827±.0090 | .9019±.0206 | .8772±.0082 | .8726±.0060 |
| | Co-teaching | .8883±.0148 | .8571±.0192 | .8378±.0169 | .9069±.0134 | .8896±.0173 | .8939±.0125 | .9289±.0281 | .9100±.0358 | .8969±.0243 |
| | Peer loss | .8867±.0189 | .8562±.0206 | .8343±.0140 | .9026±.0135 | .8908±.0169 | .8937±.0073 | .9057±.0212 | .8848±.0194 | .8918±.0217 |
| | Jaccard | .8809±.0180 | .8492±.0161 | .8334±.0124 | .9066±.0142 | .8823±.0080 | .9065±.0215 | .9010±.0092 | .8919±.0186 | .8929±.0114 |
| | GIB | .8826±.0192 | .8564±.0218 | .8375±.0294 | .9260±.0330 | .9092±.0361 | .9162±.0213 | .9007±.0128 | .8928±.0195 | .8915±.0098 |
| | SupCon | .8709±.0119 | .8462±.0121 | .8300±.0132 | .8957±.0115 | .8827±.0131 | .8805±.0116 | .8881±.0065 | .8730±.0116 | .8652±.0143 |
| | GRACE | .8286±.0224 | .7564±.0229 | .7328±.0213 | .8238±.0215 | .7615±.0379 | .7309±.0297 | .8833±.0100 | .8805±.0125 | .8807±.0119 |
| | **RGIB-REP** | .8759±.0132 | .8374±.0104 | .8269±.0134 | .9006±.0118 | .8833±.0079 | .8798±.0131 | .8993±.0086 | .8838±.0115 | .8810±.0125 |
| | **RGIB-SSL** | .9142±.0092 | .8878±.0135 | .8777±.0118 | .9234±.0053 | .8973±.0067 | .8866±.0148 | .9389±.0053 | .9347±.0088 | .9311±.0057 |
| L=4 | Standard | .8323±.0181 | .8005±.0212 | .7841±.0095 | .8616±.0107 | .8454±.0118 | .8377±.0146 | .8530±.0121 | .8357±.0082 | .8281±.0114 |
| | DropEdge | .8237±.0157 | .7782±.0072 | .7515±.0050 | .8548±.0144 | .8205±.0090 | .8000±.0151 | .8516±.0100 | .8373±.0142 | .8374±.0157 |
| | NeuralSparse | .8309±.0154 | .7954±.0118 | .7769±.0135 | .8575±.0174 | .8450±.0110 | .8277±.0148 | .8503±.0123 | .8395±.0122 | .8348±.0121 |
| | PTDNet | .8364±.0147 | .8045±.0096 | .7890±.0081 | .8669±.0132 | .8445±.0155 | .8331±.0146 | .8507±.0113 | .8399±.0096 | .8370±.0117 |
| | Co-teaching | .8294±.0222 | .8001±.0300 | .7895±.0171 | .8696±.0196 | .8344±.0163 | .8423±.0180 | .8534±.0145 | .8374±.0313 | .8544±.0169 |
| | Peer loss | .8344±.0276 | .8067±.0254 | .7933±.0142 | .8520±.0202 | .8353±.0145 | .8376±.0216 | .8533±.0121 | .8527±.0075 | .8444±.0203 |
| | Jaccard | .8319±.0237 | .8001±.0235 | .7932±.0107 | .8634±.0130 | .8449±.0272 | .8406±.0141 | .8537±.0171 | .8402±.0134 | .8354±.0152 |
| | GIB | .8352±.0367 | .8111±.0244 | .7945±.0279 | .8860±.0099 | .8579±.0163 | .8493±.0404 | .8604±.0166 | .8434±.0094 | .8340±.0130 |
| | SupCon | .8324±.0102 | .8033±.0099 | .7776±.0145 | .8620±.0162 | .8441±.0098 | .8337±.0386 | .8514±.0138 | .8381±.0092 | .8318±.0121 |
| | GRACE | .7403±.0347 | .6711±.0695 | .6656±.0578 | .7707±.0267 | .7154±.0366 | .7146±.0237 | .8040±.0447 | .7988±.0292 | .8321±.0230 |
| | **RGIB-REP** | .8274±.0153 | .7882±.0134 | .7552±.0657 | .8652±.0138 | .8370±.0118 | .8154±.0147 | .8480±.0181 | .8332±.0129 | .8259±.0163 |
| | **RGIB-SSL** | .8760±.0112 | .8469±.0101 | .8304±.0176 | .8865±.0125 | .8553±.0127 | .8349±.0130 | .9163±.0090 | .9075±.0087 | .9036±.0087 |
| L=6 | Standard | .8067±.0302 | .7514±.0389 | .7032±.0452 | .8414±.0127 | .7891±.0537 | .7678±.0641 | .8376±.0097 | .8154±.0069 | .8024±.0169 |
| | DropEdge | .8051±.0111 | .7375±.0354 | .7110±.0388 | .8198±.0132 | .7514±.0615 | .7248±.0511 | .8499±.0135 | .8312±.0163 | .8112±.0079 |
| | NeuralSparse | .8169±.0130 | .7726±.0165 | .7149±.0596 | .8443±.0183 | .7997±.0149 | .7273±.0592 | .8460±.0113 | .8257±.0123 | .8149±.0224 |
| | PTDNet | .8207±.0166 | .7460±.0714 | .7145±.0621 | .8253±.0514 | .8209±.0231 | .7759±.0339 | .8464±.0106 | .8234±.0168 | .8159±.0148 |
| | Co-teaching | .8059±.0312 | .7576±.0386 | .7070±.0455 | .8496±.0165 | .7969±.0542 | .7717±.0691 | .8549±.0298 | .8204±.0284 | .8215±.0369 |
| | Peer loss | .8133±.0336 | .7572±.0411 | .7070±.0521 | .8455±.0214 | .7938±.0547 | .7679±.0737 | .8405±.0229 | .8318±.0197 | .8049±.0224 |
| | Jaccard | .8155±.0400 | .7537±.0466 | .7123±.0454 | .8495±.0142 | .7947±.0561 | .7762±.0771 | .8374±.0160 | .8245±.0067 | .8108±.0180 |
| | GIB | .8188±.0386 | .7509±.0439 | .7014±.0442 | .8452±.0392 | .8039±.0706 | .7923±.0718 | .8366±.0137 | .8220±.0108 | .8090±.0222 |
| | SupCon | .7586±.0629 | .6434±.0457 | .6115±.0607 | .7535±.0856 | .7102±.0655 | .6241±.0433 | .8088±.0568 | .8040±.0384 | .7869±.0392 |
| | GRACE | .5748±.0659 | .5949±.0650 | .5611±.0608 | .5675±.0788 | .6125±.0782 | .5537±.0607 | .5632±.0583 | .5588±.0692 | .6176±.0961 |
| | **RGIB-REP** | .8148±.0158 | .7553±.0179 | .6842±.0264 | .8404±.0129 | .8001±.0178 | .7433±.0663 | .8366±.0107 | .8274±.0151 | .8192±.0179 |
| | **RGIB-SSL** | .8613±.0107 | .8194±.0158 | .7858±.0133 | .8657±.0118 | .8213±.0189 | .8045±.0153 | .9049±.0059 | .8960±.0141 | .8985±.0117 |

Table 18: Full results on Cora dataset with SAGE.

| | | SAGE | | | | | | | | |
|---|---|---|---|---|---|---|---|---|---|---|
| layers | method | mixed noise | | | input noise | | | label noise | | |
| | | 20% | 40% | 60% | 20% | 40% | 60% | 20% | 40% | 60% |
| L=2 | Standard | .8733±.0101 | .8520±.0137 | .8469±.0056 | .9006±.0074 | .8857±.0093 | .8917±.0063 | .8868±.0070 | .8695±.0106 | .8564±.0091 |
| | DropEdge | .8944±.0105 | .8600±.0083 | .8478±.0095 | .9013±.0080 | .9026±.0086 | .9050±.0084 | .8985±.0079 | .8725±.0107 | .8699±.0091 |
| | NeuralSparse | .8821±.0123 | .8570±.0090 | .8491±.0105 | .9020±.0090 | .8880±.0087 | .8949±.0098 | .8885±.0111 | .8681±.0110 | .8638±.0075 |
| | PTDNet | .8860±.0109 | .8536±.0101 | .8474±.0099 | .9040±.0101 | .8925±.0085 | .8947±.0070 | .8902±.0096 | .8721±.0068 | .8586±.0075 |
| | Co-teaching | .8794±.0097 | .8569±.0141 | .8557±.0080 | .8910±.0099 | .8891±.0100 | .8935±.0123 | .9071±.0216 | .8759±.0291 | .8632±.0156 |
| | Peer loss | .8817±.0102 | .8559±.0132 | .8480±.0058 | .9042±.0099 | .8866±.0174 | .8960±.0064 | .8876±.0111 | .8768±.0251 | .8739±.0149 |
| | Jaccard | .8828±.0170 | .8513±.0185 | .8474±.0081 | .9080±.0209 | .8975±.0210 | .8897±.0045 | .8922±.0112 | .8695±.0183 | .8596±.0133 |
| | GIB | .8765±.0196 | .8679±.0125 | .8546±.0246 | .9089±.0232 | .9004±.0098 | .8998±.0347 | .8938±.0082 | .8727±.0101 | .8614±.0125 |
| | SupCon | .8844±.0117 | .8507±.0135 | .8499±.0076 | .8974±.0075 | .8904±.0076 | .8943±.0122 | .8916±.0094 | .8742±.0092 | .8601±.0093 |
| | GRACE | .8123±.0113 | .7978±.0090 | .7944±.0137 | .8082±.0199 | .8010±.0160 | .7962±.0120 | .8280±.0092 | .8228±.0170 | .8261±.0102 |
| | **RGIB-REP** | .8748±.0094 | .8484±.0149 | .8380±.0102 | .9016±.0074 | .8876±.0075 | .8914±.0093 | .8863±.0078 | .8628±.0089 | .8449±.0077 |
| | **RGIB-SSL** | .9102±.0074 | .8967±.0114 | .8993±.0076 | .9196±.0072 | .9059±.0090 | .9082±.0084 | .9278±.0073 | .9174±.0089 | .9163±.0076 |
| L=4 | Standard | .8225±.0079 | .7833±.0093 | .7595±.0259 | .8607±.0110 | .8437±.0099 | .8387±.0183 | .8309±.0090 | .8046±.0074 | .7920±.0191 |
| | DropEdge | .8323±.0146 | .7944±.0130 | .7842±.0149 | .8734±.0082 | .8635±.0120 | .8628±.0146 | .8466±.0111 | .8199±.0155 | .8125±.0059 |
| | NeuralSparse | .8292±.0157 | .7930±.0108 | .7573±.0198 | .8703±.0108 | .8549±.0145 | .8596±.0061 | .8418±.0148 | .8025±.0148 | .8069±.0106 |
| | PTDNet | .8310±.0149 | .7847±.0174 | .7690±.0177 | .8626±.0141 | .8607±.0090 | .8630±.0154 | .8435±.0112 | .8125±.0117 | .7961±.0231 |
| | Co-teaching | .8237±.0139 | .7855±.0103 | .7658±.0343 | .8660±.0168 | .8533±.0149 | .8424±.0196 | .8580±.0345 | .8117±.0187 | .8088±.0275 |
| | Peer loss | .8230±.0107 | .7863±.0174 | .7626±.0294 | .8669±.0175 | .8534±.0179 | .8451±.0234 | .8363±.0142 | .8124±.0191 | .7904±.0231 |
| | Jaccard | .8261±.0153 | .7850±.0115 | .7602±.0348 | .8619±.0253 | .8465±.0227 | .8496±.0233 | .8378±.0106 | .8046±.0084 | .7976±.0288 |
| | GIB | .8286±.0059 | .7947±.0240 | .7729±.0378 | .8844±.0280 | .8437±.0158 | .8574±.0207 | .8374±.0185 | .8052±.0103 | .7947±.0229 |
| | SupCon | .8295±.0143 | .7809±.0176 | .7383±.0218 | .8568±.0115 | .8450±.0153 | .8445±.0187 | .8426±.0105 | .8150±.0170 | .7943±.0129 |
| | GRACE | .6242±.0245 | .6424±.0290 | .6711±.0452 | .6465±.0381 | .6172±.0320 | .6496±.0544 | .6434±.0384 | .6376±.0251 | .6438±.0449 |
| | **RGIB-REP** | .8274±.0112 | .7822±.0143 | .7692±.0202 | .8634±.0121 | .8470±.0144 | .8528±.0131 | .8367±.0149 | .8087±.0187 | .7991±.0120 |
| | **RGIB-SSL** | .8837±.0065 | .8728±.0116 | .8613±.0148 | .8960±.0109 | .8817±.0119 | .8825±.0113 | .9130±.0038 | .9041±.0075 | .9023±.0072 |
| L=6 | Standard | .7787±.0423 | .7420±.0251 | .7180±.0248 | .8256±.0222 | .7947±.0561 | .8005±.0421 | .8158±.0168 | .7707±.0235 | .7660±.0134 |
| | DropEdge | .8035±.0228 | .7398±.0560 | .7176±.0389 | .8262±.0153 | .8193±.0679 | .8089±.0260 | .8340±.0161 | .7993±.0091 | .7897±.0144 |
| | NeuralSparse | .7953±.0177 | .7378±.0180 | .7292±.0238 | .8384±.0120 | .8234±.0288 | .7980±.0701 | .8214±.0107 | .7908±.0136 | .7622±.0160 |
| | PTDNet | .7999±.0151 | .7604±.0169 | .7352±.0202 | .8311±.0143 | .8267±.0078 | .8109±.0140 | .8222±.0121 | .7823±.0078 | .7745±.0231 |
| | Co-teaching | .7817±.0477 | .7445±.0312 | .7212±.0332 | .8306±.0256 | .7991±.0595 | .8007±.0439 | .8324±.0256 | .7720±.0263 | .7687±.0266 |
| | Peer loss | .7781±.0451 | .7445±.0286 | .7192±.0277 | .8300±.0234 | .8020±.0624 | .8043±.0449 | .8309±.0149 | .7734±.0388 | .7652±.0262 |
| | Jaccard | .7779±.0437 | .7493±.0245 | .7277±.0238 | .8333±.0323 | .8075±.0605 | .8037±.0494 | .8148±.0186 | .7707±.0243 | .7709±.0204 |
| | GIB | .7814±.0493 | .7473±.0442 | .7349±.0437 | .8366±.0194 | .8106±.0689 | .8040±.0617 | .8172±.0258 | .7806±.0265 | .7689±.0180 |
| | SupCon | .7879±.0356 | .7019±.0285 | .6673±.0317 | .8219±.0469 | .7648±.0666 | .7159±.0717 | .8242±.0159 | .7880±.0152 | .7686±.0148 |
| | GRACE | .6866±.0160 | .6437±.0455 | .5967±.0248 | .6949±.0181 | .6536±.0365 | .6114±.0394 | .7239±.0231 | .7035±.0160 | .7014±.0111 |
| | **RGIB-REP** | .8049±.0146 | .7157±.0725 | .7099±.0473 | .8391±.0215 | .8149±.0234 | .7927±.0171 | .8358±.0100 | .7974±.0140 | .8046±.0135 |
| | **RGIB-SSL** | .8662±.0130 | .8430±.0178 | .8306±.0108 | .8746±.0091 | .8634±.0099 | .8603±.0156 | .8982±.0089 | .8930±.0108 | .8940±.0076 |

Table 19: Full results on Citeseer dataset with GCN.

| | | GCN | | | | | | | | |
|---|---|---|---|---|---|---|---|---|---|---|
| layers | method | mixed noise | | | input noise | | | label noise | | |
| | | 20% | 40% | 60% | 20% | 40% | 60% | 20% | 40% | 60% |
| L=2 | Standard | .8651±.0084 | .8355±.0154 | .8254±.0066 | .8767±.0097 | .8615±.0090 | .8585±.0085 | .8834±.0155 | .8649±.0104 | .8674±.0122 |
| | DropEdge | .8613±.0112 | .8317±.0168 | .8112±.0158 | .8755±.0117 | .8557±.0122 | .8483±.0164 | .8862±.0084 | .8695±.0133 | .8688±.0143 |
| | NeuralSparse | .8605±.0119 | .8402±.0138 | .8239±.0069 | .8801±.0085 | .8634±.0109 | .8614±.0134 | .8827±.0094 | .8753±.0111 | .8720±.0156 |
| | PTDNet | .8646±.0155 | .8404±.0111 | .8223±.0142 | .8805±.0095 | .8647±.0112 | .8571±.0116 | .8813±.0074 | .8734±.0126 | .8708±.0097 |
| | Co-teaching | .8689±.0100 | .8416±.0215 | .8278±.0061 | .8849±.0167 | .8659±.0180 | .8659±.0090 | .8833±.0231 | .8744±.0281 | .8847±.0094 |
| | Peer loss | .8728±.0159 | .8374±.0182 | .8308±.0062 | .8783±.0089 | .8665±.0152 | .8618±.0131 | .8909±.0144 | .8653±.0276 | .8808±.0212 |
| | Jaccard | .8682±.0164 | .8406±.0230 | .8292±.0137 | .8725±.0090 | .8743±.0186 | .8696±.0158 | .8849±.0231 | .8718±.0105 | .8680±.0154 |
| | GIB | .8825±.0119 | .8476±.0331 | .8306±.0156 | .8819±.0381 | .8660±.0318 | .8688±.0136 | .8914±.0253 | .8681±.0128 | .8707±.0144 |
| | SupCon | .8344±.0101 | .8173±.0155 | .8140±.0121 | .8501±.0100 | .8325±.0129 | .8387±.0186 | .8436±.0128 | .8337±.0139 | .8269±.0131 |
| | GRACE | .8450±.0123 | .8225±.0169 | .7898±.0081 | .8529±.0074 | .8209±.0137 | .7967±.0171 | .8697±.0110 | .8646±.0122 | .8699±.0156 |
| | **RGIB-REP** | .8585±.0088 | .8347±.0213 | .8167±.0113 | .8751±.0074 | .8637±.0122 | .8600±.0119 | .8795±.0101 | .8638±.0175 | .8644±.0095 |
| | **RGIB-SSL** | .9199±.0091 | .8957±.0099 | .8759±.0087 | .9271±.0055 | .9018±.0090 | .8942±.0145 | .9495±.0059 | .9515±.0049 | .9480±.0062 |
| L=4 | Standard | .7864±.0145 | .7380±.0201 | .7085±.0146 | .8054±.0240 | .7708±.0325 | .7583±.0258 | .7965±.0124 | .7850±.0174 | .7659±.0240 |
| | DropEdge | .7635±.0106 | .7393±.0170 | .7094±.0190 | .8025±.0106 | .7730±.0147 | .7473±.0161 | .7937±.0145 | .7853±.0109 | .7632±.0142 |
| | NeuralSparse | .7765±.0123 | .7397±.0168 | .7148±.0237 | .8093±.0129 | .7809±.0160 | .7468±.0289 | .7968±.0198 | .7921±.0129 | .7752±.0209 |
| | PTDNet | .7795±.0131 | .7423±.0200 | .7283±.0130 | .8119±.0103 | .7811±.0191 | .7638±.0167 | .7968±.0141 | .7765±.0124 | .7622±.0215 |
| | Co-teaching | .7533±.0181 | .7238±.0245 | .7131±.0157 | .8059±.0263 | .7753±.0408 | .7668±.0272 | .7974±.0381 | .7877±.0291 | .7913±.0343 |
| | Peer loss | .7423±.0215 | .7345±.0213 | .7104±.0242 | .8106±.0250 | .7767±.0400 | .7653±.0315 | .7991±.0241 | .7990±.0289 | .7751±.0245 |
| | Jaccard | .7473±.0160 | .7324±.0204 | .7107±.0172 | .8176±.0283 | .7776±.0471 | .7725±.0432 | .8061±.0187 | .7887±.0196 | .7689±.0310 |
| | GIB | .7509±.0336 | .7388±.0240 | .7121±.0210 | .8070±.0398 | .7717±.0612 | .7798±.0421 | .7986±.0120 | .7852±.0186 | .7649±.0275 |
| | SupCon | .7554±.0196 | .7458±.0176 | .7299±.0218 | .8076±.0099 | .7767±.0111 | .7655±.0164 | .8024±.0108 | .7983±.0123 | .7807±.0166 |
| | GRACE | .7632±.0224 | .7242±.0219 | .6844±.0226 | .7615±.0152 | .7151±.0193 | .6830±.0232 | .7909±.0211 | .7630±.0196 | .7737±.0307 |
| | **RGIB-REP** | .7875±.0131 | .7519±.0181 | .7312±.0227 | .8299±.0134 | .7996±.0130 | .7771±.0178 | .8083±.0152 | .7846±.0234 | .7945±.0203 |
| | **RGIB-SSL** | .8694±.0108 | .8427±.0174 | .8137±.0170 | .8747±.0144 | .8461±.0109 | .8245±.0142 | .9204±.0085 | .9218±.0098 | .9250±.0071 |
| L=6 | Standard | .7355±.0387 | .7110±.0222 | .7106±.0168 | .7720±.0161 | .7460±.0401 | .7212±.0470 | .7900±.0131 | .7741±.0157 | .7648±.0229 |
| | DropEdge | .7653±.0139 | .7191±.0159 | .7006±.0304 | .7645±.0441 | .7442±.0160 | .7209±.0182 | .8007±.0152 | .7823±.0141 | .7656±.0196 |
| | NeuralSparse | .7644±.0180 | .7361±.0191 | .7036±.0387 | .7913±.0152 | .7476±.0279 | .7391±.0235 | .7793±.0391 | .7739±.0188 | .7688±.0256 |
| | PTDNet | .7661±.0153 | .7401±.0134 | .7072±.0385 | .7882±.0134 | .7728±.0164 | .7464±.0188 | .7913±.0122 | .7742±.0155 | .7582±.0151 |
| | Co-teaching | .7375±.0468 | .7171±.0299 | .7188±.0248 | .7813±.0185 | .7550±.0475 | .7310±.0496 | .8034±.0117 | .7885±.0275 | .7701±.0245 |
| | Peer loss | .7398±.0387 | .7105±.0288 | .7166±.0250 | .7783±.0418 | .7532±.0488 | .7301±.0509 | .7915±.0169 | .7748±.0299 | .7695±.0239 |
| | Jaccard | .7415±.0444 | .7162±.0307 | .7135±.0171 | .7793±.0263 | .7459±.0416 | .7369±.0473 | .7925±.0176 | .7825±.0245 | .7710±.0298 |
| | GIB | .7370±.0531 | .7226±.0268 | .7175±.0234 | .7743±.0435 | .7722±.0697 | .7276±.0534 | .7948±.0178 | .7803±.0221 | .7681±.0251 |
| | SupCon | .7714±.0122 | .7413±.0166 | .7205±.0231 | .7907±.0161 | .7842±.0109 | .7670±.0183 | .7745±.0231 | .7659±.0114 | .7573±.0180 |
| | GRACE | .6995±.0221 | .6901±.0210 | .6946±.0152 | .7041±.0240 | .7037±.0247 | .6818±.0169 | .7752±.0230 | .7739±.0151 | .7773±.0351 |
| | **RGIB-REP** | .7725±.0077 | .7429±.0202 | .7232±.0110 | .7781±.0218 | .7559±.0130 | .7415±.0221 | .7900±.0252 | .7797±.0181 | .7886±.0194 |
| | **RGIB-SSL** | .8417±.0169 | .7995±.0148 | .7673±.0137 | .8379±.0065 | .8026±.0173 | .7793±.0207 | .8984±.0144 | .9062±.0066 | .9060±.0098 |

Table 20: Full results on Citeseer dataset with GAT.

| layers | method | mixed noise | | | input noise | | | label noise | | |
|---|---|---|---|---|---|---|---|---|---|---|
| | | 20% | 40% | 60% | 20% | 40% | 60% | 20% | 40% | 60% |
| L=2 | Standard | .8586±.0125 | .8338±.0127 | .8207±.0121 | .8689±.0096 | .8526±.0130 | .8512±.0174 | .8762±.0076 | .8650±.0102 | .8648±.0166 |
| | DropEdge | .8566±.0113 | .8333±.0183 | .8100±.0098 | .8750±.0079 | .8496±.0101 | .8512±.0121 | .8820±.0086 | .8679±.0112 | .8673±.0114 |
| | NeuralSparse | .8573±.0101 | .8431±.0151 | .8222±.0092 | .8743±.0117 | .8577±.0067 | .8580±.0135 | .8826±.0080 | .8724±.0076 | .8657±.0089 |
| | PTDNet | .8602±.0107 | .8381±.0137 | .8157±.0075 | .8755±.0090 | .8560±.0084 | .8574±.0154 | .8784±.0120 | .8693±.0098 | .8669±.0142 |
| | Co-teaching | .8628±.0220 | .8366±.0124 | .8199±.0194 | .8720±.0128 | .8521±.0139 | .8510±.0224 | .8924±.0122 | .8888±.0365 | .8919±.0305 |
| | Peer loss | .8637±.0125 | .8378±.0170 | .8235±.0120 | .8721±.0172 | .8529±.0173 | .8559±.0216 | .8878±.0185 | .8653±.0288 | .8631±.0258 |
| | Jaccard | .8615±.0197 | .8379±.0222 | .8223±.0124 | .8841±.0079 | .8556±.0119 | .8498±.0309 | .8843±.0143 | .8676±.0195 | .8661±.0256 |
| | GIB | .8610±.0230 | .8462±.0114 | .8324±.0316 | .8909±.0091 | .8823±.0188 | .8488±.0276 | .8781±.0135 | .8739±.0144 | .8741±.0156 |
| | SupCon | .8495±.0100 | .8138±.0174 | .8155±.0099 | .8611±.0086 | .8454±.0111 | .8393±.0172 | .8558±.0137 | .8459±.0170 | .8379±.0185 |
| | GRACE | .8092±.0221 | .7564±.0264 | .7479±.0278 | .8014±.0370 | .7628±.0240 | .7433±.0245 | .8788±.0146 | .8768±.0073 | .8654±.0172 |
| | **RGIB-REP** | .8545±.0108 | .8310±.0127 | .8137±.0091 | .8736±.0107 | .8566±.0097 | .8503±.0159 | .8778±.0093 | .8696±.0081 | .8614±.0084 |
| | **RGIB-SSL** | .9106±.0102 | .8829±.0058 | .8677±.0095 | .9172±.0072 | .8909±.0086 | .8785±.0121 | .9419±.0071 | .9410±.0047 | .9410±.0090 |
| L=4 | Standard | .8026±.0157 | .7775±.0248 | .7518±.0183 | .8191±.0092 | .8043±.0105 | .7912±.0073 | .8174±.0172 | .7998±.0143 | .7934±.0156 |
| | DropEdge | .8063±.0079 | .7624±.0211 | .7434±.0124 | .8171±.0132 | .7977±.0178 | .7814±.0162 | .8262±.0148 | .8103±.0178 | .8057±.0148 |
| | NeuralSparse | .7958±.0142 | .7761±.0172 | .7550±.0129 | .8282±.0130 | .8088±.0088 | .7911±.0174 | .8259±.0119 | .8135±.0092 | .7986±.0109 |
| | PTDNet | .8000±.0113 | .7734±.0198 | .7597±.0185 | .8254±.0105 | .8132±.0089 | .7950±.0143 | .8137±.0243 | .8082±.0094 | .8036±.0139 |
| | Co-teaching | .8016±.0184 | .7807±.0315 | .7521±.0267 | .8213±.0173 | .8068±.0156 | .7903±.0105 | .8402±.0220 | .8109±.0316 | .7947±.0350 |
| | Peer loss | .8064±.0178 | .7802±.0253 | .7544±.0191 | .8246±.0145 | .8108±.0122 | .7945±.0113 | .8160±.0329 | .8045±.0185 | .7925±.0207 |
| | Jaccard | .8098±.0222 | .7771±.0273 | .7517±.0186 | .8258±.0124 | .8083±.0138 | .7901±.0073 | .8206±.0168 | .8036±.0176 | .7999±.0215 |
| | GIB | .8170±.0230 | .7884±.0341 | .7645±.0247 | .8422±.0365 | .8112±.0212 | .7972±.0305 | .8192±.0249 | .8080±.0155 | .8010±.0177 |
| | SupCon | .7940±.0114 | .7728±.0125 | .7478±.0145 | .8137±.0115 | .8003±.0116 | .7777±.0409 | .8038±.0114 | .7972±.0198 | .7852±.0201 |
| | GRACE | .7319±.0433 | .6611±.0395 | .6449±.0579 | .7216±.0252 | .5947±.0660 | .6060±.0507 | .7775±.1040 | .7739±.0475 | .7882±.0328 |
| | **RGIB-REP** | .7991±.0107 | .7743±.0164 | .7418±.0121 | .8155±.0156 | .7905±.0157 | .7372±.0908 | .8108±.0118 | .7946±.0180 | .7935±.0131 |
| | **RGIB-SSL** | .8520±.0145 | .8306±.0149 | .8029±.0098 | .8592±.0120 | .8251±.0132 | .8145±.0110 | .9084±.0091 | .9101±.0076 | .9102±.0117 |
| L=6 | Standard | .7807±.0117 | .7373±.0270 | .7139±.0251 | .7970±.0134 | .7860±.0107 | .7741±.0126 | .7963±.0129 | .7883±.0162 | .7801±.0161 |
| | DropEdge | .7768±.0088 | .7477±.0195 | .7116±.0119 | .7854±.0232 | .7640±.0188 | .7425±.0362 | .8114±.0132 | .7840±.0217 | .7826±.0186 |
| | NeuralSparse | .7704±.0099 | .7462±.0170 | .7242±.0138 | .8047±.0101 | .7647±.0372 | .7248±.0596 | .8087±.0235 | .7855±.0176 | .7880±.0148 |
| | PTDNet | .7805±.0193 | .7503±.0223 | .7286±.0237 | .7927±.0287 | .7822±.0132 | .7579±.0355 | .8002±.0085 | .7977±.0134 | .7890±.0145 |
| | Co-teaching | .7819±.0141 | .7399±.0335 | .7236±.0292 | .7964±.0189 | .7809±.0183 | .7740±.0185 | .7933±.0406 | .7918±.0348 | .7979±.0245 |
| | Peer loss | .7846±.0214 | .7459±.0294 | .7187±.0259 | .7979±.0172 | .7955±.0168 | .7796±.0218 | .7957±.0273 | .7865±.0285 | .7912±.0148 |
| | Jaccard | .7902±.0117 | .7365±.0332 | .7157±.0248 | .8056±.0264 | .8038±.0226 | .7733±.0245 | .7964±.0226 | .7936±.0255 | .7847±.0218 |
| | GIB | .7818±.0230 | .7378±.0285 | .7137±.0416 | .8161±.0267 | .7995±.0183 | .7762±.0176 | .8002±.0155 | .7955±.0166 | .7794±.0244 |
| | SupCon | .7370±.0524 | .7160±.0462 | .6670±.0442 | .7667±.0402 | .7729±.0356 | .6999±.0597 | .7810±.0219 | .7752±.0119 | .7591±.0362 |
| | GRACE | .5068±.0128 | .5034±.0106 | .5108±.0319 | .5058±.0096 | .4956±.0069 | .5379±.0427 | .5181±.0547 | .5288±.0467 | .5068±.0178 |
| | **RGIB-REP** | .7817±.0129 | .7062±.0681 | .7254±.0188 | .7883±.0160 | .7769±.0168 | .7620±.0176 | .7981±.0092 | .7711±.0487 | .7817±.0164 |
| | **RGIB-SSL** | .8275±.0148 | .7989±.0136 | .7681±.0140 | .8261±.0096 | .8024±.0087 | .7806±.0174 | .8855±.0103 | .8918±.0143 | .8940±.0119 |

Table 21: Full results on Citeseer dataset with SAGE.

| layers | method | mixed noise | | | input noise | | | label noise | | |
|---|---|---|---|---|---|---|---|---|---|---|
| | | 20% | 40% | 60% | 20% | 40% | 60% | 20% | 40% | 60% |
| L=2 | Standard | .8438±.0143 | .8404±.0113 | .8279±.0098 | .8644±.0109 | .8608±.0064 | .8647±.0161 | .8560±.0120 | .8533±.0112 | .8412±.0122 |
| | DropEdge | .8654±.0118 | .8593±.0129 | .8503±.0125 | .8834±.0059 | .8791±.0108 | .8806±.0101 | .8809±.0085 | .8734±.0064 | .8710±.0170 |
| | NeuralSparse | .8658±.0079 | .8478±.0109 | .8437±.0069 | .8746±.0102 | .8674±.0086 | .8722±.0183 | .8670±.0120 | .8645±.0136 | .8485±.0105 |
| | PTDNet | .8620±.0125 | .8488±.0113 | .8477±.0078 | .8740±.0094 | .8675±.0120 | .8702±.0134 | .8657±.0138 | .8576±.0147 | .8530±.0073 |
| | Co-teaching | .8505±.0143 | .8436±.0179 | .8323±.0144 | .8652±.0133 | .8606±.0067 | .8650±.0247 | .8787±.0269 | .8660±.0101 | .8392±.0363 |
| | Peer loss | .8525±.0157 | .8440±.0120 | .8319±.0186 | .8661±.0199 | .8693±.0113 | .8735±.0151 | .8563±.0278 | .8535±.0107 | .8563±.0241 |
| | Jaccard | .8514±.0208 | .8492±.0143 | .8291±.0183 | .8691±.0282 | .8805±.0205 | .8715±.0152 | .8616±.0120 | .8629±.0210 | .8413±.0167 |
| | GIB | .8574±.0192 | .8577±.0134 | .8323±.0125 | .8693±.0186 | .8632±.0310 | .8739±.0405 | .8600±.0197 | .8576±.0182 | .8403±.0157 |
| | SupCon | .8344±.0106 | .8241±.0123 | .8168±.0111 | .8485±.0133 | .8493±.0147 | .8484±.0161 | .8487±.0109 | .8394±.0096 | .8323±.0113 |
| | GRACE | .8283±.0295 | .8319±.0192 | .8253±.0208 | .8361±.0190 | .8273±.0222 | .8351±.0134 | .8434±.0193 | .8422±.0119 | .8400±.0233 |
| | **RGIB-REP** | .8514±.0120 | .8359±.0093 | .8213±.0112 | .8614±.0118 | .8537±.0105 | .8664±.0153 | .8592±.0111 | .8533±.0134 | .8455±.0123 |
| | **RGIB-SSL** | .9003±.0104 | .8894±.0119 | .8916±.0078 | .9045±.0076 | .8945±.0097 | .8992±.0141 | .9143±.0082 | .9075±.0080 | .9087±.0108 |
| L=4 | Standard | .7914±.0101 | .7686±.0213 | .7539±.0149 | .8171±.0173 | .8226±.0141 | .8157±.0203 | .8068±.0116 | .7891±.0136 | .7705±.0172 |
| | DropEdge | .7889±.0182 | .7850±.0117 | .7678±.0206 | .8300±.0130 | .8032±.0953 | .8271±.0115 | .8058±.0118 | .7899±.0173 | .7826±.0282 |
| | NeuralSparse | .7934±.0115 | .7746±.0195 | .7639±.0175 | .8331±.0119 | .8242±.0176 | .8324±.0144 | .8111±.0146 | .7904±.0229 | .7747±.0242 |
| | PTDNet | .7972±.0105 | .7804±.0201 | .7563±.0230 | .8299±.0177 | .8259±.0119 | .8374±.0110 | .8121±.0168 | .7892±.0103 | .7845±.0166 |
| | Co-teaching | .7928±.0189 | .7679±.0230 | .7557±.0214 | .8226±.0172 | .8297±.0202 | .8256±.0263 | .8233±.0203 | .8083±.0218 | .7836±.0347 |
| | Peer loss | .7913±.0116 | .7710±.0307 | .7556±.0238 | .8197±.0264 | .8321±.0155 | .8239±.0275 | .8135±.0131 | .7940±.0194 | .7885±.0195 |
| | Jaccard | .7904±.0108 | .7691±.0269 | .7626±.0183 | .8359±.0251 | .8244±.0189 | .8290±.0264 | .8167±.0119 | .7972±.0157 | .7744±.0227 |
| | GIB | .7931±.0134 | .7739±.0254 | .7691±.0291 | .8199±.0278 | .8306±.0209 | .8217±.0273 | .8151±.0152 | .7947±.0170 | .7737±.0266 |
| | SupCon | .7870±.0130 | .7672±.0145 | .7641±.0125 | .8191±.0197 | .8081±.0138 | .8036±.0211 | .7982±.0137 | .7898±.0121 | .7783±.0172 |
| | GRACE | .6196±.0253 | .6404±.0258 | .6308±.0323 | .6286±.0287 | .6365±.0223 | .6198±.0157 | .6495±.0202 | .6431±.0210 | .6411±.0191 |
| | **RGIB-REP** | .7854±.0123 | .7703±.0123 | .7562±.0200 | .8195±.0140 | .8151±.0118 | .8134±.0221 | .8038±.0158 | .7863±.0140 | .7799±.0212 |
| | **RGIB-SSL** | .8545±.0157 | .8482±.0147 | .8352±.0127 | .8706±.0074 | .8525±.0118 | .8564±.0141 | .8867±.0088 | .8866±.0161 | .8903±.0124 |
| L=6 | Standard | .7708±.0168 | .7223±.0614 | .7204±.0236 | .8086±.0145 | .7997±.0183 | .7872±.0145 | .7903±.0219 | .7690±.0201 | .7564±.0096 |
| | DropEdge | .7756±.0143 | .7485±.0104 | .7290±.0261 | .8070±.0125 | .8060±.0165 | .8026±.0195 | .7861±.0174 | .7769±.0149 | .7534±.0212 |
| | NeuralSparse | .7757±.0175 | .7564±.0205 | .7306±.0173 | .8039±.0144 | .8008±.0157 | .8016±.0223 | .7902±.0189 | .7762±.0177 | .7588±.0160 |
| | PTDNet | .7844±.0116 | .7525±.0270 | .7435±.0108 | .8070±.0130 | .8128±.0108 | .8085±.0135 | .7948±.0165 | .7830±.0182 | .7634±.0210 |
| | Co-teaching | .7792±.0246 | .7248±.0642 | .7268±.0297 | .8087±.0136 | .8076±.0234 | .7864±.0147 | .8011±.0394 | .7671±.0406 | .7626±.0232 |
| | Peer loss | .7796±.0158 | .7213±.0628 | .7294±.0233 | .8135±.0225 | .8012±.0206 | .7914±.0208 | .8005±.0409 | .7710±.0316 | .7672±.0098 |
| | Jaccard | .7800±.0257 | .7267±.0606 | .7248±.0330 | .8166±.0328 | .8058±.0177 | .7980±.0250 | .7906±.0274 | .7749±.0269 | .7617±.0147 |
| | GIB | .7823±.0358 | .7352±.0808 | .7247±.0332 | .8180±.0388 | .8073±.0241 | .8049±.0378 | .7967±.0223 | .7710±.0267 | .7576±.0097 |
| | SupCon | .7649±.0176 | .7193±.0426 | .7040±.0231 | .7864±.0187 | .7802±.0219 | .7529±.0419 | .7775±.0174 | .7715±.0206 | .7517±.0235 |
| | GRACE | .6608±.0371 | .6767±.0441 | .6433±.0494 | .6591±.0344 | .6660±.0366 | .6670±.0292 | .6966±.0367 | .6678±.0682 | .6979±.0476 |
| | **RGIB-REP** | .7766±.0070 | .7479±.0150 | .7427±.0127 | .7981±.0104 | .7939±.0149 | .7894±.0238 | .7959±.0137 | .7865±.0163 | .7695±.0169 |
| | **RGIB-SSL** | .8372±.0169 | .8226±.0189 | .8184±.0135 | .8441±.0078 | .8252±.0129 | .8288±.0114 | .8795±.0112 | .8802±.0100 | .8749±.0178 |

Table 22: Full results on Pubmed dataset with GCN.

| | | GCN | | | | | | | | |
|---|---|---|---|---|---|---|---|---|---|---|
| | | mixed noise | | | input noise | | | label noise | | |
| layers | method | 20% | 40% | 60% | 20% | 40% | 60% | 20% | 40% | 60% |
| L=2 | Standard | .9473±.0019 | .9271±.0027 | .9141±.0041 | .9590±.0022 | .9468±.0022 | .9337±.0016 | .9646±.0024 | .9637±.0021 | .9597±.0022 |
| | DropEdge | .9394±.0025 | .9155±.0027 | .8994±.0036 | .9467±.0022 | .9302±.0021 | .9146±.0022 | .9594±.0026 | .9558±.0027 | .9519±.0019 |
| | NeuralSparse | .9479±.0021 | .9251±.0039 | .9120±.0029 | .9558±.0019 | .9315±.0033 | .9269±.0014 | .9654±.0015 | .9525±.0023 | .9588±.0017 |
| | PTDNet | .9467±.0018 | .9264±.0025 | .9111±.0032 | .9554±.0026 | .9320±.0035 | .9272±.0027 | .9651±.0018 | .9616±.0021 | .9584±.0021 |
| | Co-teaching | .9502±.0085 | .9335±.0104 | .9160±.0096 | .9510±.0098 | .9331±.0054 | .9255±.0113 | .9676±.0018 | .9560±.0127 | .9608±.0098 |
| | Peer loss | .9500±.0034 | .9339±.0044 | .9140±.0138 | .9558±.0068 | .9397±.0019 | .9283±.0090 | .9615±.0175 | .9521±.0178 | .9545±.0144 |
| | Jaccard | .9496±.0039 | .9325±.0087 | .9235±.0081 | .9554±.0019 | .9327±.0067 | .9230±.0159 | .9580±.0032 | .9532±.0086 | .9591±.0054 |
| | GIB | .9509±.0205 | .9268±.0074 | .9131±.0066 | .9599±.0303 | .9303±.0047 | .9228±.0234 | .9559±.0076 | .9573±.0040 | .9597±.0099 |
| | SupCon | .9345±.0020 | .9257±.0017 | .9118±.0031 | .9583±.0015 | .9345±.0030 | .9214±.0023 | .9625±.0021 | .9522±.0018 | .9506±.0010 |
| | GRACE | .9341±.0032 | .9319±.0027 | .9154±.0049 | .9409±.0046 | .9321±.0032 | .9225±.0078 | .9489±.0034 | .9516±.0027 | .9511±.0022 |
| | **RGIB-REP** | .9537±.0013 | .9368±.0018 | .9270±.0036 | .9579±.0022 | .9467±.0023 | .9365±.0016 | .9696±.0033 | .9690±.0023 | .9671±.0015 |
| | **RGIB-SSL** | .9585±.0022 | .9471±.0032 | .9301±.0021 | .9425±.0088 | .9305±.0098 | .9121±.0125 | .9719±.0020 | .9724±.0015 | .9711±.0012 |
| L=4 | Standard | .8870±.0041 | .8748±.0031 | .8641±.0041 | .8854±.0051 | .8759±.0031 | .8651±.0040 | .9030±.0082 | .9039±.0029 | .9070±.0062 |
| | DropEdge | .8711±.0149 | .8482±.0045 | .8354±.0062 | .8682±.0158 | .8456±.0036 | .8376±.0046 | .9313±.0071 | .9201±.0091 | .9240±.0077 |
| | NeuralSparse | .8908±.0080 | .8733±.0022 | .8630±.0049 | .8931±.0090 | .8720±.0043 | .8649±.0041 | .9272±.0108 | .9136±.0117 | .9089±.0084 |
| | PTDNet | .8872±.0071 | .8733±.0036 | .8623±.0050 | .8903±.0087 | .8776±.0078 | .8609±.0055 | .9219±.0122 | .9099±.0104 | .9093±.0101 |
| | Co-teaching | .8943±.0090 | .8760±.0117 | .8638±.0093 | .8931±.0045 | .8792±.0036 | .8606±.0083 | .9315±.0075 | .9291±.0327 | .9319±.0324 |
| | Peer loss | .8961±.0130 | .8815±.0099 | .8566±.0057 | .8917±.0076 | .8811±.0127 | .8643±.0129 | .9126±.0116 | .9101±.0046 | .9210±.0095 |
| | Jaccard | .8872±.0036 | .8803±.0060 | .8512±.0136 | .8987±.0221 | .8764±.0099 | .8639±.0073 | .9098±.0110 | .9135±.0116 | .9096±.0132 |
| | GIB | .8899±.0239 | .8729±.0205 | .8544±.0051 | .8932±.0256 | .8808±.0053 | .8618±.0317 | .9037±.0089 | .9114±.0065 | .9064±.0059 |
| | SupCon | .8853±.0061 | .8718±.0110 | .8525±.0108 | .8867±.0080 | .8739±.0033 | .8558±.0042 | .9131±.0068 | .9108±.0095 | .9162±.0125 |
| | GRACE | .8922±.0034 | .8749±.0098 | .8588±.0042 | .8810±.0034 | .8795±.0099 | .8593±.0040 | .9234±.0088 | .9252±.0052 | .9255±.0043 |
| | **RGIB-REP** | .9017±.0044 | .8834±.0082 | .8652±.0038 | .9008±.0033 | .8822±.0054 | .8687±.0056 | .9357±.0028 | .9343±.0062 | .9332±.0062 |
| | **RGIB-SSL** | .9225±.0125 | .8918±.0065 | .8697±.0053 | .9126±.0046 | .8889±.0052 | .8693±.0049 | .9594±.0026 | .9604±.0028 | .9613±.0023 |
| L=6 | Standard | .8870±.0056 | .8731±.0032 | .8640±.0036 | .8855±.0025 | .8742±.0029 | .8652±.0041 | .9050±.0083 | .9112±.0059 | .9063±.0036 |
| | DropEdge | .8623±.0039 | .8421±.0044 | .8342±.0058 | .8623±.0054 | .8407±.0051 | .8328±.0058 | .9140±.0054 | .9102±.0082 | .9092±.0072 |
| | NeuralSparse | .8814±.0049 | .8586±.0029 | .8603±.0060 | .8792±.0072 | .8695±.0031 | .8612±.0046 | .9140±.0065 | .9130±.0047 | .9080±.0077 |
| | PTDNet | .8807±.0053 | .8610±.0026 | .8518±.0040 | .8791±.0040 | .8708±.0032 | .8619±.0040 | .9123±.0074 | .9086±.0079 | .9105±.0072 |
| | Co-teaching | .8850±.0126 | .8698±.0101 | .8568±.0050 | .8773±.0023 | .8767±.0113 | .8691±.0095 | .9103±.0372 | .9311±.0290 | .9204±.0162 |
| | Peer loss | .8860±.0125 | .8633±.0079 | .8534±.0048 | .8734±.0023 | .8775±.0107 | .8668±.0042 | .9155±.0070 | .9172±.0089 | .9251±.0107 |
| | Jaccard | .8852±.0092 | .8658±.0085 | .8555±.0055 | .8711±.0031 | .8794±.0126 | .8670±.0200 | .9045±.0137 | .9118±.0108 | .9065±.0059 |
| | GIB | .8807±.0046 | .8738±.0140 | .8605±.0193 | .8718±.0070 | .8767±.0110 | .8623±.0114 | .9081±.0131 | .9136±.0147 | .9155±.0106 |
| | SupCon | .8716±.0090 | .8627±.0043 | .8533±.0045 | .8705±.0108 | .8733±.0026 | .8643±.0047 | .9232±.0081 | .9294±.0075 | .9218±.0092 |
| | GRACE | .8798±.0079 | .8664±.0031 | .8584±.0051 | .8806±.0043 | .8675±.0035 | .8591±.0045 | .9296±.0064 | .9190±.0052 | .9110±.0049 |
| | **RGIB-REP** | .8846±.0071 | .8715±.0035 | .8640±.0043 | .8818±.0054 | .8716±.0021 | .8646±.0047 | .9161±.0081 | .9156±.0075 | .9129±.0095 |
| | **RGIB-SSL** | .8915±.0062 | .8737±.0029 | .8633±.0036 | .8891±.0057 | .8732±.0034 | .8639±.0051 | .9450±.0042 | .9488±.0065 | .9484±.0027 |

Table 23: Full results on Pubmed dataset with GAT.

| | | GAT | | | | | | | | |
|---|---|---|---|---|---|---|---|---|---|---|
| | | mixed noise | | | input noise | | | label noise | | |
| layers | method | 20% | 40% | 60% | 20% | 40% | 60% | 20% | 40% | 60% |
| L=2 | Standard | .9173±.0028 | .8984±.0030 | .8884±.0033 | .9255±.0024 | .9176±.0035 | .9102±.0030 | .9306±.0038 | .9271±.0030 | .9232±.0034 |
| | DropEdge | .9102±.0023 | .8970±.0032 | .8837±.0034 | .9208±.0030 | .9177±.0028 | .9087±.0037 | .9374±.0031 | .9309±.0020 | .9267±.0019 |
| | NeuralSparse | .9106±.0021 | .8952±.0031 | .8840±.0029 | .9179±.0029 | .9210±.0032 | .9141±.0038 | .9310±.0039 | .9297±.0026 | .9252±.0021 |
| | PTDNet | .9119±.0015 | .8943±.0030 | .8844±.0037 | .9068±.0027 | .9210±.0026 | .9143±.0025 | .9311±.0031 | .9280±.0034 | .9267±.0024 |
| | Co-teaching | .9163±.0044 | .8924±.0050 | .8881±.0100 | .9058±.0089 | .9180±.0089 | .9136±.0068 | .9556±.0098 | .9364±.0207 | .9376±.0249 |
| | Peer loss | .9156±.0115 | .8968±.0083 | .8823±.0041 | .9147±.0055 | .9171±.0089 | .9173±.0070 | .9407±.0225 | .9296±.0084 | .9219±.0229 |
| | Jaccard | .9128±.0057 | .8940±.0094 | .8831±.0086 | .9042±.0092 | .9181±.0175 | .9228±.0214 | .9304±.0039 | .9355±.0127 | .9242±.0109 |
| | GIB | .9136±.0061 | .8965±.0110 | .8881±.0115 | .9052±.0313 | .9208±.0097 | .9383±.0268 | .9366±.0069 | .9291±.0065 | .9277±.0089 |
| | SupCon | .9072±.0036 | .8881±.0037 | .8763±.0027 | .9076±.0025 | .9080±.0033 | .9002±.0037 | .9202±.0032 | .9123±.0040 | .9059±.0027 |
| | GRACE | .8230±.0512 | .7882±.0693 | .7914±.0767 | .8273±.0671 | .8053±.0498 | .7993±.0695 | .8792±.0342 | .8926±.0257 | .8946±.0211 |
| | **RGIB-REP** | .9190±.0025 | .9034±.0017 | .8939±.0056 | .9250±.0033 | .9164±.0024 | .9099±.0029 | .9311±.0036 | .9306±.0025 | .9276±.0022 |
| | **RGIB-SSL** | .9223±.0027 | .9054±.0032 | .8960±.0027 | .9183±.0021 | .9071±.0023 | .8977±.0029 | .9405±.0027 | .9407±.0022 | .9378±.0018 |
| L=4 | Standard | .8610±.0045 | .8434±.0042 | .8339±.0048 | .8668±.0040 | .8547±.0050 | .8476±.0037 | .8817±.0042 | .8772±.0040 | .8696±.0062 |
| | DropEdge | .8600±.0034 | .8462±.0028 | .8325±.0044 | .8605±.0041 | .8557±.0057 | .8454±.0029 | .8852±.0052 | .8851±.0055 | .8871±.0034 |
| | NeuralSparse | .8691±.0034 | .8505±.0025 | .8402±.0055 | .8612±.0042 | .8572±.0048 | .8459±.0038 | .8906±.0059 | .8806±.0076 | .8796±.0075 |
| | PTDNet | .8614±.0038 | .8568±.0026 | .8372±.0338 | .8640±.0035 | .8541±.0050 | .8562±.0027 | .8909±.0052 | .8855±.0054 | .8785±.0064 |
| | Co-teaching | .8688±.0125 | .8450±.0065 | .8413±.0081 | .8666±.0066 | .8556±.0103 | .8468±.0118 | .8792±.0107 | .8946±.0335 | .8968±.0132 |
| | Peer loss | .8613±.0040 | .8493±.0105 | .8342±.0108 | .8652±.0061 | .8504±.0143 | .8471±.0059 | .8851±.0110 | .8839±.0085 | .8735±.0069 |
| | Jaccard | .8630±.0089 | .8502±.0082 | .8365±.0136 | .8677±.0198 | .8495±.0065 | .8485±.0116 | .8863±.0078 | .8846±.0060 | .8750±.0119 |
| | GIB | .8616±.0036 | .8534±.0143 | .8370±.0141 | .8814±.0189 | .8486±.0059 | .8421±.0220 | .8828±.0111 | .8827±.0066 | .8700±.0076 |
| | SupCon | .8534±.0061 | .8292±.0128 | .8048±.0164 | .8625±.0085 | .8447±.0069 | .8360±.0141 | .8768±.0057 | .8686±.0062 | .8629±.0065 |
| | GRACE | .8355±.0477 | .8202±.0387 | .8046±.0409 | .8439±.0344 | .8078±.0649 | .7822±.0686 | .7878±.0839 | .8185±.0702 | .8008±.0794 |
| | **RGIB-REP** | .8656±.0043 | .8443±.0057 | .8339±.0033 | .8674±.0058 | .8528±.0043 | .8392±.0036 | .8909±.0084 | .8925±.0051 | .8833±.0062 |
| | **RGIB-SSL** | .8891±.0037 | .8651±.0036 | .8480±.0068 | .8878±.0039 | .8650±.0044 | .8489±.0055 | .9276±.0042 | .9277±.0022 | .9264±.0024 |
| L=6 | Standard | .8315±.0059 | .8116±.0073 | .8040±.0092 | .8374±.0036 | .8201±.0321 | .8067±.0306 | .8480±.0086 | .8414±.0116 | .8313±.0071 |
| | DropEdge | .8468±.0407 | .8236±.0050 | .8004±.0071 | .8655±.0037 | .8303±.0044 | .7915±.0334 | .8501±.0058 | .8568±.0068 | .8523±.0381 |
| | NeuralSparse | .8426±.0035 | .8234±.0032 | .8098±.0067 | .8323±.0342 | .8258±.0090 | .7977±.0295 | .8622±.0062 | .8563±.0091 | .8577±.0075 |
| | PTDNet | .8384±.0319 | .8117±.0384 | .8031±.0380 | .8508±.0044 | .8251±.0284 | .8204±.0034 | .8635±.0064 | .8498±.0251 | .8452±.0065 |
| | Co-teaching | .8360±.0122 | .8111±.0122 | .8081±.0113 | .8391±.0101 | .8246±.0398 | .8071±.0312 | .8729±.0090 | .8647±.0107 | .8424±.0079 |
| | Peer loss | .8354±.0128 | .8186±.0111 | .8044±.0104 | .8416±.0122 | .8277±.0359 | .8133±.0336 | .8504±.0132 | .8563±.0215 | .8310±.0089 |
| | Jaccard | .8333±.0140 | .8191±.0073 | .8044±.0082 | .8408±.0221 | .8340±.0424 | .8216±.0497 | .8528±.0100 | .8507±.0210 | .8304±.0108 |
| | GIB | .8316±.0125 | .8266±.0055 | .8064±.0241 | .8359±.0136 | .8205±.0345 | .8274±.0430 | .8551±.0149 | .8493±.0134 | .8361±.0072 |
| | SupCon | .8218±.0236 | .7763±.0306 | .7113±.0241 | .8137±.0821 | .8025±.0286 | .7555±.0297 | .8543±.0139 | .8390±.0370 | .8540±.0071 |
| | GRACE | .7773±.0737 | .7729±.0344 | .6821±.1217 | .7015±.1340 | .6953±.0805 | .6643±.1009 | .5212±.0262 | .5455±.0459 | .5743±.0549 |
| | **RGIB-REP** | .8324±.0039 | .8162±.0088 | .7934±.0068 | .8248±.0070 | .8145±.0088 | .7892±.0107 | .8518±.0056 | .8496±.0083 | .8492±.0113 |
| | **RGIB-SSL** | .8738±.0051 | .8443±.0033 | .8124±.0084 | .8709±.0023 | .8432±.0046 | .8187±.0050 | .9208±.0043 | .9250±.0029 | .9235±.0031 |

Table 24: Full results on Pubmed dataset with SAGE.

| | | SAGE | | | | | | | | |
|---|---|---|---|---|---|---|---|---|---|---|
| layers | method | mixed noise | | | input noise | | | label noise | | |
| | | 20% | 40% | 60% | 20% | 40% | 60% | 20% | 40% | 60% |
| L=2 | Standard | .9136±.0032 | .9094±.0035 | .9035±.0040 | .9295±.0025 | .9378±.0022 | .9410±.0023 | .8817±.0034 | .8793±.0043 | .8757±.0041 |
| | DropEdge | .9142±.0026 | .9188±.0029 | .9014±.0038 | .9179±.0020 | .9178±.0013 | .9165±.0013 | .9021±.0021 | .9003±.0018 | .9011±.0020 |
| | NeuralSparse | .9118±.0030 | .9137±.0031 | .9094±.0034 | .9271±.0019 | .9209±.0026 | .9234±.0017 | .9033±.0028 | .8925±.0030 | .8864±.0032 |
| | PTDNet | .9114±.0026 | .9138±.0030 | .9104±.0037 | .9267±.0020 | .9211±.0027 | .9229±.0024 | .9030±.0028 | .8941±.0040 | .8868±.0033 |
| | Co-teaching | .9164±.0031 | .9160±.0027 | .9092±.0132 | .9362±.0061 | .9258±.0072 | .9228±.0017 | .8827±.0026 | .9054±.0236 | .8856±.0113 |
| | Peer loss | .9103±.0104 | .9174±.0094 | .9044±.0087 | .9308±.0079 | .9238±.0051 | .9307±.0076 | .9005±.0118 | .8833±.0066 | .8909±.0058 |
| | Jaccard | .9108±.0123 | .9167±.0105 | .9109±.0031 | .9236±.0098 | .9259±.0175 | .9308±.0141 | .8861±.0052 | .8785±.0092 | .8842±.0048 |
| | GIB | .9173±.0049 | .9125±.0211 | .9096±.0020 | .9338±.0152 | .9294±.0125 | .9299±.0194 | .8880±.0125 | .8865±.0129 | .8853±.0058 |
| | SupCon | .9167±.0026 | .9148±.0031 | .9106±.0035 | .9259±.0034 | .9305±.0031 | .9230±.0033 | .9033±.0055 | .8929±.0051 | .8888±.0034 |
| | GRACE | .8317±.0063 | .8291±.0059 | .8336±.0068 | .8288±.0071 | .8344±.0074 | .8305±.0052 | .8320±.0046 | .8344±.0078 | .8328±.0075 |
| | **RGIB-REP** | .9192±.0022 | .9136±.0027 | .9119±.0037 | .9278±.0031 | .9322±.0031 | .9331±.0034 | .9055±.0033 | .9015±.0051 | .9000±.0043 |
| | **RGIB-SSL** | .9276±.0052 | .9285±.0030 | .9313±.0031 | .9273±.0056 | .9305±.0040 | .9334±.0025 | .9304±.0017 | .9294±.0049 | .9274±.0066 |
| L=4 | Standard | .8627±.0056 | .8663±.0061 | .8619±.0073 | .8715±.0080 | .8901±.0082 | .9033±.0057 | .8386±.0085 | .8330±.0104 | .8268±.0092 |
| | DropEdge | .8737±.0060 | .8740±.0065 | .8816±.0077 | .9072±.0120 | .9016±.0078 | .9087±.0019 | .9037±.0024 | .9036±.0048 | .9048±.0059 |
| | NeuralSparse | .8888±.0037 | .8825±.0046 | .8822±.0051 | .8982±.0067 | .9146±.0029 | .9190±.0041 | .8647±.0075 | .8535±.0091 | .8472±.0079 |
| | PTDNet | .8842±.0045 | .8853±.0059 | .8846±.0052 | .8885±.0055 | .9057±.0055 | .9127±.0058 | .8464±.0105 | .8458±.0066 | .8408±.0065 |
| | Co-teaching | .8678±.0104 | .8752±.0090 | .8667±.0122 | .8727±.0079 | .8971±.0159 | .9061±.0098 | .8601±.0240 | .8498±.0368 | .8514±.0388 |
| | Peer loss | .8683±.0136 | .8717±.0100 | .8665±.0117 | .8736±.0075 | .8997±.0079 | .9095±.0119 | .8510±.0151 | .8514±.0115 | .8423±.0128 |
| | Jaccard | .8646±.0124 | .8725±.0141 | .8625±.0140 | .8773±.0125 | .9017±.0205 | .9155±.0164 | .8378±.0137 | .8383±.0118 | .8275±.0083 |
| | GIB | .8816±.0203 | .8768±.0098 | .8641±.0238 | .8824±.0102 | .8918±.0095 | .9221±.0243 | .8479±.0103 | .8410±.0151 | .8330±.0120 |
| | SupCon | .8765±.0086 | .8682±.0100 | .8518±.0198 | .8907±.0054 | .8957±.0104 | .8879±.0113 | .8865±.0046 | .8811±.0052 | .8810±.0044 |
| | GRACE | .7142±.0451 | .6741±.0484 | .7227±.0446 | .6578±.0701 | .6816±.0536 | .7012±.0516 | .6498±.0896 | .6647±.0535 | .6540±.0424 |
| | **RGIB-REP** | .8746±.0049 | .8708±.0064 | .8758±.0071 | .8800±.0045 | .8904±.0048 | .8941±.0051 | .8504±.0078 | .8466±.0070 | .8533±.0118 |
| | **RGIB-SSL** | .9315±.0036 | .9237±.0035 | .9214±.0033 | .9276±.0046 | .9310±.0039 | .9286±.0035 | .9463±.0030 | .9448±.0037 | .9430±.0021 |
| L=6 | Standard | .8314±.0081 | .8105±.0279 | .8333±.0089 | .8224±.0335 | .8538±.0105 | .8566±.0199 | .8242±.0104 | .8161±.0071 | .8090±.0129 |
| | DropEdge | .8781±.0041 | .8665±.0076 | .8556±.0081 | .8800±.0050 | .8731±.0122 | .8628±.0152 | .8503±.0070 | .8698±.0074 | .8426±.0096 |
| | NeuralSparse | .8611±.0101 | .8589±.0081 | .8482±.0182 | .8694±.0083 | .8858±.0068 | .8889±.0057 | .8368±.0074 | .8344±.0090 | .8353±.0081 |
| | PTDNet | .8425±.0073 | .8511±.0067 | .8525±.0064 | .8430±.0107 | .8627±.0115 | .8739±.0103 | .8232±.0076 | .8195±.0071 | .8198±.0060 |
| | Co-teaching | .8385±.0114 | .8202±.0303 | .8349±.0147 | .8265±.0385 | .8607±.0164 | .8612±.0190 | .8299±.0352 | .8243±.0074 | .8143±.0409 |
| | Peer loss | .8339±.0162 | .8111±.0285 | .8335±.0084 | .8317±.0412 | .8610±.0135 | .8605±.0223 | .8277±.0224 | .8260±.0197 | .8204±.0316 |
| | Jaccard | .8355±.0110 | .8180±.0272 | .8347±.0124 | .8298±.0436 | .8564±.0107 | .8607±.0394 | .8260±.0138 | .8207±.0130 | .8140±.0192 |
| | GIB | .8309±.0062 | .8195±.0329 | .8336±.0206 | .8226±.0355 | .8527±.0208 | .8647±.0344 | .8310±.0142 | .8237±.0128 | .8141±.0165 |
| | SupCon | .8288±.0109 | .8276±.0351 | .8174±.0070 | .8617±.0122 | .8443±.0528 | .8365±.0350 | .8745±.0071 | .8692±.0132 | .8661±.0076 |
| | GRACE | .7293±.0133 | .7151±.0368 | .6737±.0522 | .7102±.0296 | .7392±.0257 | .6751±.0444 | .7120±.0166 | .7042±.0588 | .7230±.0286 |
| | **RGIB-REP** | .8493±.0057 | .8424±.0083 | .8373±.0079 | .8439±.0063 | .8527±.0092 | .8536±.0112 | .8275±.0218 | .8148±.0098 | .8211±.0145 |
| | **RGIB-SSL** | .8915±.0060 | .8971±.0051 | .8956±.0046 | .8948±.0079 | .9026±.0042 | .8995±.0040 | .9313±.0042 | .9311±.0036 | .9294±.0053 |

Table 25: Full results on Facebook dataset with GCN.

| | | GCN | | | | | | | | |
|---|---|---|---|---|---|---|---|---|---|---|
| layers | method | mixed noise | | | input noise | | | label noise | | |
| | | 20% | 40% | 60% | 20% | 40% | 60% | 20% | 40% | 60% |
| L=2 | Standard | .9880±.0007 | .9866±.0007 | .9843±.0010 | .9878±.0008 | .9852±.0006 | .9834±.0011 | .9892±.0006 | .9888±.0008 | .9888±.0007 |
| | DropEdge | .9865±.0007 | .9845±.0008 | .9821±.0009 | .9764±.0010 | .9835±.0007 | .9708±.0012 | .9783±.0006 | .9879±.0008 | .9881±.0007 |
| | NeuralSparse | .9877±.0007 | .9861±.0008 | .9837±.0009 | .9772±.0007 | .9852±.0007 | .9732±.0010 | .9788±.0006 | .9885±.0007 | .9888±.0005 |
| | PTDNet | .9876±.0008 | .9862±.0008 | .9838±.0008 | .9776±.0007 | .9848±.0008 | .9728±.0011 | .9859±.0006 | .9786±.0009 | .9886±.0006 |
| | Co-teaching | .9871±.0052 | .9722±.0104 | .9876±.0009 | .9738±.0018 | .9848±.0077 | .9750±.0002 | .9716±.0266 | .9873±.0121 | .9825±.0009 |
| | Peer loss | .9755±.0095 | .9737±.0033 | .9895±.0096 | .9797±.0024 | .9855±.0051 | .9720±.0019 | .9763±.0001 | .9732±.0110 | .9774±.0034 |
| | Jaccard | .9759±.0043 | .9796±.0056 | .9707±.0006 | .9739±.0098 | .9807±.0040 | .9793±.0150 | .9729±.0030 | .9773±.0044 | .9729±.0042 |
| | GIB | .9752±.0009 | .9735±0.021 | .9720±.0154 | .9796±.0245 | .9820±.0173 | .9707±.0024 | .9724±.0017 | .9774±.1404 | .9717±.0086 |
| | SupCon | .9779±.0008 | .9748±.0014 | .9665±.0022 | .9724±.0009 | .9803±.0011 | .9783±.0010 | .9713±.0066 | .9569±.0160 | .9701±.0024 |
| | GRACE | .8883±.0120 | .8811±.0077 | .8412±.0224 | .8950±.0081 | .8792±.0119 | .8530±.0221 | .8864±.0077 | .8826±.0051 | .8784±.0104 |
| | **RGIB-REP** | .9881±.0007 | .9866±.0008 | .9846±.0009 | .9880±.0007 | .9854±.0005 | .9841±.0011 | .9895±.0007 | .9891±.0008 | .9894±.0008 |
| | **RGIB-SSL** | .9845±.0005 | .9810±.0005 | .9765±.0014 | .9840±.0011 | .9808±.0012 | .9787±.0009 | .9862±.0005 | .9859±.0010 | .9866±.0007 |
| L=4 | Standard | .9829±.0020 | .9520±.0424 | .9438±.0402 | .9819±.0015 | .9668±.0147 | .9622±.0154 | .9882±.0007 | .9880±.0006 | .9886±.0006 |
| | DropEdge | .9811±.0028 | .9682±.0096 | .9473±.0120 | .9803±.0016 | .9685±.0033 | .9531±.0112 | .9673±.0008 | .9771±.0008 | .9776±.0009 |
| | NeuralSparse | .9825±.0020 | .9638±.0039 | .9456±.0067 | .9712±.0022 | .9691±.0045 | .9583±.0071 | .9781±.0007 | .9781±.0008 | .9784±.0008 |
| | PTDNet | .9725±.0017 | .9674±.0131 | .9485±.0049 | .9725±.0014 | .9668±.0089 | .9493±.0267 | .9879±.0003 | .9880±.0008 | .9783±.0008 |
| | Co-teaching | .9820±.0113 | .9526±.0439 | .9480±.0450 | .9712±.0084 | .9707±.0141 | .9714±.0146 | .9762±.0070 | .9797±.0167 | .9638±.0154 |
| | Peer loss | .9807±.0016 | .9536±.0503 | .9430±.0437 | .9758±.0011 | .9703±.0204 | .9622±.0176 | .9769±.0085 | .9750±.0128 | .9734±.0137 |
| | Jaccard | .9794±.0031 | .9579±.0471 | .9428±.0452 | .9784±.0127 | .9702±.0295 | .9638±.0340 | .9702±.0016 | .9725±.0064 | .9758±.0035 |
| | GIB | .9773±.0182 | .9608±.0548 | .9417±.0441 | .9796±.0260 | .9647±.0200 | .9650±.0373 | .9742±.0073 | .9703±.0023 | .9771±.0030 |
| | SupCon | .9588±.0067 | .9508±.0100 | .9297±.0121 | .9647±.0088 | .9517±.0113 | .9401±.0135 | .9647±.0164 | .9567±.0149 | .9553±.0133 |
| | GRACE | .8899±.0100 | .8865±.0226 | .8315±.0108 | .9015±.0049 | .8833±.0285 | .8395±.0157 | .8913±.0036 | .8972±.0046 | .8887±.0123 |
| | **RGIB-REP** | .9832±.0026 | .9770±.0032 | .9519±.0063 | .9833±.0014 | .9723±.0062 | .9682±.0087 | .9884±.0006 | .9883±.0007 | .9889±.0007 |
| | **RGIB-SSL** | .9829±.0023 | .9711±.0025 | .9643±.0029 | .9821±.0019 | .9707±.0021 | .9668±.0040 | .9857±.0010 | .9881±.0008 | .9857±.0007 |
| L=6 | Standard | .9798±.0013 | .9609±.0138 | .9368±.0179 | .9764±.0034 | .9502±.0096 | .9469±.0160 | .9863±.0013 | .9865±.0012 | .9876±.0010 |
| | DropEdge | .9774±.0030 | .9635±.0063 | .9500±.0103 | .9540±.0034 | .9579±.0083 | .9411±.0084 | .9556±.0014 | .9642±.0006 | .9559±.0012 |
| | NeuralSparse | .9774±.0020 | .9596±.0067 | .9435±.0114 | .9449±.0029 | .9496±.0073 | .9490±.0087 | .9564±.0012 | .9562±.0011 | .9465±.0016 |
| | PTDNet | .9767±.0066 | .9606±.0131 | .9449±.0175 | .9555±.0034 | .9568±.0064 | .9416±.0145 | .9560±.0011 | .9460±.0014 | .9569±.0009 |
| | Co-teaching | .9638±.0051 | .9565±.0235 | .9426±.0261 | .9520±.0051 | .9568±.0148 | .9489±.0228 | .9474±.0068 | .9458±.0143 | .9561±.0169 |
| | Peer loss | .9699±.0045 | .9557±.0141 | .9362±.0221 | .9554±.0053 | .9494±.0135 | .9445±.0234 | .9498±.0063 | .9581±.0200 | .9650±.0021 |
| | Jaccard | .9641±.0032 | .9553±.0137 | .9425±.0178 | .9507±.0064 | .9616±.0125 | .9480±.0170 | .9528±.0060 | .9525±.0075 | .9423±.0057 |
| | GIB | .9649±.0145 | .9602±.0198 | .9416±.0236 | .9511±.0333 | .9552±.0261 | .9555±.0346 | .9556±.0099 | .9430±.0106 | .9578±.0052 |
| | SupCon | .9462±.0050 | .9377±.0054 | .9282±.0113 | .9460±.0073 | .9355±.0118 | .9253±.0110 | .9411±.0088 | .9550±.0087 | .9510±.0122 |
| | GRACE | .8939±.0118 | .8563±.0206 | .8259±.0019 | .9071±.0077 | .8582±.0228 | .8274±.0029 | .8962±.0093 | .9026±.0035 | .8977±.0100 |
| | **RGIB-REP** | .9774±.0031 | .9680±.0097 | .9501±.0134 | .9780±.0024 | .9619±.0069 | .9423±.0085 | .9868±.0008 | .9863±.0010 | .9872±.0011 |
| | **RGIB-SSL** | .9751±.0023 | .9641±.0037 | .9395±.0122 | .9763±.0034 | .9633±.0040 | .9451±.0084 | .9845±.0008 | .9843±.0012 | .9847±.0011 |

Table 26: Full results on Facebook dataset with GAT.

| layers | method | GAT mixed noise 20% | 40% | 60% | input noise 20% | 40% | 60% | label noise 20% | 40% | 60% |
|---|---|---|---|---|---|---|---|---|---|---|
| L=2 | Standard | .9869±.0005 | .9856±.0008 | .9836±.0007 | .9878±.0006 | .9861±.0006 | .9858±.0008 | .9872±.0006 | .9864±.0009 | .9857±.0008 |
| | DropEdge | .9645±.0007 | .9532±.0010 | .9517±.0009 | .9652±.0007 | .9636±.0007 | .9733±.0008 | .9755±.0007 | .9743±.0010 | .9739±.0012 |
| | NeuralSparse | .9661±.0007 | .9651±.0009 | .9638±.0006 | .9765±.0008 | .9652±.0007 | .9744±.0006 | .9662±.0010 | .9657±.0006 | .9656±.0007 |
| | PTDNet | .9762±.0007 | .9551±.0008 | .9735±.0007 | .9766±.0007 | .9650±.0007 | .9645±.0009 | .9766±.0006 | .9759±.0009 | .9757±.0008 |
| | Co-teaching | .9723±.0006 | .9685±.0054 | .9762±.0039 | .9632±.0057 | .9720±.0001 | .9741±.0046 | .9703±0.001 | .9789±.0232 | .9792±.0288 |
| | Peer loss | .9762±.0032 | .9644±.0020 | .9725±.0076 | .9699±.0020 | .9720±.0104 | .9708±.0103 | .9632±.0194 | .9807±.0207 | .9713±.0078 |
| | Jaccard | .9771±.0038 | .9661±.0057 | .9736±.0078 | .9556±.0064 | .9758±.0180 | .9780±.0012 | .9720±.0100 | .9749±.0064 | .9701±.0087 |
| | GIB | .9879±.0075 | .9616±.0007 | .9709±.0199 | .9564±.0122 | .9746±.0004 | .9645±.0225 | .9758±.0101 | .9738±.0019 | .9799±.0094 |
| | SupCon | .9718±.0028 | .9646±.0085 | .9659±.0087 | .9753±.0008 | .9737±.0009 | .9629±.0009 | .9720±.0038 | .9456±.0038 | .9419±.0029 |
| | GRACE | .8377±.0309 | .8716±.0263 | .8519±.0264 | .8343±.0341 | .8502±.0271 | .8487±.0407 | .8768±.0457 | .8370±.0416 | .8794±.0319 |
| | **RGIB-REP** | .9865±.0006 | .9854±.0009 | .9837±.0006 | .9872±.0007 | .9855±.0007 | .9855±.0008 | .9871±.0007 | .9861±.0009 | .9859±.0006 |
| | **RGIB-SSL** | .9788±.0011 | .9764±.0017 | .9717±.0021 | .9784±.0012 | .9764±.0015 | .9734±.0021 | .9778±.0008 | .9784±.0009 | .9791±.0009 |
| L=4 | Standard | .9857±.0015 | .9850±.0012 | .9820±.0014 | .9855±.0011 | .9827±.0019 | .9773±.0046 | .9873±.0010 | .9874±.0010 | .9874±.0005 |
| | DropEdge | .9839±.0009 | .9791±.0032 | .9767±.0024 | .9836±.0009 | .9772±.0022 | .9705±.0030 | .9852±.0005 | .9845±.0009 | .9844±.0007 |
| | NeuralSparse | .9854±.0009 | .9812±.0019 | .9768±.0027 | .9839±.0012 | .9790±.0028 | .9736±.0030 | .9872±.0009 | .9869±.0009 | .9868±.0007 |
| | PTDNet | .9843±.0039 | .9830±.0018 | .9745±.0076 | .9844±.0012 | .9787±.0039 | .9752±.0027 | .9872±.0005 | .9868±.0010 | .9872±.0007 |
| | Co-teaching | .9789±.0039 | .9777±.0086 | .9717±.0111 | .9703±.0050 | .9718±.0071 | .9782±.0049 | .9645±.0105 | .9730±.0267 | .9646±.0014 |
| | Peer loss | .9944±.0029 | .9630±.0105 | .9687±.0046 | .9610±.0068 | .9610±.0064 | .9783±.0073 | .9677±.0053 | .9638±.0050 | .9691±.0203 |
| | Jaccard | .9757±.0079 | .9863±.0087 | .9883±.0055 | .9607±.0182 | .9616±.0184 | .9655±.0151 | .9689±.0025 | .9625±.0009 | .9604±.0036 |
| | GIB | .9809±.0134 | .9747±.0077 | .9709±.0204 | .9721±.0297 | .9741±.0199 | .9740±.0111 | .9747±.0038 | .9768±.0093 | .9799±.0016 |
| | SupCon | .9709±.0156 | .9464±.0254 | .9134±.0309 | .9824±.0008 | .9745±.0062 | .9677±.0113 | .9600±.0154 | .9580±.0183 | .9482±.0174 |
| | GRACE | .7803±.0649 | .7947±.0600 | .7618±.1058 | .8259±.0669 | .7588±.0963 | .7260±.1331 | .8440±.0370 | .8448±.0337 | .8122±.0403 |
| | **RGIB-REP** | .9855±.0014 | .9839±.0016 | .9783±.0034 | .9838±.0019 | .9806±.0013 | .9765±.0037 | .9872±.0008 | .9871±.0008 | .9867±.0007 |
| | **RGIB-SSL** | .9751±.0033 | .9707±.0061 | .9602±.0043 | .9767±.0030 | .9683±.0053 | .9603±.0062 | .9755±.0028 | .9772±.0032 | .9762±.0047 |
| L=6 | Standard | .9805±.0025 | .9658±.0321 | .9577±.0314 | .9804±.0018 | .9738±.0044 | .9710±.0036 | .9854±.0015 | .9860±.0007 | .9867±.0012 |
| | DropEdge | .9643±.0041 | .9638±.0054 | .9500±.0274 | .9736±.0042 | .9589±.0089 | .9606±.0055 | .9735±.0008 | .9726±.0012 | .9736±.0009 |
| | NeuralSparse | .9746±.0077 | .9667±.0092 | .9560±.0122 | .9740±.0047 | .9668±.0056 | .9652±.0043 | .9735±.0017 | .9742±.0018 | .9733±.0030 |
| | PTDNet | .9769±.0037 | .9679±.0071 | .9582±.0296 | .9754±.0027 | .9688±.0045 | .9645±.0039 | .9738±.0041 | .9741±.0022 | .9756±.0012 |
| | Co-teaching | .9745±.0034 | .9686±.0316 | .9625±.0408 | .9747±.0084 | .9833±.0108 | .9738±.0035 | .9622±.0024 | .9676±.0072 | .9721±.0215 |
| | Peer loss | .9690±.0114 | .9728±.0334 | .9588±.0309 | .9716±.0023 | .9776±.0075 | .9766±.0074 | .9871±.0207 | .9605±.0202 | .9768±.0063 |
| | Jaccard | .9789±.0084 | .9651±.0360 | .9620±.0337 | .9673±.0007 | .9809±.0110 | .9731±.0021 | .9652±.0064 | .9607±.0004 | .9751±.0030 |
| | GIB | .9839±.0037 | .9777±.0420 | .9584±.0374 | .9868±.0290 | .9760±.0135 | .9791±.0249 | .9648±.0054 | .9640±.0046 | .9774±.0029 |
| | SupCon | .9428±.0194 | .9097±.0361 | .8779±.0347 | .9653±.0171 | .9459±.0378 | .9312±.0344 | .9465±.0173 | .9484±.0194 | .9023±.0333 |
| | GRACE | .7234±.0687 | .6604±.1398 | .6075±.1089 | .7415±.1242 | .6176±.1345 | .6216±.1341 | .7613±.1157 | .7256±.1351 | .7062±.1042 |
| | **RGIB-REP** | .9779±.0031 | .9711±.0042 | .9433±.0392 | .9753±.0041 | .9666±.0087 | .9614±.0057 | .9847±.0017 | .9848±.0013 | .9856±.0009 |
| | **RGIB-SSL** | .9653±.0107 | .9491±.0105 | .9254±.0206 | .9676±.0057 | .9502±.0087 | .9384±.0067 | .9737±.0065 | .9675±.0052 | .9713±.0103 |

Table 27: Full results on Facebook dataset with SAGE.

| layers | method | SAGE mixed noise 20% | 40% | 60% | input noise 20% | 40% | 60% | label noise 20% | 40% | 60% |
|---|---|---|---|---|---|---|---|---|---|---|
| L=2 | Standard | .9858±.0008 | .9827±.0008 | .9788±.0012 | .9881±.0008 | .9862±.0007 | .9862±.0008 | .9867±.0006 | .9840±.0006 | .9825±.0014 |
| | DropEdge | .9745±.0008 | .9807±.0011 | .9765±.0011 | .9766±.0007 | .9749±.0011 | .9752±.0008 | .9761±.0008 | .9739±.0010 | .9719±.0011 |
| | NeuralSparse | .9753±.0008 | .9619±.0011 | .9683±.0007 | .9675±.0008 | .9660±.0006 | .9659±.0011 | .9667±.0007 | .9640±.0007 | .9627±.0011 |
| | PTDNet | .9755±.0007 | .9723±.0009 | .9779±.0012 | .9775±.0007 | .9761±.0008 | .9754±.0007 | .9766±.0007 | .9743±.0007 | .9726±.0014 |
| | Co-teaching | .9759±.0101 | .9722±.0004 | .9728±.0100 | .9702±.0050 | .9742±.0023 | .9739±.0081 | .9764±.0125 | .9788±.0205 | .9700±.0282 |
| | Peer loss | .9765±.0052 | .9720±.0072 | .9790±.0103 | .9787±.0005 | .9781±.0105 | .9787±.0075 | .9721±.0019 | .9842±.0136 | .9793±.0031 |
| | Jaccard | .9747±.0018 | .9720±.0009 | .9721±.0096 | .9729±.0166 | .9760±.0139 | .9752±.0102 | .9733±.0053 | .9706±.0059 | .9716±.0098 |
| | GIB | .9798±.0089 | .9780±.0040 | .9765±.0020 | .9774±.0139 | .9767±.0267 | .9778±.0006 | .9780±.0012 | .9713±.0067 | .9772±.0050 |
| | SupCon | .9482±.0166 | .9288±.0046 | .9141±.0139 | .9793±.0043 | .9782±.0072 | .9786±.0078 | .9278±.0048 | .9252±.0074 | .9156±.0041 |
| | GRACE | .9113±.0116 | .8907±.0295 | .9077±.0089 | .9012±.0113 | .8885±.0177 | .9031±.0104 | .9053±.0126 | .9050±.0173 | .9067±.0137 |
| | **RGIB-REP** | .9860±.0009 | .9834±.0008 | .9802±.0014 | .9878±.0009 | .9863±.0007 | .9863±.0011 | .9875±.0006 | .9860±.0007 | .9855±.0011 |
| | **RGIB-SSL** | .9840±.0007 | .9829±.0010 | .9811±.0006 | .9838±.0007 | .9821±.0006 | .9823±.0004 | .9841±.0009 | .9839±.0007 | .9844±.0010 |
| L=4 | Standard | .9824±.0015 | .9783±.0025 | .9698±.0040 | .9849±.0015 | .9815±.0024 | .9815±.0025 | .9844±.0015 | .9817±.0007 | .9809±.0009 |
| | DropEdge | .9818±.0026 | .9764±.0034 | .9722±.0022 | .9625±.0015 | .9601±.0043 | .9600±.0023 | .9649±.0008 | .9627±.0017 | .9615±.0012 |
| | NeuralSparse | .9811±.0031 | .9776±.0021 | .9740±.0031 | .9636±.0023 | .9600±.0026 | .9615±.0016 | .9648±.0008 | .9628±.0014 | .9613±.0025 |
| | PTDNet | .9823±.0012 | .9757±.0041 | .9735±.0032 | .9623±.0041 | .9609±.0031 | .9607±.0022 | .9652±.0007 | .9632±.0009 | .9623±.0024 |
| | Co-teaching | .9660±.0009 | .9653±.0075 | .9659±.0114 | .9600±.0098 | .9634±.0091 | .9626±.0079 | .9654±.0253 | .9706±.0136 | .9628±.0263 |
| | Peer loss | .9668±.0105 | .9676±.0064 | .9716±.0030 | .9681±.0079 | .9641±.0044 | .9603±.0112 | .9671±.0204 | .9609±.0088 | .9660±.0032 |
| | Jaccard | .9621±.0019 | .9715±.0056 | .9772±.0068 | .9662±.0143 | .9675±.0094 | .9782±.0056 | .9725±.0107 | .9726±.0063 | .9705±9.276 |
| | GIB | .9774±0.000 | .9758±.0200 | .9732±.0025 | .9708±.0195 | .9626±.0286 | .9679±.0171 | .9765±.0101 | .9714±.0018 | .9710±.0107 |
| | SupCon | .9169±.0144 | .9096±.0191 | .9001±.0143 | .9324±.0208 | .9188±.0100 | .9185±.0192 | .9295±.0114 | .9328±.0158 | .9325±.0148 |
| | GRACE | .8683±.0292 | .8355±.0696 | .8545±.0415 | .8585±.0424 | .8721±.0261 | .8413±.0364 | .8108±.0213 | .8527±.0375 | .8376±.0330 |
| | **RGIB-REP** | .9828±.0016 | .9804±.0020 | .9750±.0024 | .9843±.0014 | .9811±.0030 | .9808±.0030 | .9857±.0004 | .9833±.0025 | .9830±.0019 |
| | **RGIB-SSL** | .9817±.0016 | .9797±.0014 | .9768±.0009 | .9828±.0018 | .9798±.0014 | .9797±.0013 | .9821±.0009 | .9820±.0011 | .9826±.0010 |
| L=6 | Standard | .9751±.0055 | .9603±.0365 | .9616±.0091 | .9715±.0077 | .9672±.0057 | .9517±.0228 | .9761±.0136 | .9788±.0045 | .9749±.0151 |
| | DropEdge | .9652±.0039 | .9676±.0088 | .9578±.0149 | .9734±.0050 | .9616±.0126 | .9665±.0073 | .9631±.0022 | .9606±.0046 | .9651±.0145 |
| | NeuralSparse | .9640±.0075 | .9577±.0222 | .9672±.0061 | .9647±.0048 | .9694±.0156 | .9673±.0036 | .9632±.0017 | .9618±.0037 | .9615±.0051 |
| | PTDNet | .9668±.0029 | .9633±.0142 | .9617±.0111 | .9774±.0028 | .9693±.0053 | .9642±.0055 | .9629±.0035 | .9736±.0010 | .9708±.0037 |
| | Co-teaching | .9644±.0116 | .9699±.0463 | .9658±.0182 | .9730±.0107 | .9738±.0076 | .9583±.0239 | .9742±.0138 | .9744±.0106 | .9720±.0202 |
| | Peer loss | .9644±.0141 | .9602±.0369 | .9681±.0085 | .9613±.0167 | .9728±.0053 | .9524±.0288 | .9655±.0220 | .9769±.0028 | .9650±.0256 |
| | Jaccard | .9750±.0119 | .9666±.0453 | .9686±.0094 | .9666±.0261 | .9612±.0056 | .9683±.0383 | .9634±.0170 | .9784±.0110 | .9781±.0178 |
| | GIB | .9739±.0197 | .9656±.0564 | .9749±.0125 | .9743±.0351 | .9717±.0034 | .9615±.0291 | .9640±.0153 | .9779±.0059 | .9743±.0222 |
| | SupCon | .9007±.0112 | .8859±.0189 | .8730±.0140 | .8997±.0142 | .8940±.0104 | .9006±.0232 | .9146±.0114 | .9166±.0143 | .9185±.0122 |
| | GRACE | .8202±.0365 | .7225±.0840 | .7901±.1065 | .8018±.0575 | .8224±.0547 | .7960±.0915 | .8061±.0367 | .8262±.0306 | .8103±.0351 |
| | **RGIB-REP** | .9770±.0040 | .9557±.0119 | .9574±.0149 | .9749±.0046 | .9683±.0072 | .9584±.0093 | .9838±.0020 | .9829±.0015 | .9814±.0029 |
| | **RGIB-SSL** | .9773±.0032 | .9720±.0031 | .9678±.0030 | .9781±.0027 | .9739±.0023 | .9692±.0050 | .9774±.0016 | .9786±.0012 | .9786±.0017 |

Table 28: Full results on Chameleon dataset with GCN.

| layers | method | mixed noise 20% | 40% | 60% | input noise 20% | 40% | 60% | label noise 20% | 40% | 60% |
|---|---|---|---|---|---|---|---|---|---|---|
| **GCN** | | | | | | | | | | |
| L=2 | Standard | .9753±.0025 | .9696±.0022 | .9657±.0029 | .9784±.0017 | .9762±.0016 | .9754±.0023 | .9775±.0018 | .9769±.0018 | .9755±.0036 |
| | DropEdge | .9716±.0030 | .9640±.0025 | .9579±.0029 | .9646±.0026 | .9602±.0024 | .9661±.0024 | .9686±.0018 | .9664±.0029 | .9651±.0031 |
| | NeuralSparse | .9741±.0032 | .9692±.0030 | .9633±.0030 | .9682±.0026 | .9646±.0023 | .9627±.0022 | .9679±.0017 | .9669±.0029 | .9651±.0039 |
| | PTDNet | .9748±.0026 | .9682±.0032 | .9637±.0031 | .9679±.0021 | .9649±.0019 | .9627±.0025 | .9690±.0019 | .9674±.0021 | .9648±.0033 |
| | Co-teaching | .9679±.0059 | .9641±.0086 | .9676±.0088 | .9767±.0096 | .9712±.0084 | .9709±.0106 | .9790±.0218 | .9729±.0015 | .9745±.0272 |
| | Peer loss | .9647±.0057 | .9662±.0106 | .9658±.0059 | .9781±.0043 | .9784±.0014 | .9759±.0095 | .9762±.0159 | .9695±.0090 | .9736±.0036 |
| | Jaccard | .9684±.0103 | .9657±.0118 | .9660±.0030 | .9611±.0190 | .9799±.0005 | .9774±.0128 | .9711±.0010 | .9754±.0091 | .9750±.0086 |
| | GIB | .9649±.0052 | .9711±.0050 | .9634±.0177 | .9709±.0106 | .9714±.0257 | .9786±.0012 | .9674±.0066 | .9660±.0070 | .9778±.0089 |
| | SupCon | .9625±.0024 | .9677±.0024 | .9613±.0032 | .9757±.0018 | .9718±.0023 | .9697±.0034 | .9749±.0013 | .9755±.0024 | .9750±.0030 |
| | GRACE | .9145±.0105 | .8978±.0110 | .8915±.0033 | .9081±.0121 | .9005±.0062 | .8931±.0042 | .9171±.0042 | .9184±.0054 | .9196±.0058 |
| | **RGIB-REP** | .9754±.0026 | .9698±.0026 | .9652±.0027 | .9797±.0023 | .9757±.0016 | .9744±.0028 | .9804±.0023 | .9798±.0016 | .9782±.0028 |
| | **RGIB-SSL** | .9719±.0023 | .9658±.0025 | .9595±.0026 | .9709±.0017 | .9664±.0024 | .9638±.0025 | .9786±.0016 | .9797±.0022 | .9789±.0034 |
| L=4 | Standard | .9616±.0033 | .9496±.0190 | .9274±.0276 | .9608±.0038 | .9433±.0261 | .9368±.0271 | .9686±.0020 | .9580±.0021 | .9362±.0035 |
| | DropEdge | .9568±.0044 | .9548±.0058 | .9407±.0050 | .9567±.0039 | .9433±.0055 | .9432±.0088 | .9580±.0020 | .9579±.0033 | .9578±.0033 |
| | NeuralSparse | .9599±.0026 | .9497±.0032 | .9402±.0057 | .9609±.0027 | .9540±.0178 | .9348±.0169 | .9583±.0017 | .9583±.0030 | .9571±.0031 |
| | PTDNet | .9607±.0030 | .9514±.0036 | .9424±.0097 | .9610±.0024 | .9457±.0194 | .9360±.0090 | .9585±.0022 | .9576±.0030 | .9665±.0035 |
| | Co-teaching | .9595±.0112 | .9516±.0199 | .9483±.0374 | .9524±.0123 | .9446±.0288 | .9447±.0286 | .9642±.0079 | .9650±.0058 | .9533±.0050 |
| | Peer loss | .9543±.0090 | .9533±.0192 | .9267±.0320 | .9558±.0037 | .9482±.0280 | .9412±.0269 | .9621±.0161 | .9501±.0055 | .9569±.0227 |
| | Jaccard | .9503±.0061 | .9538±.0223 | .9344±.0364 | .9507±.0056 | .9436±.0286 | .9364±.0385 | .9603±.0051 | .9659±.0020 | .9557±.0070 |
| | GIB | .9554±.0036 | .9561±.0292 | .9321±.0267 | .9605±.0069 | .9521±.0231 | .9416±.0512 | .9651±.0061 | .9582±.0100 | .9489±.0029 |
| | SupCon | .9561±.0046 | .9531±.0043 | .9467±.0045 | .9606±.0031 | .9536±.0031 | .9468±.0075 | .9584±.0015 | .9580±.0051 | .9477±.0038 |
| | GRACE | .8978±.0145 | .8987±.0050 | .8949±.0030 | .8994±.0141 | .9007±.0045 | .8964±.0043 | .9053±.0110 | .9074±.0136 | .9075±.0113 |
| | **RGIB-REP** | .9723±.0035 | .9621±.0036 | .9519±.0052 | .9705±.0027 | .9604±.0052 | .9480±.0053 | .9785±.0031 | .9797±.0017 | .9785±.0032 |
| | **RGIB-SSL** | .9655±.0022 | .9592±.0043 | .9500±.0053 | .9658±.0023 | .9570±.0047 | .9486±.0029 | .9730±.0017 | .9752±.0030 | .9744±.0032 |
| L=6 | Standard | .9662±.0042 | .9511±.0079 | .9286±.0067 | .9656±.0045 | .9450±.0177 | .9276±.0229 | .9752±.0027 | .9766±.0035 | .9745±.0040 |
| | DropEdge | .9601±.0032 | .9484±.0053 | .9331±.0066 | .9608±.0020 | .9481±.0129 | .9331±.0073 | .9537±.0022 | .9559±.0027 | .9547±.0024 |
| | NeuralSparse | .9644±.0035 | .9550±.0033 | .9411±.0046 | .9538±.0035 | .9468±.0038 | .9387±.0053 | .9525±.0033 | .9541±.0031 | .9535±.0037 |
| | PTDNet | .9655±.0043 | .9585±.0033 | .9384±.0088 | .9657±.0024 | .9553±.0051 | .9391±.0044 | .9537±.0037 | .9559±.0025 | .9543±.0032 |
| | Co-teaching | .9654±.0058 | .9582±.0139 | .9319±.0163 | .9505±.0121 | .9534±.0222 | .9271±.0308 | .9679±.0198 | .9557±.0112 | .9557±.0282 |
| | Peer loss | .9658±.0120 | .9551±.0174 | .9308±.0099 | .9552±.0040 | .9441±.0226 | .9358±.0328 | .9575±.0164 | .9667±.0230 | .9593±.0236 |
| | Jaccard | .9754±.0063 | .9549±.0088 | .9292±.0104 | .9637±.0035 | .9495±.0213 | .9457±.0350 | .9500±.0040 | .9550±.0128 | .9433±.0066 |
| | GIB | .9800±.0203 | .9621±.0246 | .9329±.0238 | .9544±.0212 | .9542±.0395 | .9456±.0219 | .9619±.0072 | .9505±.0112 | .9536±.0124 |
| | SupCon | .9537±.0054 | .9452±.0073 | .9352±.0046 | .9534±.0068 | .9454±.0068 | .9375±.0023 | .9618±.0036 | .9627±.0032 | .9557±.0067 |
| | GRACE | .9024±.0098 | .8995±.0030 | .8936±.0029 | .9025±.0097 | .8983±.0055 | .8952±.0042 | .9154±.0092 | .9166±.0092 | .9151±.0082 |
| | **RGIB-REP** | .9666±.0028 | .9580±.0074 | .9467±.0067 | .9662±.0027 | .9606±.0041 | .9453±.0049 | .9740±.0033 | .9776±.0021 | .9753±.0031 |
| | **RGIB-SSL** | .9633±.0032 | .9416±.0036 | .9177±.0082 | .9627±.0037 | .9482±.0037 | .9276±.0093 | .9712±.0017 | .9742±.0025 | .9726±.0032 |

Table 29: Full results on Chameleon dataset with GAT.

| layers | method | mixed noise 20% | 40% | 60% | input noise 20% | 40% | 60% | label noise 20% | 40% | 60% |
|---|---|---|---|---|---|---|---|---|---|---|
| **GAT** | | | | | | | | | | |
| L=2 | Standard | .9725±.0027 | .9650±.0018 | .9625±.0018 | .9767±.0026 | .9747±.0020 | .9759±.0018 | .9746±.0023 | .9743±.0017 | .9711±.0041 |
| | DropEdge | .9716±.0023 | .9630±.0036 | .9563±.0023 | .9747±.0026 | .9717±.0030 | .9708±.0025 | .9752±.0018 | .9726±.0026 | .9698±.0035 |
| | NeuralSparse | .9733±.0029 | .9667±.0028 | .9615±.0028 | .9762±.0028 | .9748±.0028 | .9741±.0015 | .9762±.0022 | .9742±.0033 | .9704±.0042 |
| | PTDNet | .9736±.0028 | .9657±.0024 | .9620±.0027 | .9768±.0023 | .9735±.0021 | .9739±.0022 | .9670±.0021 | .9631±.0028 | .9611±.0058 |
| | Co-teaching | .9617±.0071 | .9629±.0047 | .9626±.0051 | .9624±.0084 | .9645±.0027 | .9647±.0104 | .9684±.0279 | .9610±.0139 | .9688±.0092 |
| | Peer loss | .9675±.0064 | .9646±.0101 | .9683±.0037 | .9602±.0046 | .9621±.0112 | .9611±.0078 | .9638±.0045 | .9626±.0074 | .9620±.0089 |
| | Jaccard | .9662±.0063 | .9665±.0062 | .9697±.0073 | .9652±.0034 | .9649±.0051 | .9616±.0123 | .9608±.0076 | .9612±.0047 | .9603±.0134 |
| | GIB | .9628±.0018 | .9639±.0115 | .9625±.0208 | .9681±.0091 | .9643±.0257 | .9601±.0285 | .9644±.0119 | .9688±.0096 | .9631±.0113 |
| | SupCon | .9709±.0059 | .9611±.0195 | .9607±.0032 | .9752±.0011 | .9721±.0026 | .9706±.0022 | .9692±.0035 | .9701±.0030 | .9689±.0057 |
| | GRACE | .7908±.0348 | .7829±.0494 | .7598±.0693 | .7992±.0371 | .7960±.0371 | .7913±.0418 | .7905±.0261 | .7985±.0322 | .7894±.0340 |
| | **RGIB-REP** | .9736±.0031 | .9668±.0021 | .9629±.0026 | .9769±.0018 | .9752±.0016 | .9752±.0018 | .9766±.0021 | .9753±.0029 | .9732±.0041 |
| | **RGIB-SSL** | .9635±.0037 | .9593±.0037 | .9536±.0041 | .9640±.0062 | .9587±.0048 | .9545±.0035 | .9709±.0015 | .9728±.0028 | .9719±.0046 |
| L=4 | Standard | .9721±.0035 | .9652±.0023 | .9605±.0031 | .9741±.0028 | .9686±.0030 | .9674±.0027 | .9740±.0037 | .9738±.0027 | .9712±.0047 |
| | DropEdge | .9666±.0031 | .9572±.0034 | .9469±.0043 | .9659±.0028 | .9598±.0024 | .9496±.0040 | .9749±.0024 | .9747±.0016 | .9718±.0031 |
| | NeuralSparse | .9687±.0027 | .9603±.0026 | .9536±.0033 | .9699±.0025 | .9600±.0036 | .9502±.0037 | .9748±.0022 | .9756±.0022 | .9727±.0035 |
| | PTDNet | .9707±.0041 | .9640±.0031 | .9549±.0034 | .9708±.0027 | .9602±.0025 | .9548±.0058 | .9754±.0025 | .9761±.0021 | .9709±.0041 |
| | Co-teaching | .9726±.0096 | .9648±.0030 | .9639±.0088 | .9551±.0037 | .9503±.0027 | .9566±.0028 | .9691±.0039 | .9664±.7565 | .9622±.0242 |
| | Peer loss | .9745±.0092 | .9714±.0088 | .9668±.0103 | .9637±.0041 | .9602±.0088 | .9658±.0064 | .9549±.0073 | .9531±.0096 | .9587±.0068 |
| | Jaccard | .9776±.0089 | .9692±.0098 | .9663±.0112 | .9729±.0024 | .9677±.0214 | .9744±.0012 | .9587±.0078 | .9519±.0092 | .9573±.0137 |
| | GIB | .9600±.0148 | .9646±.0041 | .9641±.0089 | .9639±.0197 | .9642±.0189 | .9636±.0047 | .9646±.0044 | .9624±.0103 | .9609±.0064 |
| | SupCon | .9644±.0033 | .9537±.0106 | .9381±.0125 | .9675±.0028 | .9631±.0042 | .9555±.0068 | .9594±.0131 | .9442±.0136 | .9386±.0102 |
| | GRACE | .5429±.0465 | .6085±.1132 | .6273±.1475 | .6178±.0940 | .5839±.1022 | .6513±.1544 | .5534±.0439 | .6195±.1055 | .6333±.0818 |
| | **RGIB-REP** | .9704±.0035 | .9645±.0027 | .9591±.0036 | .9713±.0023 | .9654±.0046 | .9572±.0047 | .9741±.0040 | .9745±.0028 | .9718±.0039 |
| | **RGIB-SSL** | .9507±.0034 | .9432±.0034 | .9389±.0059 | .9526±.0037 | .9451±.0045 | .9360±.0083 | .9619±.0037 | .9645±.0042 | .9643±.0038 |
| L=6 | Standard | .9659±.0029 | .9573±.0036 | .9482±.0054 | .9644±.0033 | .9543±.0075 | .9474±.0074 | .9722±.0043 | .9688±.0055 | .9698±.0065 |
| | DropEdge | .9461±.0082 | .9295±.0063 | .9062±.0218 | .9469±.0059 | .9293±.0087 | .9182±.0063 | .9698±.0036 | .9689±.0046 | .9694±.0037 |
| | NeuralSparse | .9563±.0079 | .9391±.0057 | .9185±.0157 | .9473±.0149 | .9338±.0043 | .9250±.0037 | .9688±.0037 | .9705±.0026 | .9696±.0063 |
| | PTDNet | .9508±.0075 | .9359±.0222 | .9288±.0153 | .9535±.0082 | .9424±.0053 | .9332±.0043 | .9723±.0033 | .9717±.0036 | .9697±.0046 |
| | Co-teaching | .9699±.0033 | .9627±.0129 | .9501±.0141 | .9709±.0129 | .9569±.0115 | .9482±.0155 | .9673±.0285 | .9698±.0260 | .9631±.0176 |
| | Peer loss | .9736±.0116 | .9570±.0133 | .9485±.0076 | .9659±.0060 | .9555±.0157 | .9469±.0086 | .9704±.0136 | .9701±.0196 | .9682±.0078 |
| | Jaccard | .9680±.0057 | .9607±.0055 | .9511±.0106 | .9815±.0185 | .9563±.0145 | .9468±.0100 | .9750±.0124 | .9732±.0093 | .9772±.0120 |
| | GIB | .9652±.0138 | .9666±.0113 | .9612±.0165 | .9776±.0272 | .9610±.0322 | .9624±.0365 | .9587±.0079 | .9514±.0078 | .9585±.0138 |
| | SupCon | .9481±.0185 | .9310±.0171 | .9015±.0169 | .9455±.0267 | .9490±.0157 | .9194±.0239 | .9413±.0206 | .9370±.0174 | .9241±.0144 |
| | GRACE | .5681±.0785 | .5107±.0287 | .5147±.0361 | .5599±.0776 | .5046±.0124 | .5167±.0502 | .5850±.0827 | .5864±.0795 | .5518±.0404 |
| | **RGIB-REP** | .9612±.0042 | .9410±.0069 | .9258±.0169 | .9531±.0054 | .9417±.0058 | .9313±.0092 | .9716±.0049 | .9747±.0036 | .9707±.0056 |
| | **RGIB-SSL** | .9367±.0058 | .9233±.0046 | .9199±.0053 | .9418±.0085 | .9236±.0056 | .9205±.0050 | .9576±.0066 | .9562±.0079 | .9557±.0083 |

Table 30: Full results on Chameleon dataset with SAGE.

| | | SAGE | | | | | | | | |
|---|---|---|---|---|---|---|---|---|---|---|
| layers | method | mixed noise | | | input noise | | | label noise | | |
| | | 20% | 40% | 60% | 20% | 40% | 60% | 20% | 40% | 60% |
| L=2 | Standard | .9683±.0036 | .9605±.0038 | .9536±.0028 | .9754±.0019 | .9734±.0015 | .9762±.0018 | .9700±.0019 | .9670±.0026 | .9603±.0049 |
| | DropEdge | .9604±.0025 | .9614±.0028 | .9556±.0022 | .9677±.0022 | .9659±.0030 | .9673±.0026 | .9630±.0016 | .9678±.0033 | .9625±.0047 |
| | NeuralSparse | .9669±.0031 | .9584±.0029 | .9550±.0029 | .9751±.0025 | .9735±.0022 | .9745±.0021 | .9705±.0014 | .9661±.0031 | .9606±.0051 |
| | PTDNet | .9674±.0035 | .9594±.0028 | .9538±.0019 | .9752±.0022 | .9637±.0033 | .9651±.0029 | .9603±.0016 | .9658±.0036 | .9598±.0042 |
| | Co-teaching | .9581±.0040 | .9549±.0110 | .9543±.0093 | .9453±.0044 | .9425±.0083 | .9469±.0089 | .9585±.0006 | .9595±.0039 | .9525±.0309 |
| | Peer loss | .9638±.0096 | .9581±.0098 | .9512±.0021 | .9541±.0091 | .9561±.0075 | .9557±.0104 | .9625±.0146 | .9506±.0155 | .9587±.0227 |
| | Jaccard | .9672±.0074 | .9650±.0123 | .9589±.0056 | .9503±.0007 | .9540±.0190 | .9517±.0015 | .9592±.0010 | .9522±.0027 | .9545±.0118 |
| | GIB | .9650±.0066 | .9593±.0214 | .9530±.0021 | .9671±.0007 | .9606±.0.000 | .9639±.0275 | .9537±.0055 | .9523±.0048 | .9586±.0068 |
| | SupCon | .9636±.0031 | .9656±.0017 | .9599±.0041 | .9506±.0022 | .9585±.0020 | .9593±.0019 | .9567±.0024 | .9530±.0027 | .9516±.0037 |
| | GRACE | .8215±.0332 | .8327±.0178 | .8372±.0245 | .8369±.0146 | .8251±.0441 | .8295±.0422 | .8610±.0055 | .8583±.0114 | .8641±.0137 |
| | **RGIB-REP** | .9699±.0028 | .9626±.0031 | .9577±.0034 | .9765±.0022 | .9741±.0018 | .9770±.0014 | .9733±.0017 | .9715±.0025 | .9670±.0037 |
| | **RGIB-SSL** | .9740±.0030 | .9709±.0027 | .9686±.0025 | .9752±.0020 | .9737±.0031 | .9741±.0035 | .9789±.0016 | .9791±.0028 | .9781±.0029 |
| L=4 | Standard | .9645±.0035 | .9580±.0035 | .9505±.0010 | .9721±.0017 | .9721±.0021 | .9721±.0016 | .9678±.0026 | .9641±.0028 | .9605±.0051 |
| | DropEdge | .9674±.0041 | .9585±.0026 | .9516±.0024 | .9652±.0029 | .9638±.0039 | .9652±.0019 | .9602±.0021 | .9663±.0030 | .9614±.0038 |
| | NeuralSparse | .9649±.0022 | .9576±.0035 | .9495±.0041 | .9619±.0032 | .9602±.0022 | .9620±.0031 | .9675±.0017 | .9642±.0031 | .9603±.0046 |
| | PTDNet | .9651±.0036 | .9574±.0034 | .9506±.0037 | .9624±.0027 | .9622±.0015 | .9630±.0035 | .9687±.0024 | .9646±.0025 | .9596±.0056 |
| | Co-teaching | .9722±.0090 | .9665±.0040 | .9582±.0007 | .9683±.0083 | .9633±.0101 | .9613±.0067 | .9617±.0255 | .9638±.0070 | .9637±.0066 |
| | Peer loss | .9543±.0099 | .9663±.0044 | .9496±.0096 | .9610±.0110 | .9667±.0086 | .9601±.0060 | .9657±.0035 | .9679±.0026 | .9661±.0180 |
| | Jaccard | .9438±.0076 | .9485±.0047 | .9444±.0069 | .9653±.0115 | .9683±.0218 | .9697±.0157 | .9696±.0043 | .9537±.0048 | .9685±.0088 |
| | GIB | .9557±.0177 | .9574±.0043 | .9538±.0117 | .9644±.0214 | .9614±.0204 | .9559±.0038 | .9576±.0026 | .9464±.0080 | .9303±.0065 |
| | SupCon | .9494±.0130 | .9351±.0093 | .9263±.0065 | .9592±.0122 | .9538±.0122 | .9512±.0125 | .9478±.0099 | .9501±.0126 | .9449±.0099 |
| | GRACE | .8040±.0364 | .8542±.0415 | .8266±.0443 | .8362±.0272 | .8251±.0484 | .8642±.0383 | .7462±.0412 | .7970±.0306 | .7892±.0341 |
| | **RGIB-REP** | .9665±.0031 | .9582±.0035 | .9520±.0035 | .9740±.0018 | .9719±.0028 | .9722±.0022 | .9690±.0021 | .9696±.0023 | .9671±.0058 |
| | **RGIB-SSL** | .9725±.0027 | .9682±.0028 | .9655±.0049 | .9740±.0021 | .9707±.0040 | .9721±.0044 | .9748±.0023 | .9762±.0031 | .9754±.0040 |
| L=6 | Standard | .9606±.0046 | .9541±.0058 | .9470±.0035 | .9679±.0031 | .9658±.0041 | .9640±.0059 | .9673±.0026 | .9641±.0038 | .9625±.0042 |
| | DropEdge | .9532±.0028 | .9540±.0024 | .9487±.0043 | .9592±.0028 | .9582±.0035 | .9580±.0299 | .9500±.0014 | .9561±.0030 | .9537±.0024 |
| | NeuralSparse | .9525±.0043 | .9383±.0365 | .9480±.0036 | .9597±.0027 | .9569±.0037 | .9530±.0057 | .9589±.0024 | .9552±.0025 | .9515±.0034 |
| | PTDNet | .9536±.0042 | .9557±.0038 | .9483±.0047 | .9706±.0035 | .9581±.0040 | .9459±.0074 | .9580±.0024 | .9559±.0021 | .9429±.0045 |
| | Co-teaching | .9503±.0070 | .9466±.0113 | .9355±.0039 | .9492±.0031 | .9457±.0059 | .9403±.0118 | .9445±.0009 | .9465±.0055 | .9497±.0115 |
| | Peer loss | .9553±.0064 | .9409±.0086 | .9330±.0076 | .9697±.0084 | .9656±.0110 | .9652±.0066 | .9521±.0210 | .9595±.0195 | .9589±.0047 |
| | Jaccard | .9416±.0113 | .9535±.0116 | .9535±.0111 | .9610±.0042 | .9568±.0210 | .9463±.0085 | .9518±.0057 | .9421±.0032 | .9433±.0137 |
| | GIB | .9609±.0173 | .9579±.0050 | .9568±.0070 | .9691±.0315 | .9698±.0037 | .9592±.0279 | .9433±.0065 | .9366±.0074 | .9347±.0115 |
| | SupCon | .9310±.0107 | .9134±.0087 | .9178±.0101 | .9291±.0114 | .9244±.0271 | .9181±.0281 | .9342±.0132 | .9373±.0116 | .9313±.0087 |
| | GRACE | .7479±.0742 | .7756±.0708 | .7718±.0880 | .7275±.0906 | .7914±.0755 | .8109±.0655 | .7754±.0258 | .7629±.0508 | .7686±.0437 |
| | **RGIB-REP** | .9636±.0022 | .9558±.0032 | .9455±.0038 | .9703±.0022 | .9641±.0075 | .9607±.0050 | .9683±.0013 | .9681±.0033 | .9664±.0042 |
| | **RGIB-SSL** | .9665±.0030 | .9601±.0045 | .9563±.0036 | .9662±.0052 | .9654±.0040 | .9610±.0032 | .9699±.0022 | .9716±.0033 | .9698±.0031 |

Table 31: Full results on Squirrel dataset with GCN.

| | | GCN | | | | | | | | |
|---|---|---|---|---|---|---|---|---|---|---|
| layers | method | mixed noise | | | input noise | | | label noise | | |
| | | 20% | 40% | 60% | 20% | 40% | 60% | 20% | 40% | 60% |
| L=2 | Standard | .9621±.0018 | .9519±.0020 | .9444±.0024 | .9610±.0028 | .9490±.0031 | .9401±.0036 | .9744±.0013 | .9731±.0010 | .9722±.0011 |
| | DropEdge | .9571±.0015 | .9480±.0032 | .9370±.0028 | .9587±.0011 | .9486±.0026 | .9349±.0026 | .9623±.0015 | .9612±.0008 | .9604±.0014 |
| | NeuralSparse | .9611±.0019 | .9528±.0027 | .9409±.0023 | .9621±.0017 | .9495±.0016 | .9364±.0022 | .9741±.0013 | .9724±.0018 | .9722±.0010 |
| | PTDNet | .9612±.0018 | .9513±.0026 | .9424±.0024 | .9619±.0016 | .9488±.0035 | .9363±.0022 | .9744±.0011 | .9723±.0015 | .9718±.0009 |
| | Co-teaching | .9701±.0076 | .9535±.0044 | .9465±.0059 | .9599±.0097 | .9525±.0120 | .9454±.0109 | .9629±.0047 | .9666±.0304 | .9631±.0004 |
| | Peer loss | .9610±.0015 | .9608±.0057 | .9451±.0092 | .9636±.0107 | .9491±.0125 | .9436±.0091 | .9694±.0012 | .9628±.0134 | .9676±.0149 |
| | Jaccard | .9613±.0027 | .9568±.0046 | .9509±.0014 | .9622±.0159 | .9594±.0111 | .9397±.0119 | .9528±.0017 | .9405±.0089 | .9401±.0010 |
| | GIB | .9522±.0195 | .9478±.0094 | .9489±.0159 | .9494±.0260 | .9461±.0091 | .9432±.0068 | .9457±.0099 | .9462±.0017 | .9492±.0060 |
| | SupCon | .9494±.0013 | .9439±.0009 | .9458±.0024 | .9421±.0008 | .9549±.0013 | .9489±.0030 | .9444±.0011 | .9423±.0015 | .9318±.0009 |
| | GRACE | .9292±.0011 | .9316±.0010 | .9330±.0013 | .9290±.0012 | .9317±.0010 | .9327±.0016 | .9054±.0018 | .9058±.0015 | .9052±.0018 |
| | **RGIB-REP** | .9614±.0019 | .9554±.0026 | .9471±.0035 | .9611±.0021 | .9512±.0032 | .9392±.0043 | .9761±.0015 | .9753±.0011 | .9745±.0015 |
| | **RGIB-SSL** | .9571±.0019 | .9476±.0029 | .9402±.0020 | .9534±.0016 | .9486±.0027 | .9451±.0031 | .9752±.0023 | .9749±.0019 | .9738±.0023 |
| L=4 | Standard | .9432±.0036 | .9406±.0031 | .9386±.0025 | .9416±.0042 | .9395±.0011 | .9411±.0040 | .9720±.0016 | .9720±.0016 | .9710±.0021 |
| | DropEdge | .9439±.0063 | .9377±.0024 | .9365±.0045 | .9426±.0042 | .9376±.0033 | .9358±.0023 | .9608±.0008 | .9603±.0013 | .9698±.0016 |
| | NeuralSparse | .9494±.0065 | .9309±.0028 | .9297±.0033 | .9469±.0047 | .9403±.0028 | .9417±.0062 | .9633±.0009 | .9626±.0016 | .9625±.0016 |
| | PTDNet | .9485±.0056 | .9326±.0046 | .9304±.0048 | .9469±.0048 | .9400±.0030 | .9379±.0026 | .9633±.0018 | .9623±.0012 | .9626±.0012 |
| | Co-teaching | .9461±.0099 | .9352±.0064 | .9374±.0051 | .9462±.0101 | .9425±.0032 | .9306±.0053 | .9675±.0292 | .9641±.0115 | .9655±.0264 |
| | Peer loss | .9457±.0117 | .9345±.0117 | .9286±.0044 | .9362±.0038 | .9386±.0058 | .9336±.0043 | .9636±.0012 | .9694±.0033 | .9696±.0006 |
| | Jaccard | .9443±.0047 | .9327±.0079 | .9244±.0054 | .9388±.0179 | .9345±.0146 | .9240±.0055 | .9529±.0009 | .9512±.0089 | .9501±.0120 |
| | GIB | .9472±.0127 | .9329±.0155 | .9302±.0202 | .9390±.0045 | .9406±.0196 | .9397±.0249 | .9641±.0025 | .9628±.0032 | .9601±.0042 |
| | SupCon | .9473±.0010 | .9348±.0017 | .9301±.0029 | .9372±.0010 | .9343±.0037 | .9305±.0033 | .9516±.0042 | .9595±.0011 | .9511±.0040 |
| | GRACE | .9394±.0008 | .9380±.0007 | .9363±.0016 | .9392±.0011 | .9378±.0008 | .9363±.0017 | .9171±.0019 | .9174±.0014 | .9166±.0013 |
| | **RGIB-REP** | .9509±.0046 | .9455±.0071 | .9434±.0059 | .9495±.0053 | .9432±.0059 | .9405±.0050 | .9735±.0013 | .9733±.0015 | .9737±.0009 |
| | **RGIB-SSL** | .9499±.0040 | .9426±.0019 | .9425±.0021 | .9479±.0051 | .9429±.0017 | .9429±.0024 | .9727±.0015 | .9729±.0019 | .9726±.0011 |
| L=6 | Standard | .9484±.0049 | .9429±.0038 | .9408±.0039 | .9489±.0057 | .9408±.0021 | .9386±.0022 | .9688±.0028 | .9675±.0027 | .9656±.0034 |
| | DropEdge | .9460±.0063 | .9375±.0028 | .9349±.0017 | .9474±.0057 | .9409±.0030 | .9365±.0022 | .9669±.0022 | .9652±.0031 | .9638±.0038 |
| | NeuralSparse | .9508±.0049 | .9429±.0047 | .9391±.0029 | .9532±.0060 | .9412±.0044 | .9388±.0036 | .9681±.0020 | .9647±.0018 | .9671±.0019 |
| | PTDNet | .9535±.0047 | .9451±.0050 | .9395±.0033 | .9514±.0041 | .9432±.0068 | .9380±.0020 | .9690±.0023 | .9667±.0035 | .9649±.0025 |
| | Co-teaching | .9480±.0144 | .9454±.0074 | .9463±.0076 | .9538±.0051 | .9424±.0048 | .9379±.0012 | .9672±.0179 | .9694±.0283 | .9654±.0195 |
| | Peer loss | .9539±.0051 | .9420±.0127 | .9405±.0060 | .9445±.0139 | .9446±.0012 | .9462±.0081 | .9684±.0182 | .9639±.0188 | .9624±.0189 |
| | Jaccard | .9519±.0138 | .9459±.0100 | .9398±.0115 | .9483±.0070 | .9429±.0203 | .9428±.0105 | .9697±.0074 | .9687±.0062 | .9673±.0077 |
| | GIB | .9447±.0103 | .9459±.0026 | .9410±.0108 | .9431±.0289 | .9413±.0301 | .9386±.0238 | .9633±.0041 | .9619±.0114 | .9612±.0064 |
| | SupCon | .9590±.0023 | .9507±.0019 | .9463±.0028 | .9580±.0024 | .9519±.0018 | .9464±.0026 | .9638±.0051 | .9524±.0048 | .9419±.0012 |
| | GRACE | .9399±.0007 | .9372±.0006 | .9348±.0017 | .9397±.0012 | .9370±.0009 | .9349±.0017 | .9213±.0017 | .9218±.0013 | .9211±.0014 |
| | **RGIB-REP** | .9556±.0028 | .9492±.0060 | .9463±.0063 | .9570±.0018 | .9496±.0060 | .9420±.0056 | .9677±.0029 | .9676±.0031 | .9662±.0018 |
| | **RGIB-SSL** | .9462±.0029 | .9471±.0020 | .9448±.0032 | .9509±.0032 | .9474±.0017 | .9449±.0030 | .9642±.0012 | .9659±.0027 | .9647±.0010 |

Table 32: Full results on Squirrel dataset with GAT.

| layers | method | mixed noise 20% | 40% | 60% | input noise 20% | 40% | 60% | label noise 20% | 40% | 60% |
|---|---|---|---|---|---|---|---|---|---|---|
| | | GAT | | | | | | | | |
| L=2 | Standard | .9680±.0007 | .9635±.0017 | .9588±.0025 | .9702±.0008 | .9690±.0010 | .9659±.0014 | .9719±.0018 | .9701±.0017 | .9686±.0012 |
| | DropEdge | .9619±.0014 | .9567±.0020 | .9505±.0022 | .9627±.0008 | .9593±.0014 | .9551±.0012 | .9684±.0015 | .9662±.0026 | .9661±.0018 |
| | NeuralSparse | .9656±.0009 | .9602±.0012 | .9533±.0019 | .9662±.0011 | .9629±.0010 | .9584±.0017 | .9709±.0012 | .9692±.0014 | .9685±.0009 |
| | PTDNet | .9655±.0011 | .9602±.0012 | .9546±.0024 | .9668±.0010 | .9630±.0015 | .9603±.0021 | .9610±.0015 | .9698±.0012 | .9689±.0016 |
| | Co-teaching | .9566±.0034 | .9562±.0066 | .9516±.0026 | .9505±.0045 | .9509±.0092 | .9509±.0046 | .9591±.0293 | .9880±.0174 | .9360±.0089 |
| | Peer loss | .9720±.0097 | .9726±.0039 | .9682±.0106 | .9706±6.649 | .9715±.0064 | .9704±.0107 | .9681±.0121 | .9635±.0069 | .9683±.0116 |
| | Jaccard | .9715±.0098 | .9651±.0053 | .9648±.0094 | .9807±.0202 | .9768±.0041 | .9645±.0174 | .9637±.0081 | .9668±.0104 | .9627±.0002 |
| | GIB | .9727±.0019 | .9820±.0058 | .9610±.0069 | .9780±0.001 | .9965±0.000 | .9838±.0123 | .9660±.0067 | .9611±.0022 | .9679±.0061 |
| | SupCon | .9471±.0019 | .9386±.0022 | .9353±.0026 | .9539±.0032 | .9463±.0026 | .9411±.0034 | .9629±.0028 | .9631±.0019 | .9636±.0021 |
| | GRACE | .7300±.1193 | .7583±.1135 | .7015±.1383 | .7494±.1101 | .6903±.1807 | .7032±.1659 | .8109±.0382 | .7780±.1044 | .7608±.0620 |
| | **RGIB-REP** | .9675±.0014 | .9631±.0012 | .9587±.0023 | .9692±.0007 | .9660±.0013 | .9632±.0016 | .9735±.0012 | .9719±.0013 | .9715±.0010 |
| | **RGIB-SSL** | .9453±.0020 | .9421±.0017 | .9385±.0022 | .9418±.0024 | .9394±.0033 | .9362±.0023 | .9726±.0007 | .9705±.0037 | .9695±.0024 |
| L=4 | Standard | .9581±.0046 | .9436±.0063 | .9335±.0062 | .9592±.0047 | .9455±.0075 | .9415±.0061 | .9682±.0030 | .9690±.0028 | .9686±.0021 |
| | DropEdge | .9501±.0041 | .9340±.0046 | .9234±.0059 | .9476±.0054 | .9354±.0067 | .9292±.0039 | .9529±.0047 | .9513±.0054 | .9510±.0037 |
| | NeuralSparse | .9524±.0041 | .9383±.0046 | .9320±.0044 | .9401±.0071 | .9377±.0045 | .9334±.0041 | .9559±.0031 | .9555±.0032 | .9556±.0022 |
| | PTDNet | .9416±.0054 | .9310±.0059 | .9302±.0032 | .9447±.0031 | .9393±.0053 | .9305±.0131 | .9574±.0039 | .9557±.0025 | .9552±.0042 |
| | Co-teaching | .9324±.0109 | .9024±.0120 | .9029±.0126 | .9421±.0074 | .9345±.0126 | .9371±.0091 | .9576±.0124 | .9592±.0064 | .9551±.0087 |
| | Peer loss | .9520±.0071 | .9506±.0145 | .9370±.0140 | .9560±.0063 | .9451±.0086 | .9448±.0142 | .9548±.0128 | .9500±.0088 | .9593±.0134 |
| | Jaccard | .9561±.0133 | .9423±.0160 | .9396±.0117 | .9578±.0185 | .9583±.0256 | .9558±.0050 | .9572±.0074 | .9572±.0083 | .9554±.0089 |
| | GIB | .9353±.0080 | .9271±.0093 | .9098±.0079 | .9545±.0156 | .9504±.0240 | .9530±.0348 | .9544±.0023 | .9549±.0073 | .9524±.0060 |
| | SupCon | .9397±.0044 | .9322±.0038 | .9266±.0047 | .9365±.0039 | .9340±.0021 | .9314±.0029 | .9574±.0019 | .9557±.0065 | .9556±.0049 |
| | GRACE | .5710±.1110 | .5658±.1143 | .6203±.1324 | .5694±.1202 | .5771±.0945 | .6021±.1203 | .5291±.0694 | .5097±.0139 | .5029±.0072 |
| | **RGIB-REP** | .9541±.0059 | .9432±.0059 | .9316±.0063 | .9526±.0037 | .9401±.0072 | .9389±.0056 | .9681±.0028 | .9672±.0029 | .9660±.0048 |
| | **RGIB-SSL** | .9347±.0038 | .9321±.0031 | .9260±.0032 | .9321±.0022 | .9327±.0020 | .9270±.0028 | .9592±.0031 | .9651±.0014 | .9641±.0021 |
| L=6 | Standard | .9507±.0050 | .9309±.0164 | .9254±.0089 | .9487±.0065 | .9419±.0041 | .9255±.0073 | .9585±.0097 | .9520±.0070 | .9507±.0162 |
| | DropEdge | .9347±.0049 | .9279±.0118 | .9181±.0078 | .9368±.0082 | .9262±.0076 | .9175±.0127 | .9412±.0092 | .9431±.0076 | .9444±.0090 |
| | NeuralSparse | .9399±.0088 | .9279±.0167 | .9224±.0051 | .9423±.0063 | .9342±.0023 | .9213±.0115 | .9468±.0118 | .9474±.0102 | .9504±.0051 |
| | PTDNet | .9433±.0105 | .9348±.0072 | .9213±.0100 | .9460±.0048 | .9340±.0052 | .9273±.0094 | .9482±.0088 | .9541±.0092 | .9453±.0077 |
| | Co-teaching | .9219±.0059 | .9329±.0231 | .9229±.0185 | .9392±.0093 | .9352±.0098 | .9261±.0100 | .9496±.0383 | .9445±.0272 | .9490±.0284 |
| | Peer loss | .9300±.0136 | .9316±.0220 | .9349±.0147 | .9271±.0119 | .9170±.0117 | .9057±.0127 | .9230±.0221 | .9341±.0113 | .9187±.0325 |
| | Jaccard | .9577±.0112 | .9374±.0225 | .9297±.0158 | .9469±.0111 | .9406±.0111 | .9309±.0168 | .9393±.0153 | .9390±.0151 | .9314±.0223 |
| | GIB | .9504±.0239 | .9475±.0221 | .9445±.0127 | .9484±.0211 | .9487±.0139 | .9438±.0307 | .9421±.0133 | .9484±.0101 | .9486±.0206 |
| | SupCon | .9269±.0089 | .9174±.0092 | .9107±.0052 | .9278±.0057 | .9223±.0054 | .9210±.0038 | .9140±.0072 | .9173±.0085 | .9182±.0081 |
| | GRACE | .5003±.0009 | .5427±.0961 | .6004±.1210 | .5657±.1221 | .6023±.0931 | .5342±.0684 | .5064±.0145 | .5017±.0031 | .5074±.0190 |
| | **RGIB-REP** | .9404±.0067 | .9248±.0161 | .9218±.0051 | .9437±.0084 | .9299±.0070 | .9238±.0063 | .9473±.0071 | .9486±.0073 | .9397±.0045 |
| | **RGIB-SSL** | .9286±.0033 | .9241±.0055 | .9165±.0055 | .9276±.0023 | .9245±.0015 | .9257±.0037 | .9593±.0043 | .9586±.0027 | .9499 ±.0067 |

Table 33: Full results on Squirrel dataset with SAGE.

| layers | method | mixed noise 20% | 40% | 60% | input noise 20% | 40% | 60% | label noise 20% | 40% | 60% |
|---|---|---|---|---|---|---|---|---|---|---|
| | | SAGE | | | | | | | | |
| L=2 | Standard | .9680±.0015 | .9626±.0015 | .9570±.0016 | .9737±.0010 | .9736±.0010 | .9721±.0011 | .9692±.0015 | .9643±.0012 | .9606±.0017 |
| | DropEdge | .9657±.0011 | .9602±.0015 | .9535±.0020 | .9622±.0010 | .9625±.0009 | .9609±.0020 | .9593±.0009 | .9530±.0009 | .9593±.0007 |
| | NeuralSparse | .9572±.0008 | .9517±.0012 | .9562±.0019 | .9534±.0007 | .9534±.0010 | .9521±.0016 | .9589±.0012 | .9534±.0009 | .9591±.0010 |
| | PTDNet | .9578±.0009 | .9513±.0013 | .9559±.0021 | .9630±.0010 | .9504±.0007 | .9524±.0010 | .9596±.0014 | .9529±.0010 | .9596±.0019 |
| | Co-teaching | .9526±.0031 | .9560±.0071 | .9526±.0029 | .9403±.0050 | .9585±.0061 | .9585±.0101 | .9533±.0022 | .9533±.0306 | .9539±.0026 |
| | Peer loss | .9552±.0079 | .9507±.0050 | .9588±.0066 | .9512±.0052 | .9534±.0109 | .9599±.0013 | .9525±.0105 | .9555±.0178 | .9510±.0187 |
| | Jaccard | .9658±.0111 | .9619±.0068 | .9630±.0027 | .9600±.0189 | .9649±.0173 | .9617±.0171 | .9620±.0105 | .9608±.0068 | .9603±.0044 |
| | GIB | .9656±.0079 | .9660±.0077 | .9661±.0060 | .9632±.0260 | .9517±.0213 | .9565±.0010 | .9565±.0009 | .9593±.0002 | .9548±.0111 |
| | SupCon | .9628±.0015 | .9541±.0019 | .9409±.0040 | .9585±.0061 | .9585±.0101 | .9533±.0022 | .9533±.0306 | .9539±.0026 | .9400±.0021 |
| | GRACE | .8543±.0076 | .8572±.0094 | .8528±.0166 | .8563±.0068 | .8601±.0087 | .8528±.0075 | .8477±.0119 | .8459±.0105 | .8452±.0141 |
| | **RGIB-REP** | .9701±.0010 | .9647±.0014 | .9597±.0018 | .9757±.0009 | .9747±.0007 | .9737±.0011 | .9735±.0010 | .9707±.0007 | .9682±.0016 |
| | **RGIB-SSL** | .9536±.0016 | .9480±.0027 | .9448±.0030 | .9734±.0014 | .9739±.0014 | .9803±.0024 | .9765±.0012 | .9702±.0015 | .9694±.0005 |
| L=4 | Standard | .9584±.0107 | .9577±.0076 | .9541±.0021 | .9637±.0092 | .9630±.0079 | .9607±.0090 | .9663±.0020 | .9612±.0049 | .9612±.0020 |
| | DropEdge | .9518±.0030 | .9577±.0021 | .9515±.0033 | .9656±.0051 | .9612±.0093 | .9573±.0111 | .9683±.0020 | .9628±.0014 | .9608±.0012 |
| | NeuralSparse | .9604±.0083 | .9570±.0040 | .9513±.0040 | .9644±.0066 | .9599±.0084 | .9605±.0061 | .9652±.0035 | .9632±.0023 | .9605±.0021 |
| | PTDNet | .9624±.0050 | .9581±.0039 | .9536±.0033 | .9667±.0045 | .9668±.0057 | .9649±.0052 | .9675±.0043 | .9635±.0040 | .9607±.0026 |
| | Co-teaching | .9518±.0181 | .9468±.0093 | .9479±.0024 | .9525±.0086 | .9577±.0159 | .9528±.0137 | .9625±.0244 | .9610±.0220 | .9675±.0097 |
| | Peer loss | .9632±.0196 | .9662±.0085 | .9602±.0088 | .9674±.0145 | .9626±.0120 | .9643±.0145 | .9637±.0004 | .9617±.0170 | .9691±.0016 |
| | Jaccard | .9622±.0135 | .9606±.0157 | .9587±.0105 | .9647±.0084 | .9630±.0253 | .9640±.0278 | .9505±.0026 | .9508±.0063 | .9599±.0039 |
| | GIB | .9527±.0122 | .9538±.0275 | .9531±.0174 | .9603±.0081 | .9620±.0115 | .9580±.0330 | .9598±.0031 | .9523±.0064 | .9559±.0091 |
| | SupCon | .9478±.0042 | .9412±.0059 | .9342±.0039 | .9267±.0045 | .9368±.0028 | .9249±.0026 | .9375±.0034 | .9335±.0012 | .9317±.0021 |
| | GRACE | .8859±.0115 | .8751±.0085 | .8609±.0212 | .8824±.0113 | .8805±.0261 | .8633±.0178 | .8386±.0130 | .8466±.0103 | .8347±.0180 |
| | **RGIB-REP** | .9611±.0058 | .9578±.0035 | .9536±.0037 | .9682±.0049 | .9667±.0051 | .9584±.0074 | .9689±.0014 | .9664±.0027 | .9635±.0022 |
| | **RGIB-SSL** | .9498±.0032 | .9443±.0047 | .9422±.0044 | .9531±.0071 | .9599±.0023 | .9551±.0019 | .9623±.0021 | .9603±.0087 | .9601±.0091 |
| L=6 | Standard | .9555±.0065 | .9528±.0038 | .9461±.0054 | .9592±.0053 | .9600±.0036 | .9551±.0042 | .9574±.0192 | .9540±.0207 | .9583±.0023 |
| | DropEdge | .9544±.0069 | .9411±.0200 | .9463±.0041 | .9566±.0070 | .9473±.0172 | .9519±.0052 | .9464±.0023 | .9402±.0084 | .9396±.0025 |
| | NeuralSparse | .9563±.0060 | .9515±.0036 | .9457±.0054 | .9591±.0046 | .9460±.0052 | .9415±.0061 | .9371±.0025 | .9344±.0017 | .9319±.0033 |
| | PTDNet | .9565±.0060 | .9524±.0050 | .9463±.0056 | .9593±.0045 | .9584±.0044 | .9492±.0038 | .9384±.0023 | .9342±.0018 | .9236±.0018 |
| | Co-teaching | .9423±.0071 | .9396±.0126 | .9251±.0137 | .9382±.0063 | .9433±.0054 | .9388±.0088 | .9344±.0232 | .9496±.0204 | .9374±.0083 |
| | Peer loss | .9473±.0108 | .9371±.0040 | .9333±.0084 | .9339±.0053 | .9312±.0068 | .9331±.0066 | .9396±.0211 | .9375±.0195 | .9335±.0035 |
| | Jaccard | .9335±.0058 | .9305±.0057 | .9225±.0146 | .9309±.0059 | .9386±.0084 | .9433±.0147 | .9348±.0200 | .9210±.0285 | .9273±.0029 |
| | GIB | .9437±.0258 | .9323±.0155 | .9204±.0182 | .9492±.0239 | .9406±.0137 | .9322±.0335 | .9407±.0269 | .9364±.0274 | .9394±.0079 |
| | SupCon | .9392±.0049 | .9330±.0098 | .9221±.0138 | .9266±.0024 | .9313±.0102 | .9309±.0052 | .9264±.0021 | .9202±.0084 | .9196±.0021 |
| | GRACE | .8418±.0983 | .8470±.0823 | .8732±.0151 | .8747±.0151 | .8126±.1348 | .8716±.0148 | .8385±.0097 | .8370±.0295 | .8487±.0121 |
| | **RGIB-REP** | .9558±.0047 | .9474±.0066 | .9455±.0052 | .9583±.0044 | .9566±.0035 | .9447±.0030 | .9662±.0057 | .9646±.0021 | .9634±.0030 |
| | **RGIB-SSL** | .9437±.0026 | .9391±.0036 | .9366±.0042 | .9441±.0064 | .9549±.0012 | .9469±.0012 | .9678±.0031 | .9613±.0076 | .9601±.0041 |

