# OpenReview forum: "Towards Reliable Link Prediction with Robust Graph Information Bottleneck"
_ICLR.cc/2023/Conference — Submitted to ICLR 2023_

### Official Review · Reviewer_zAFv · 2022-10-23

**Confidence:** 3
**Correctness:** 3
**Technical Novelty And Significance:** 3
**Empirical Novelty And Significance:** 3
**Recommendation:** 6

**Clarity, Quality, Novelty And Reproducibility:**

This article is clearly structured and well written. The authors have a clear definition of the problem to be addressed and use sufficient theory to prove their views. Although information bottleneck theories are not a new topic, the authors use them skillfully. For reproducibility, the authors have not provided source code.


**Strength And Weaknesses:**

Pros: The paper is well written, and despite the considerable complexity of the method, its presentation is relatively easy to follow. The task of interest is well-defined, and has clearly been effectively solved on the datasets considered. The experiments provided in this paper are extensive and quite comprehensive. Specifically, the authors verified the effect and superiority of their proposed RGIB method in detail from different angles. The experimental results are also convincing.

Cons:
I am mainly concerned with the following issues.

(1). The baselines do not appear to be new enough. Almost all of the RGIB baselines in the experimental section are from 2020. Can the authors include baselines from 2021 and 2022?

(2). The work mainly focuses on robustness. Is there a metric to evaluate robustness of these methods?


**Summary Of The Paper:**

This work find that the inherent edge noise can perturb the graph topology and labels, which may reduce link prediction performance. Thus, the authors propose an information bottleneck guided method, namely RGIB. RGIB achieves robustness representation of graphs. The experiments show effectiveness of RGIB. This paper is very well written and the theoretical proof is very detailed.


**Summary Of The Review:**

I thought the article was well written. The experimental part is slightly lacking and I hope the authors can answer the questions being asked.

---

> ### Author Response · Authors · 2022-11-15
> **Response to Reviewer zAFv (Part 1/2)**
>
> We thank the reviewer for the insightful comments. Please kindly find the detailed responses to the comments below.
>
> - **Q1.** *Lack of up-to-date methods from 2021 and 2022 for comparison.*
>
> **Reply:** Thanks for this valuable comment. We have added two relevant baselines [1,2] published this year, as suggested by Reviewer sr13. In short, [1] is a variational information bottleneck-guided graph structure learning framework, aiming to deal with the structural noise on the input side. It is also based on GIB and deduces a variational approximation for irregular graph data to form a tractable objective function. Besides, [2] is another method for graph denoising from the perspective of information theory. It extends the principle of relevant information from a standard scalar random variable setting to structured data, and directly purifies the noisy graph structure in a learnable manner.
>
> Empirically, we justify that the proposed RGIB significantly outperforms these two baselines in dealing with the inherent noise. According to the results shown below, we have three following observations. (1) Generally, VIB [1] and PRI [2] are also effective in some noisy cases. (2) In contrast, it is clear that RGIB significantly outperforms these two baselines on 6 datasets, especially in dealing with the more challenging inherent noise. (3) As for the decoupled noise cases, these two methods work well in dealing with input noise, but the improvements in learning label noise are also marginal. Moreover, we will add the full evaluation results with more cases of these two methods added relevant discussions if accepted.
>
> | inherent noise ($\epsilon=40\\%$) | Cora      | Citeseer  | Pubmed    | Facebook  | Chameleon | Squirrel  |
> | ------------------------------------ | --------- | --------- | --------- | --------- | --------- | --------- |
> | standard training                    | .7419     | .7380     | .8748     | .9520     | .9496     | .9406     |
> | VIB [1]                              | .7810     | .8120     | .8825     | .9637     | .9561     | .9399     |
> | PRI [2]                              | .7330     | .7452     | .8801     | .9619     | .9499     | .9413     |
> | RGIB-SSL                             | .7966     | .7519     | .8834     | **.9770** | **.9621** | **.9455** |
> | RGIB-REP                             | **.8554** | **.8427** | **.8918** | .9711     | .9592     | .9426     |
>
> | input noise ($\epsilon=40\\%$) | Cora      | Citeseer  | Pubmed    | Facebook  | Chameleon | Squirrel  |
> | --------------------------------- | --------- | --------- | --------- | --------- | --------- | --------- |
> | standard training                 | .7856     | .7708     | .8759     | .9668     | .9433     | .9395     |
> | VIB [1]                           | **.8590** | .8399     | .8829     | .9710     | .9558     | .9402     |
> | PRI [2]                           | .7990     | .7852     | .8790     | .9698     | .9501     | .9419     |
> | RGIB-SSL                          | .8313     | .7996     | .8822     | **.9723** | **.9604** | **.9432** |
> | RGIB-REP                          | .8577     | **.8461** | **.8889** | .9707     | .9570     | .9429     |
>
> | label noise ($\epsilon=40\\%$) | Cora      | Citeseer  | Pubmed    | Facebook  | Chameleon | Squirrel  |
> | --------------------------------- | --------- | --------- | --------- | --------- | --------- | --------- |
> | standard training                 | .8054     | .7850     | .9039     | .9880     | .9580     | .9720     |
> | VIB [1]                           | .8296     | .8034     | .9042     | .9790     | .9629     | .9722     |
> | PRI [2]                           | .8139     | .7881     | .8917     | .9802     | .9593     | .9698     |
> | RGIB-SSL                          | .8318     | .7846     | .9343     | **.9883** | **.9797** | **.9733** |
> | RGIB-REP                          | **.9224** | **.9218** | **.9604** | .9881     | .9752     | .9729     |
>
> [1] Graph Structure Learning with Variational Information Bottleneck. AAAI 2022.
>
> [2] Principle of Relevant Information for Graph Sparsification. UAI 2022.

---

> > ### Author Response · Authors · 2022-11-15
> > **Response to Reviewer zAFv (Part 2/2)**
> >
> > - **Q2.** *For reproducibility, the authors have not provided source code.*
> >
> > **Reply:** Many thanks for your kind reminder. We have uploaded the source code to an anonymous repository and put its URL in the post of public response.
> >
> > - **Q3.** *Is there a metric to evaluate the robustness of these methods?*
> >
> > **Reply:** Thanks for this valuable comment. We think that it is an open problem that is also relevant to our work. Here, we provide a discussion of the robustness based on our understanding, which is in two folds.
> >
> > **(1)** Training-phase robustness. Briefly, the training data is "hard" due to it can be perturbed with input/label noise or with imbalance problems, e.g., long-tail sample distributions. By contrast, the testing data is comparably "easy" because it can be clean and well-balanced. Here, the robustness is usually evaluated by how the model trained on the "hard tasks" performs on "easy tasks", i.e., the performance on testing data. Considering the "hard tasks" can be with different difficulties, e.g., low or high noise ratios, the change in model performances is also a natural metric to quantify the robustness. A more robust model shall be with lower performance changes while maintaining good performance simultaneously.
> >
> > In addition to test accuracy, the loss visualization can be utilized to establish quantitative metrics to evaluate the intrinsic robustness of the model, e.g., the KL divergence between two predictions on an original input and a perturbed input [1]. It can provide more insights into the model compared with the aforementioned accuracy-based robustness estimation.
> >
> > **(2)** Testing-phase robustness. Here, the training data is "easy" and the testing data can be "hard", e.g., with perturbation generated by adversarial attacks, or encountering the out-of-distribution samples in testing time. The robustness is evaluated here for a trained model in the testing phase, where the property is that if a testing sample is “similar” to a training sample, i.e., the in-distribution samples, then the testing error is close to the training error. On the other hand, how this model performs on the out-of-distribution samples is usually hard to estimate, whether these samples are naturally collected or manually perturbated. Thus, measuring the performance of these "hard" testing samples, and be used to estimate the robustness of the model.
> > An early work [2] derives generalization bounds for learning algorithms based on their robustness. And recently, out-of-distribution generalization on graphs has made great progress and attracted more attention from the research community [3]. Besides, adversarial training has been demonstrated to improve model robustness against adversarial attacks as well as out-of-distribution generalization ability.
> >
> > [1] Interpreting and Evaluating Neural Network Robustness. IJCAI 2019.
> >
> > [2] Robustness and Generalization. Machine Learning 2012.
> >
> > [3] Out-Of-Distribution Generalization on Graphs: A Survey. Arxiv 2022.

---

> ### Author Response · Authors · 2022-11-17
> **Would you mind confirming if you have further questions? Thanks!**
>
> Dear Reviewer zAFv,
>
> We appreciate your comments and time! We have revised the paper following your suggestions. Would you mind checking our response and confirming if you have further questions?
>
> Best Regards,
>
> Authors

---

> ### Author Response · Authors · 2022-11-18
> **Discussion stage 1 is coming to the end. Would you mind confirming if you have further questions? Thanks!**
>
> Dear Reviewer zAFv,
>
> Thanks very much for your time and valuable comments. As discussion stage 1 is coming to the end, would you mind checking our response and confirming whether you have any further questions?
>
> Sincerely,
>
> Authors

---

> ### Author Response · Authors · 2022-11-27
> **We are anticipating your feedback!**
>
> Dear Reviewer zAFv,
>
> The conclusion of discussion period is closing, and we eagerly await your response. The authors greatly appreciate your time and effort in reviewing this paper and helping us improve it.
>
> We have provided detailed responses to every point of your concerns. Please help us to know whether they fully or partially address your concerns and if our explanations are in the right direction.
>
> Best Regards,
>
> Authors

---

> ### Author Response · Authors · 2022-11-29
> **We are anticipating your post-rebuttal feedback!**
>
> Dear Reviewer zAFv,
>
> We would like to sincerely thank you again for your time in reviewing our work!
>
> We understand you might be quite busy. However, as the discussion deadline is approaching, would you mind checking our response and confirming whether you have any further questions? Any further comments are discussions are welcomed!
>
> Best Regards,
>
> Authors

---

### Official Review · Reviewer_gCKh · 2022-10-23

**Confidence:** 3
**Correctness:** 3
**Technical Novelty And Significance:** 3
**Empirical Novelty And Significance:** Not applicable
**Recommendation:** 6

**Clarity, Quality, Novelty And Reproducibility:**

**For clarity:**

Most part of the paper is clear, but I'm not sure whether "coupled noise" is a good fit for the problem.

**For quality and originality:**

The studied problems and the proposed method are novel to my knowledge. Although RGIB does depend heavily on the previous GIB framework, the extension is non-trivial.

**For Reproducibility:**

The authors seem to promise the release of their codes.

**Strength And Weaknesses:**

This paper has several strong points that I appreciate very much.

1. The authors address a novel problem of label noise in the GIB framework.

2. Two adverse consequences of the existence of label noise in the original GIB framework, i.e., poorer alignment and worse uniformity, are clearly identified with good visualization.

3. The authors proposed a self-supervised learning-based extension to GIB, where the alignment and uniformity issues seem to be addressed with good theoretical support.

Meanwhile, I  also have some questions for the authors.

1. Since the target links are randomly selected from the existing links, isn't the structure loss and label noise the same noise rather than two different noises that are coupled together? Although I understand that the use of coupled noise makes it easier for the theoretical analysis for the discussion of GIB's issues and RGIB's advantages, I still think coupled noise may not be the best-fit description for the case.

2. In addition, there seem to be insufficient discussions regarding the limitation of the methods. I wonder whether the strength of the RGIB will hold if the "coupling" between label noise and structure noise is weak. Moreover, I'm also curious whether the RGIB framework can generalize to the same problem setting of the original RGIB for node classification, where there exists noise in the node labels?

**Summary Of The Paper:**

This paper proposes a robust graph information bottleneck (RGIB) for link prediction. This paper extends the previous graph information bottleneck (GIB) in a non-trivial manner by considering the fact that if the graph structure is assumed to be noisy, the labels in link prediction (i.e., hold-out links) are also noisy, and therefore the MI maximization term in the original GIB objective can still introducing noise in learned latent representations.

**Summary Of The Review:**

A good paper that theoretically extends the GIB framework to handle label noise problems in link predictions.

---

> ### Author Response · Authors · 2022-11-15
> **Response to Reviewer gCKh (Part 1/2)**
>
> We thank the reviewer for the valuable comments. Please kindly find the detailed responses below.
>
> - **Q1.** *Are structure noise and label noise the same noise rather than two different noises that are coupled together?*
>
> **Reply:** Thanks for this question. The structure noise and label noise does look quite similar and coupled. From the perspective of data processing, the collected noisy edges in the training set can be randomly split into the observed graph (i.e., the input of GNN) or the predictive graph (i.e., the query edges for the GNN to predict). Considering the noisy edges might come from similar sources, e.g., biases from human annotation, the corresponding noise patterns can also be similar between these two kinds of noise that are naturally coupled together to some extent.
>
> However, we would claim and justify that the two kinds of noise are fundamentally different.
>
> **(1)** Although both noises can bring severe degradation on empirical performances, they actually act in different places from the model perspective.  As the learnable weights are updated as $\bf{w} := {w} - \eta \nabla_{{w}} \mathcal{L}(\bf{H}, \tilde{Y})$, the label noise $\tilde{Y}$ acting on the backward propagation can directly influence the model. By contrast, the input noise indirectly acts on the learned weights as it appears in the front end of the forward inference, i.e., $\bf{H} = f_{w}(A, X)$.
>
> **(2)** Empirically, as results shown in Table 3 and Table 4, the standard GNN (without any defenses) performs quite differently under the same proportion of input noise and label noise. Such a phenomenon can inspire one to understand the intrinsic denoising mechanism or memorization effects of the GNN, and we would leave that as further work.
>
> **(3)** More importantly, from the perspective of defending, it could be easy and trivial to defend if these two kinds of noise are the same. However, none of the existing robust methods can effectively defend such an inherently coupled noise. As can be seen from Table 2, only marginal improvements are achieved when applying the existing robust methods to the coupled noise. While in Table 3 and Table 4, these robust methods work effectively in handling the decoupled noise, i.e., only the structure noise or label noise exists. The reason is that, the properties of coupled noise are much more complex than the single decoupled noise. Both sides of the information sources, i.e., $\tilde{A}$ and $\tilde{Y}$, should be considered noisy, based on which the defending mechanism could be devised.

---

> > ### Author Response · Authors · 2022-11-15
> > **Response to Reviewer gCKh (Part 2/2)**
> >
> > - **Q2.** *Whether the strength of the RGIB will hold if the "coupling" between label noise and structure noise is weak.*
> >
> > **Reply:** Thanks for this valuable comment. As introduced in the paper, coupled noise is common in real-world scenarios due to the random edge split manner. The noisy edges can be collected from similar sources and follow similar distributions. Thus, the patterns of structure noise and label noise can also be similar.
> >
> > For a further and deeper study of the "coupling" noise, we use the **edge homophily** metric to quantify the distribution of edges. Specifically, the homophily value $h_{ij}^{homo}$ of the edge $e_{ij}$ is computed as the cosine similarity of the node feature $x_i$ and $x_j$, i.e., $h_{ij}^{homo} = cos(x_i, x_j)$. As **Figure 8(c) shown in Appendix D.3**, the distributions of edge homophily are nearly the same for label noise and structure noise, where the envelope of the two distributions are almost overlapping. Here, we justify that the randomly split label noise and structure noise are indeed coupled together.
> >
> > Based on the measurement of edge homophily, we then decouple these two kinds of noise in the simulation. Here, we consider two cases that separate these two kinds of noise apart, i.e., (case 1) input noise with high homophily and label noise with low homophily shown in **Figure 9** and (case 2) input noise with low homophily and label noise with high homophily shown in **Figure 10**. Next, we perform standard training and two RGIB instantiations on these two decoupled cases and compare the case of coupled noise. The empirical results on two datasets are summarized as follows.
> >
> > | Cora dataset ($\epsilon=40\\%$) | Coupled | Decoupled case1 | Decoupled case2 |
> > | ------------------------------ | ------- | --------------- | --------------- |
> > | standard training              | .7419   | .7904           | .7284           |
> > | RGIB-SSL                       | **.8554**   | **.9032**           | **.8324**           |
> > | RGIB-REP                       | .7966   | .8323           | .8152           |
> >
> > | Citeseer dataset ($\epsilon=40\\%$) | Coupled | Decoupled case1 | Decoupled case2 |
> > | ---------------------------------- | ------- | --------------- | --------------- |
> > | standard training                  | .7380   | .8171           | .7870           |
> > | RGIB-SSL                           | **.8427**   | **.9226**           | **.8500**           |
> > | RGIB-REP                           | .7519   | .8222           | .7981           |
> >
> > According to the results shown above, we justify that the mutual information between structure noise and label noise is not necessarily high. In other words, these two kinds of noise can be decoupled and follow different distributions. More importantly, the RGIB also works effectively in different decoupled noise settings that significantly outperform the standard training by a large margin.
> >
> > - **Q3.** *Can the RGIB framework generalize to the node classification task under the same problem setting?*
> >
> > **Reply:** Thanks for your insightful comment. We conduct experiments with a 2-layer GCN on Cora and Citeseer datasets with random label noise on nodes, as you suggested. As the empirical results shown below, we justify that the RGIB framework can generalize to the node classification tasks with inherent label noise in nodes, where the two instantiations of RGIB also significantly outperform the standard training manner.
> >
> > | Cora dataset      | clean    | $\epsilon_y=20\\%$ | $\epsilon_y=40\\%$ | $\epsilon_y=60\\%$ |
> > | ----------------- | -------- | ----------------- | ----------------- | ----------------- |
> > | Standard training | .898     | .868              | .720              | .322              |
> > | RGIB-SSL          | **.900** | **.876**          | **.786**          | **.388**          |
> > | RGIB-REP          | .894     | .862              | .760              | .312              |
> >
> > | Citeseer dataset  | clean    | $\epsilon_y=20\\%$ | $\epsilon_y=40\\%$ | $\epsilon_y=60\\%$ |
> > | ----------------- | -------- | ----------------- | ----------------- | ----------------- |
> > | Standard training | .776     | .746              | .608              | .278              |
> > | RGIB-SSL          | **.784** | **.770**          | .646              | .324              |
> > | RGIB-REP          | .776     | .754              | **.654**          | **.364**          |

---

> > > ### Comment · Reviewer_gCKh · 2022-11-25
> > > **Thanks for the detailed response.**
> > >
> > > I appreciate the detailed responses from the authors. The responses have addressed some of my concerns, and I'm still leaning toward the acceptance side of this paper based on its merits.
> > >
> > > However, I am still confused about the coupling of structure noise and label noise in the link prediction setting. Because the paper mentions that the link prediction model is based on the graph auto-encoder model. This means that we split the edges in a graph into training, validation, and test set; we train a graph auto-encoder that takes as input and reconstructs ONLY the training edges, select the model according to the validation edges, and hope for the best that the model can generalize to the prediction testing edges.
> > >
> > > I'm wondering that does the author follow the original graph auto-encoder setting for link prediction or uses an encoder-decoder framework that doesn't only reconstruct input edges but also predicts new edges with labels that are not included in the input graph structure as well? Because if it is the first case, I still think the structure noise and label noise are the same noise, i.e., edge noise, that influences different phases of the model training/selection process.

---

> > > > ### Author Response · Authors · 2022-11-25
> > > > **Thanks for your valuable feedback.**
> > > >
> > > > We would like to clarify further the settings of link prediction in our work.
> > > >
> > > > **(1)** We indeed adopt the encoder-decoder framework to conduct the link prediction in our work, as in the second case you mentioned. The encoding process generates node representations as $H = f(A,X)$, where $A$ includes all the input edges. While in the decoding process, we only decode the logits of query edges but **not** the entire graph. Specifically, the logits of a query edge $e_{ij}$ is decoded as $\sigma(h_{i} \odot h_{j})$, where the node representations $h_i$ and $h_j$ come from the encoder.
> > > >
> > > > **(2)** The query edges, namely, the edges of labels $Y$, **do not include** the input edges $A$. As the model gets direct supervision by minimizing the classification loss, as $\min I(H;Y)$, the query edges directly influence the weights of GNN via backward propagation. On the other hand, the input edges only participate in the encoding process, and thus do **not** directly influence the weights of GNN.
> > > >
> > > > **(3)** Thus, although the input noise (in $A$) and label noise (in $Y$) are generated from similar distributions, they actually play different roles in the learning procedure. Briefly, the input noise reacts with the encoder in forward inference (as the observations), while the label noise reacts with the decoder in backward optimization (as the predicting targets).
> > > >
> > > > Thanks again for your valuable feedback. We will promote the clarity of the above-discussed content in the final version. Any further comments are discussions are welcomed!

---

> ### Author Response · Authors · 2022-11-17
> **Would you mind confirming if you have further questions? Thanks!**
>
> Dear Reviewer gCKh,
>
> We appreciate your comments and time! We have revised the paper following your suggestions. Would you mind checking our response and confirming if you have further questions?
>
> Best Regards,
>
> Authors

---

> ### Author Response · Authors · 2022-11-18
> **Discussion stage 1 is coming to the end. Would you mind confirming if you have further questions? Thanks!**
>
> Dear Reviewer gCKh,
>
> Thanks very much for your time and valuable comments. As discussion stage 1 is coming to the end, would you mind checking our response and confirming whether you have any further questions?
>
> Sincerely,
>
> Authors

---

### Official Review · Reviewer_sr13 · 2022-10-24

**Confidence:** 4
**Clarity, Quality, Novelty And Reproducibility:** Good originality. Writing and organiz…
**Correctness:** 3
**Technical Novelty And Significance:** 3
**Empirical Novelty And Significance:** 3
**Recommendation:** 5

**Strength And Weaknesses:**

Strength:
1. The coupled edge noise in the link prediction task is a good research problem and this paper proposed a reasonable solution.
2. The motivation is clear and the idea is novel.
3. The experimental results show that the proposed method is effective.

Weakness:
1. The inherent edge noise is defined as the additive edge noise, which seems to be not very unified since much noise for link prediction in the real world comes from missing relations.
2. The understanding part in section 3.2 is not very formal and rigorous. It’s unclear whether the quantitive and qualitative analysis by alignment and uniformity is specific to edge noise.
3. The design of the RGIB objective function is based on intuition, and its rigor needs to be further explained.
4. The relationship between the two instantiations, their respective advantages, limitations, and scope of application are not well explained.
5. Most of the baselines are published in 2020 and the latest one is published in 2021. Some SOTA methods should be included for comparison.
6. In the IB principle, the trade-off parameter is important, and needs more analysis.

Other questions and suggestions:
1. Some related works are missing. There is a recent work[1] that incorporates the IB to purify the graph structure, which should be included in the related work or included in the baselines. [2] is also a method for graph denoising from the perspective of information theory.
2. There are many graph adversarial attack methods that perturb structures. Since this paper focuses on the robustness of link prediction, how does the proposed RGIB perform under adversarial attacks?
3. As the scalability of graph algorithms has become increasingly important in recent years, I would like to see the performance of this method on larger datasets and its time consumption analysis.
4. I’m wondering how the distribution of H and the MI terms in the objective function change during the training process, which is helpful to understand the rationale of this method.
5. Are the alignment and uniformity specific measurements for edge noise in link prediction? Can other denoising methods also promote alignment and uniformity?

[1] Graph Structure Learning with Variational Information Bottleneck. AAAI 2022.
[2] Principle of Relevant Information for Graph Sparsification. UAI 2022.


**Summary Of The Paper:**

This paper focuses on the edge noisy scenario in the link prediction task and proposes a method named Robust Graph Information Bottleneck. The self-supervised learning technique and data reparametrization mechanism are utilized to instantiate RGIB.

**Summary Of The Review:**

The research problem and the idea are good. The analysis, experiments, and writing should be improved.

---

> ### Author Response · Authors · 2022-11-15
> **Response to Reviewer sr13 (Part 1/6)**
>
> We thank the reviewer for the valuable positive feedback. We addressed all the comments. Please kindly find the detailed responses as follows.
>
> - **Q1.** *Definition about the additive edge noise. The inherent edge noise is defined as the additive edge noise, which seems to be not very unified since much noise for link prediction in the real world comes from missing relations.*
>
> **Reply:** Thanks for your question. Basically, the edge noise can exist in the form of **false positive** edges or **false negative** noise. Specifically, the false positive edges are treated as existing edges with label 1, but in fact, such edges do not exist. On the other hand, the false negative edges are treated as non-existing edges with label 0 that the predictive probabilities of such edges will be minimized.
>
> Here, we would highlight that our work focuses on the **false positive** edges as it is more practical and common in real-world scenarios since the data annotating procedure can produce such a kind of noise [1]. Thus, if the inherent edge noise exists, it is more likely to be false positive samples. The false negative samples are often intractable to be collected and annotated in practice. This kind of noise cannot be added to the dataset to act as an explicit perturbation, and it is not the focus of our work.
>
> Actually, investigating the influences of the false negative samples is another line of research, such as [2,3], which is orthogonal and complementary to our work. Learning from a clean graph can also encounter the problem of false negative samples, which is usually due to the random sampling of negative nodes/edges. Note that tackling the false negative samples is a well-known problem in the area of link prediction while handling the false positive samples is also valuable but still remains underexplored.
>
> We have clarified the corresponding description in the latest draft.
>
> [1] A Survey of Trustworthy Graph Learning: Reliability, Explainability, and Privacy Protection. ArXiv 2022.
>
> [2] Understanding Negative Sampling in Graph Representation Learning. SIGKDD 2020.
>
> [3] Comprehensive Analysis of Negative Sampling in Knowledge Graph Representation Learning. ICML 2022.
>
> - **Q2.** *It’s unclear whether the quantitive and qualitative analysis by alignment and uniformity is specific to edge noise in Sec 3.2.*
>
> **Reply:** Thanks for your feedback. The intuition is that the alignment measures how the representations can be influenced when encountering topological perturbations, while uniformity describes how the representations distribute on the hyperplane. In other words, when taking the edge noise into consideration, the alignment actually measures the stability of GNN when the input graph is with extra edge noise in the testing phase, and the uniformity measures the denoising robustness of GNN when learning with inherent edge noise in the training phase.
>
> As introduced in section 3.2, desired representations should be invariant to the topological perturbations, i.e., with a higher alignment. On the other hand, they should also be uniformly distributed on the hyperplane to preserve as much information as possible [1]. Based on the concepts of alignment and uniformity, we then provide quantitive and qualitative analyses of the learned representations under edge noise in section 3.2. As shown, with a higher ratio of edge noise, the alignment and uniformity can be damaged to a greater extent. Thus, these two metrics can help to comprehend the learned representations by GNN under edge noise.
>
> Thus, both metrics are indeed closely connected with the edge noise here, and we reveal that a severer edge noise brings a poorer alignment and a worse uniformity.
>
> [1] Understanding Contrastive Representation Learning through Alignment and Uniformity on the Hypersphere. ICML 2020.

---

> > ### Author Response · Authors · 2022-11-15
> > **Response to Reviewer sr13 (Part 2/6)**
> >
> > - **Q3.** *The design of the RGIB objective function is based on intuition, and its rigor needs to be further explained.*
> >
> > **Reply:** Thanks for your valuable comment. Here, we would like to clarify the derival of RGIB (based on GIB) and the design of its two instantiations.
> >
> > **(1)** The GIB is intrinsically not robust to label noise since it entirely preserves the label supervision with maximizing $I(\bf H; \tilde{Y})$, as in Eqn. 1. As illustrated in Figure 1(b), the GIB decreases $I(\bf H;\tilde{A}|\tilde{Y})$ by directly constraining $I(\bf H;\tilde{A})$ to handle the input noise. Symmetrically, the label noise can be hidden in the area of $I(\bf H; \tilde{Y} | \tilde{A})$, but trivially constraining $I(\bf H ; \tilde{Y})$ to regularize $I(\bf H;\tilde{Y} | \tilde{A})$ is not ideal, since it will conflict with Eqn. 1. Besides, it cannot tackle the noise within $I(\tilde{A}; \tilde{Y})$, where the two kinds of noise can share similar patterns as the random split manner does not change their distributions in expectation (for this claim, please refer to our response for Q2 to Reviewer gCkh.).
> >
> > **(2)** Thus, it is crucial to further decouple the mutual dependence among $\tilde{A}$, $\tilde{Y}$, and $\bf H$. Regarding the representation $\bf H$, noise can exist in areas of $I(\bf H; \tilde{Y} | \tilde{A})$, $I(\bf H; \tilde{A} | \tilde{Y})$, and $I(\tilde{A} ; \tilde{Y} | \bf H)$, as shown in Figure 1(b). As $I(\tilde{A}; \tilde{Y} | \bf H) = I(\tilde{A} ; \tilde{Y}) + I(\bf H; \tilde{Y} | \tilde{A}) + I(\bf H; \tilde{A} | \tilde{Y}) - H(\bf H) + H(\bf H | \tilde{A}, \tilde{Y})$, where $\bf I(\tilde{A} ; \tilde{Y})$ is a constant and redundancy $H(\bf H | \tilde{A}, \tilde{Y})$ can be easily regularized. Therefore, the $I(\tilde{A}; \tilde{Y} | \bf H)$ containing the shared noise patterns of $\tilde{A}$ and $\tilde{Y}$ can be approximated by the other three terms, i.e., $H(\bf{H})$, $I(\bf H; \tilde{Y} | \tilde{A})$ and $I(\bf H; \tilde{A} | \tilde{Y})$. Since the two later terms are also with noise, a balance of these three informative terms can be a solution to the problem.
> >
> > **(3)** Based on the above analysis, we derive the RGIB principle that balances the three important information terms $H(\bf{H})$, $I(\bf H; \tilde{Y} | \tilde{A})$ and $I(\bf H; \tilde{A} | \tilde{Y})$. It works as an information bottleneck to filter out the noisy signals in both $\tilde{A}$ and $\tilde{Y}$, utilizing the supervision signals $I(\bf H; \tilde{Y})$ at the same time.
> >
> > **(4)** As for the detailed technical improvements of RGIB upon GIB, please refer to our response for Q7 to Reviewer mjIE. We have updated the above description in the latest draft and would like to know if you have further suggestions or comments.

---

> > > ### Author Response · Authors · 2022-11-15
> > > **Response to Reviewer sr13 (Part 3/6)**
> > >
> > > - **Q4.** *The relationship between the two instantiations, their respective advantages, limitations, and scope of application are not well explained.*
> > >
> > > **Reply:** Thanks for your feedback. We would like to clarify the comparison of the two instantiations, which is three folds.
> > >
> > > **(1)** Based on the empirical study of several datasets and GNNs, it is found that the two instantiations can be generalized to different scenarios. Generally, RGIB-SSL is more adaptive to sparser graphs, while RGIB-REP can be more suitable for denser graphs. More importantly, they can be complementary to each other with flexible options in practical applications, and we summarize such a point in Remark 5.1 of Section 5.1.
> > >
> > > **(2)** From the theoretical perspective, RGIB-SSL explicitly optimizes the representation $\bf H$ with self-supervised regularizations, i.e., alignment $I(\bf H_1; H_2)$ and uniformity $H(\bf H_1)$, $H(\bf H_2)$. By contrast, RGIB-REP implicitly optimizes $\bf H$ by purifying the noisy $\tilde{A}$ and $\tilde{Y}$ with the reparameterization mechanism to extract clean signals in the forms of latent variables $\bf Z_{Y}$ and $\bf Z_{A}$. The information constraints $I(\bf Z_{A}; \tilde{A})$, $I(\bf Z_{Y}; \tilde{Y})$ are directly acting on $\bf Z_{Y}$ and $\bf Z_{A}$ and indirectly regularizing the representation $\bf H$.
> > >
> > > **(3)** Besides, from the perspective of methodology, both instantiations are equipped with adaptive designs for obtaining an effective information bottleneck. We also conduct a further comparison and analysis of the two instantiations that are summarized as follows.
> > >
> > > | Instantiation | methodology              | advantages                                                   | disadvantages                                                |
> > > | ------------- | ------------------------ | ------------------------------------------------------------ | ------------------------------------------------------------ |
> > > | RGIB-SSL      | self-supervised learning | automated graph augmentation; good effectiveness; can be applied in entirely self-supervised settings without labels. | with expensive calculation for the contrastive objectives, especially the uniformity; requires extra graph augmentation operations. |
> > > | RGIB-REP      | data reparametrization   | no needs to do data augmentation; good efficiency; the input/output constraints do not require extra annotations for supervision and can be easily controlled. | sensitive to the hyper-parameters $\lambda$; less effective in extremely noisy cases; only applicable in fully supervised settings. |
> > >
> > > We will add the above comparison to our draft later with improved clarity and presentation.
> > >
> > > What's more, it is also possible that new instantiations based on other kinds of methodology are inspired by the robust GIB principle. We believe that such a bidirectional information bottleneck that strictly treats the information source on both input side and label side is helpful in practice, especially for extremely noisy scenarios.
> > >
> > > - **Q5.** *In the IB principle, the trade-off parameter is important, and needs more analysis.*
> > >
> > > **Reply:** Thanks for this insightful comment. We conduct an ablation study with the grid search of several hyper-parameters $\lambda$ in RGIB. For simplicity, we fix the weight of the supervision signal as one, i.e., $\lambda_s=1$. Then, the objective of RGIB can be formed as $L = L_{cls} + \lambda_1 R_{1} + \lambda_2 R_{2}$, where the information regularization terms $R_{1}/R_{2}$ are alignment and uniformity for RGIB-SSL (Eqn. 3), while topology constraint and label constraint for RGIB-REP (Eqn. 4), respectively. As the heatmaps illustrated in **Figure 12 and Figure 13 in Appendix F.2**, The $\lambda_1, \lambda_2$ are better in certain ranges. Neither too large nor too small value is not guaranteed to find a good solution.
> > >
> > > In addition to fixing the hyper-parameters $\lambda$ as constants, we also attempt dynamic tuning of these hyper-parameters within the training procedures, i.e., using the optimization schedulers. We attempt $5$ different schedulers in total, including constant, linear, sine, cosine, and exponential. As shown in Table 5, the selection of optimization schedulers greatly influences the final results, but there is no gold scheduler that consistently performs the best. Nonetheless, the constant and sine are generally better than others.

---

> > > > ### Author Response · Authors · 2022-11-15
> > > > **Response to Reviewer sr13 (Part 4/6)**
> > > >
> > > >
> > > >
> > > > - **Q6.** *Some recent methods [1,2] should be included for comparison.*
> > > >
> > > > **Reply:** Thanks for this valuable comment. We further make an extra comparison as suggested, and we have cited [1,2] in the revised version. According to the results shown below, we have three following observations. (1) Generally, VIB [1] and PRI [2] are also effective in some noisy cases. (2) In contrast, it is clear that RGIB significantly outperforms these two baselines on 6 datasets, especially in dealing with the more challenging inherent noise. (3) As for the decoupled noise cases, these two methods work well in dealing with input noise, but the improvements in learning label noise are also marginal. Moreover, we will add the full evaluation results with more cases of these two methods added relevant discussions if accepted.
> > > >
> > > > | inherent noise ($\epsilon=40\\%$) | Cora      | Citeseer  | Pubmed    | Facebook  | Chameleon | Squirrel  |
> > > > | ------------------------------------ | --------- | --------- | --------- | --------- | --------- | --------- |
> > > > | standard training                    | .7419     | .7380     | .8748     | .9520     | .9496     | .9406     |
> > > > | VIB [1]                              | .7810     | .8120     | .8825     | .9637     | .9561     | .9399     |
> > > > | PRI [2]                              | .7330     | .7452     | .8801     | .9619     | .9499     | .9413     |
> > > > | RGIB-SSL                             | .7966     | .7519     | .8834     | **.9770** | **.9621** | **.9455** |
> > > > | RGIB-REP                             | **.8554** | **.8427** | **.8918** | .9711     | .9592     | .9426     |
> > > >
> > > > | input noise ($\epsilon=40\\%$) | Cora      | Citeseer  | Pubmed    | Facebook  | Chameleon | Squirrel  |
> > > > | --------------------------------- | --------- | --------- | --------- | --------- | --------- | --------- |
> > > > | standard training                 | .7856     | .7708     | .8759     | .9668     | .9433     | .9395     |
> > > > | VIB [1]                           | **.8590** | .8399     | .8829     | .9710     | .9558     | .9402     |
> > > > | PRI [2]                           | .7990     | .7852     | .8790     | .9698     | .9501     | .9419     |
> > > > | RGIB-SSL                          | .8313     | .7996     | .8822     | **.9723** | **.9604** | **.9432** |
> > > > | RGIB-REP                          | .8577     | **.8461** | **.8889** | .9707     | .9570     | .9429     |
> > > >
> > > > | label noise ($\epsilon=40\\%$) | Cora      | Citeseer  | Pubmed    | Facebook  | Chameleon | Squirrel  |
> > > > | --------------------------------- | --------- | --------- | --------- | --------- | --------- | --------- |
> > > > | standard training                 | .8054     | .7850     | .9039     | .9880     | .9580     | .9720     |
> > > > | VIB [1]                           | .8296     | .8034     | .9042     | .9790     | .9629     | .9722     |
> > > > | PRI [2]                           | .8139     | .7881     | .8917     | .9802     | .9593     | .9698     |
> > > > | RGIB-SSL                          | .8318     | .7846     | .9343     | **.9883** | **.9797** | **.9733** |
> > > > | RGIB-REP                          | **.9224** | **.9218** | **.9604** | .9881     | .9752     | .9729     |
> > > >
> > > > [1] Graph Structure Learning with Variational Information Bottleneck. AAAI 2022.
> > > >
> > > > [2] Principle of Relevant Information for Graph Sparsification. UAI 2022.

---

> > > > > ### Author Response · Authors · 2022-11-15
> > > > > **Response to Reviewer sr13 (Part 5/6)**
> > > > >
> > > > > - **Q7.** *how does the proposed RGIB perform under adversarial attacks?*
> > > > >
> > > > > **Reply:** Thanks for this insightful comment. Adversarial attacks on graphs can be generally divided into poisoning attacks that perturb the graph in training time and evasion attacks that perturb the graph in testing time. Here, we conduct the poisoning attacks based on Nettack [1] that only that perturb graph structures, as you suggested. Notes that Nettack generates perturbations by modifying graph structure or node attributes such that perturbations maximally destroy downstream GNN’s predictions.
> > > > >
> > > > > Here, we apply Nettack on Cora and Citeseer datasets as representatives. As shown below, the adversarial attack that adds noisy edges to the input graph also significantly degenerates the GNN's performance. And comparably, the brought damage is more severe than randomly added edges. Crucially, we can observe that RGIB-SSL and RGIB-REP can also promote the robustness of GNN against adversarial attacks on graph structure.
> > > > >
> > > > > | Cora dataset      | clean     | $\epsilon_{adv}=20\\%$ | $\epsilon_{adv}=40\\%$ | $\epsilon_{adv}=60\\%$ |
> > > > > | ----------------- | --------- | --------------------- | --------------------- | --------------------- |
> > > > > | standard training | .8686     | .7971                 | .7671                 | .7014                 |
> > > > > | RGIB-SSL          | **.9260** | .8296                 | **.8095**             | **.8052**             |
> > > > > | RGIB-REP          | .8758     | **.8408**             | .7918                 | .7611                 |
> > > > >
> > > > > | Citeseer dataset  | clean     | $\epsilon_{adv}=20\\%$ | $\epsilon_{adv}=40\\%$ | $\epsilon_{adv}=60\\%$ |
> > > > > | ----------------- | --------- | --------------------- | --------------------- | --------------------- |
> > > > > | standard training | .8317     | .8139                 | .7736                 | .7481                 |
> > > > > | RGIB-SSL          | **.9148** | **.8656**             | **.8347**             | **.8022**             |
> > > > > | RGIB-REP          | .8415     | .8382                 | .8107                 | .7893                 |
> > > > >
> > > > > [1] Adversarial Attacks on Neural Networks for Graph Data. SIGKDD 2018.
> > > > >
> > > > >
> > > > > - **Q8.** *effectiveness and efficiency on large-scale datasets.*
> > > > >
> > > > > **Reply:** Thanks for this valuable comment. We conduct an empirical study on two large-scale datasets, i.e., PPI (56,944 nodes and 1,612,348 edges) and Yelp (716,847 nodes and 13,954,819 edges). **(1)** We first evaluate the performance without any defenses (i.e., standard training) with a 4-layer GCN that learns on different noisy cases. As shown in the table below, the added edge noise also degenerates the performance of GNN on these two large-scale datasets.
> > > > >
> > > > > | dataset / noise ratio    | clean | $\epsilon=20\\%$ | $\epsilon=40\\%$ | $\epsilon=60\\%$ |
> > > > > | ------------------------ | ----- | --------------- | --------------- | --------------- |
> > > > > | PPI (standard training)  | .9076 | .8505           | .8486           | .8482           |
> > > > > | Yelp (standard training) | .9143 | .8766           | .8708           | .8694           |
> > > > >
> > > > > **(2)** Next, we evaluate the effectiveness and efficiency of the proposed methods on these two datasets with $\epsilon=40\\%$ inherent noise. As shown below, RGIB can also bring promotions compared with standard training. Besides, RGIB indeed brings extra computing costs, and specifically, RGIB-SSL can be comparably more expensive than RGIB-REP. However, the extra computing costs are not so high and within an acceptable range.
> > > > >
> > > > > | model / dataset ($\epsilon=40\\%$) | PPI (test AUC) | PPI (training time) | Yelp (test AUC) | Yelp  (training time) |
> > > > > | --------------------------------- | -------------- | ------------------- | --------------- | --------------------- |
> > > > > | standard training                 | .8486          | 0.24h               | .8708           | 2.40h                 |
> > > > > | RGIB-SSL                          | .8613          | 0.25h               | .8875           | 2.71h                 |
> > > > > | RGIB-REP                          | .8565          | 0.24h               | .8909           | 2.52h                 |

---

> > > > > ### Author Response · Authors · 2022-11-15
> > > > > **Response to Reviewer sr13 (Part 6/6)**
> > > > >
> > > > >
> > > > > - **Q9.** *Further show-cases about the distribution of H and the MI terms. I’m wondering how the distribution of H and the MI terms in the objective function change during the training process, which is helpful to understand the rationale of this method.*
> > > > >
> > > > > **Reply:** Thanks for this feedback. We draw the learning curves of RGIB with constant schedulers in **Figure 16/17/18/19 of appendix F.3.** We normalize the values of each plotted line to (0,1) for better visualization.
> > > > >
> > > > > **(1)** For RGIB-SSL, the uniformity term, i.e, $H(\bf{H})$, converges quickly and remains low after 200 epochs. Similarly, the alignment term, i.e, $I(\bf{H}_1;\bf{H}_2)$ also converges in the early stages and keeps stable in the rest. At the same time, the supervised signal, i.e, $I(\bf{H};Y)$ gradually and steadily decreases as the training time moves forward. The learning processes are generally stable across different datasets.
> > > > >
> > > > > **(2)** As for RGIB-REP, we observe that the topology $I(\bf{Z}_A;\tilde{A})$ and label constraints $I(\bf{Z}_Y;\tilde{Y})$ can indeed adapt to noisy scenarios with different noise ratios. As can be seen, these two regularizations converge more significantly when learning on a more noisy case. That is, when noise ratio $\epsilon$ increases from 0 to $60\%$, these two regularizations react adaptively to the noisy data $\tilde{A}, \tilde{Y}$. Such a phenomenon shows that RGIB-REP with these two information constraints works as an effective information bottleneck to filter out the noisy signals.
> > > > >
> > > > > - **Q10.** *alignment and uniformity of other denoising methods. Are the alignment and uniformity specific measurements for edge noise in link prediction? Can other denoising methods also promote alignment and uniformity?*
> > > > >
> > > > > **Reply:** Thanks for this comment. The alignment of other methods is summarized in **Table 12**, while the uniformity is visualized in **Figure 14 and Figure 15 of appendix F.3**. We have the following three observations.
> > > > >
> > > > > **(1)** The sampling-based methods, e.g., DropEdge, PTDNet, and NeuralSparse, can also promote alignment and uniformity due to their sampling mechanisms to defend the structural perturbations.
> > > > >
> > > > > **(2)** The contrastive methods, e.g., SupCon and GRACE, are with much better alignment but much worse uniformity. The reason is that the learned representations are severely collapsed, which can be degenerated to single points seen from the uniformity plots but stay nearly unchanged when encountering structural perturbations.
> > > > >
> > > > > **(3)** The remaining methods are not observed with significant improvements in alignment or uniformity.
> > > > >
> > > > > When connecting the above observations with their empirical performances, we can draw a conclusion. That is, both alignment and uniformity are important to evaluate the robust methods from the perspective of representation learning. Besides, such a conclusion is in line with the previous study [1].
> > > > >
> > > > > [1] Understanding Contrastive Representation Learning through Alignment and Uniformity on the Hypersphere. ICML 2020.

---

> ### Author Response · Authors · 2022-11-17
> **Would you mind confirming if you have further questions? Thanks!**
>
> Dear Reviewer sr13,
>
> We would like to thank you again for your efforts and time in providing constructive feedback and comments. We have carefully considered your advice/questions and provided as much refinement and experiments to address your concerns regarding the submission.
>
> Since there are now only a few days remaining that we are allowed to further modify the submission. Would you mind checking the currently updated submission and confirming whether we have solved your concerns? If there are any further questions or suggestions, we very would like to conduct the experiments or refine the description to improve our submission.
>
> Sincerely,
>
> Authors

---

> ### Author Response · Authors · 2022-11-18
> **Discussion stage 1 is coming to the end. Would you mind confirming if you have further questions? Thanks!**
>
> Dear Reviewer sr13,
>
> Thanks very much for your time and valuable comments. As discussion stage 1 is coming to the end, would you mind checking our response and confirming whether you have any further questions?
>
> Sincerely,
>
> Authors

---

> ### Author Response · Authors · 2022-11-24
> **Would you mind confirming if you have further questions? Thanks!**
>
> Dear Reviewer sr13,
>
> We appreciate your comments and time! We have revised the paper following your suggestions. Would you mind checking our response and confirming if you have further questions?
>
> Best Regards,
>
> Authors

---

> ### Author Response · Authors · 2022-11-27
> **We are anticipating your feedback!**
>
> Dear Reviewer sr13,
>
> The conclusion of discussion period is closing, and we eagerly await your response. The authors greatly appreciate your time and effort in reviewing this paper and helping us improve it.
>
> We have provided detailed responses to every point of your concerns. Please help us to know whether they fully or partially address your concerns and if our explanations are in the right direction.
>
> Best Regards,
>
> Authors

---

> ### Author Response · Authors · 2022-11-29
> **We are anticipating your post-rebuttal feedback!**
>
> Dear Reviewer sr13,
>
> We would like to sincerely thank you again for your time in reviewing our work!
>
> We understand you might be quite busy. However, as the discussion deadline is approaching, would you mind checking our response and confirming whether you have any further questions? Any further comments are discussions are welcomed!
>
> Best Regards,
>
> Authors

---

> ### Author Response · Authors · 2022-11-30
> **We are anticipating your post-rebuttal feedback!**
>
> Dear Reviewer sr13,
>
> We sincerely thank you again for your time in reviewing our work!
>
> We have provided **six** detailed and point-to-point responses with **five** additional experiments with regard to the **ten** questions of your reviews.
>
> We understand you might be quite busy. However, as the discussion deadline is approaching, would you mind checking our response and confirming whether you have any further questions? Any further comments are discussions are welcomed!
>
> Best Regards,
>
> Authors

---

### Official Review · Reviewer_m5JE · 2022-10-24

**Confidence:** 4
**Correctness:** 2
**Technical Novelty And Significance:** 2
**Empirical Novelty And Significance:** 2
**Recommendation:** 5

**Clarity, Quality, Novelty And Reproducibility:**

CL.

Fig 1. a shows two illustrations of the same set of nodes; on the left the solid lines (clean edges) are connecting one set of the nodes, and on the right they are connecting another set; the same with the blue dashed line (noisy edges). The left graph has a title noisy input A_tilde and the right noisy labels Y_tilde. I do not understand what is the input and what are the labels here? An input graph is in my understanding a set of nodes and edges, this can also include labels (class labels) for each node but why the edges on the right are changed completely I do not follow.

V = {v i } but i is from 1 to ?

Tab. 1 has some interesting plots, which seem to indicate that the signal is lost as more randomly generated edges are added to the graph. However, it is unclear what the x and y axis are, it's unclear what dimension H_1 and H_2 are, it's unclear why both A_1 and A_2 are perturbed, I would imagine one would be clean and the other has the noise added? In the table with the values .616, .445 etc.. are representing what exactly, alignment maybe, but how is this quantified?

QU.

"Other methods (Rong et al., 2020; Zheng et al., 2020; Luo et al., 2021) achieve a similar goal by actively selecting the informative nodes or edges and pruning the task-irrelevant ones." I am familiar with Rong et al. and they do not actively select the informative edges, they do random sampling of the edges at each epoch so this sentence is inaccurate. Also this reference is incomplete and has a typo: "Y. Rong, W. Huang, T. Xu, and J. Huang. Dropedge: Towards volutional networks on node classification. In ICLR, 2020.".

"Since existing benchmarks are usually well-annotated and clean, there is a need to simulate the inherent edge noise properly to investigate the impact of noise." Can you substantiate this claim?

In Table 2 including the clean graph results will give a reference to how well the performance can be with no randomly added edges and can also show if the proposed method can improve on the results of 'clean' data, which is likely to have false positive edges even without artificially added ones.

NO.

It's not clear what additions are made to GIB to improve it's robustness.


RE.

It's not clear how the edges are added, I assume the additional synthetically added noisy edges simply sampled uniformly over all candidate edges. Also, the updates to the labels is not clear in this experiment?



**Strength And Weaknesses:**


Addressing noisy edges in a graph is a sensible and practical direction of research, as it is not often considered, is likely to be present in real data and will reduce model performance error. This paper demonstrates that errors increase for the link prediciotn task with synthetically added edges to public datasets.

However, it is many parts of this paper are unclear, how noise is added, what a graph illustration is trying to show, what the x and y values represetn, what values in a table represent, how well this approach works on the original data without synthetically added edges and more. This paper needs to explain more clearly many parts of the work.


**Summary Of The Paper:**

A method to handle graph-edge noise is introduced based on an information-bottleneck theory. It is shown that adding edge noise to real data will increase the link prediction error. The introduced method is shown to reduce this error.

**Summary Of The Review:**


Sadly this paper is too unclear to follow and understand. As well as addressing the above pointed out issues, I recommend focusing on explaining more clearly the edge data generation steps, the contribution added starting from the GIB work, and explaining why these additions improve on the existing work. I look forward to see the revised version as this is a very intersting research area.

[After Review Period]

The paper has a very nice contribution with respect to the motivation [identify and remove noise from graphs for link prediction] and the intuition behind the idea [extend GIB to separate the label and feature noise], and the paper is clearly structured. Many of the concerns have been addressed and modified in the paper and so I feel that the paper is closer to being ready for publication. It is still unclear how the claim that the public datasets either do not have noise, or that this noise is always FPs. Another reviewer commented that typically the noise in graphs are missing edges, which contradicts the authors comment. The experiments on the public dataset without added noise also show improvements with their method, implying that there must be some noise. This leaves a number of unanswered questions about the noise in the data. Maybe the authors can inspect samples of the edges being updated by RGIB in the clean datasets to see if qualitatively they can start to investigate these questions. Also, could the synthetically generate data to analyse the behaviour of their method? Due to this and the number of initially found issues I believe this is still below the acceptance level, although it has improved during the review period.

---

> ### Author Response · Authors · 2022-11-15
> **Response to Reviewer m5JE (Part 1/3)**
>
> We thank the reviewer for the valuable feedback, and we have addressed all the comments.
>
> Please kindly find the detailed point-to-point responses below.
>
> - **Q1.** *It's not clear how the edges are added.*
>
> **Reply:** Thanks for your feedback. We would like to clarify the simulation process of the inherent edge noise introduced in our work, which is in three folds.
>
> **(1)** We elaborate on the detailed noise simulation approach in Definition 3.1 and further introduce the details of how such noise simulation is compared with the common data split manner in the setup part of Section 5.
>
> **(2)** As in Definition 3.1, we simulate the inherent edge noise by adding random edges to the originally clean datasets. More importantly, the way that we add the edge noise is kept the **same** in all the experiments shown in our work.
>
> **(3)** In addition, we add a notation table in the Appendix. B to further clarify all the variables used in our work.
>
> - **Q2.** *How well this approach works on the original data without synthetically added edges?*
>
> **Reply:** Thanks for this valuable comment. We have conducted this supplement experiment that evaluates all the baseline methods on clean datasets. Due to the limiting space in the main content, we extend Table 2 with results on clean graphs to **Table 11 of Appendix F.1**. As shown in Table 11 of the latest draft, the proposed two instantiations of RGIB can also **boost** the predicting performance when learning on clean graphs, and significantly **outperform** other baselines in most cases.
>
> - **Q3.** *What are the inputs and labels in Fig 1 (a)?*
>
> **Reply:**  Thanks for your feedback. Our understanding of the input graph is consistent with yours, i.e., a graph denoted as $G=(A,X)$ consists of edges (i.e., non-zero elements in $A$) and nodes (with features $X$), as we have introduced in the notation part in Section 2. Here, we would like to explain Figure 1 further (a) as follows.
>
> **(1)** The inputs of a GNN here are the graph topology $A$ and node features $X$, and the outputs of the GNN are the logits of the query edges that are not observed in $A$.  As we focus on the link prediction task in this work, the node labels are not utilized for training the GNN here.
>
> **(2)** Note that predicting the node labels is the main focus of node-level graph learning tasks.  While in the link-level tasks, instead, one should pay more attention to the prediction of missing edges (links) in the graph, which is usually incomplete in its original topology.
>
> **(3)** Instead, the binary labels of these query edges are denoted as the labels $Y$, i.e., 0 for negative edges and 1 for positive edges. The noisy inputs $\tilde{A}$ and noisy labels $\tilde{Y}$ in Fig 1(a) are added with random edges with dashed blue lines. The way we simulate the inherent noise is to add random edges to the inputs and labels. Please refer to the elaborated simulation in Definition 3.1.
>
> Besides, we have clarified the corresponding description in the latest version.

---

> > ### Author Response · Authors · 2022-11-15
> > **Response to Reviewer m5JE (Part 2/3)**
> >
> >
> >
> > - **Q4.** *In Table1, what are the dimensions of ${H}_1$ and ${H}_2$, why both $A_1$ and $A_2$ are perturbed? How do the alignment values in Table1 are quantified? What are the x and y axes here?*
> >
> > **Reply:**  Thanks for your feedback. We would like to clarify the technical details here.
> >
> > **(1)** The edge representations $H_1$ and $H_2$ are of dimension $\mathbb{R}^{M \times D}$, where $M$ is the number of query edges of the test set (i.e., half positive edges and half negative edges). $D$ is the hidden dimension of edge representations, which is the same as the node representations since the edge representation $h_{ij}$ is obtained by the element-wise product of node representations, i.e., $h_{ij} = h_{i} \odot h_{j}$.
> >
> > **(2)** For quantifying the alignment measurement, we follow the previous work [1] where the alignment is computed as distanced between features of positive pairs with two random augmentations, i.e., both $A_1$ and $A_2$ are perturbed here. To obtain a more precise measurement, we repeat the augmentation $50$ times, and compute the mean values as the final results shown in Table 1 and Table 4. Specifically, the mean alignment value is exactly computed as $Alignment_{mean}=\frac{1}{50}\sum_{i=1}^{50}|| {H}_1^{i} - {H}_2^{i} ||_2$, where the representations ${H}_1^{i} = f({A}_1^{i}, X)$ and ${H}_2^{i} = f({A}_2^{i}, X)$, and the perturbed ${A}_1^{i}, {A}_2^{i}$ are generated by randomly adding/removing edges w.r.t. the $A$ seen in the training phase.
> >
> > **(3)** Besides, we also conduct a further experiment that uses an alternative way to quantify the alignment metric. That is, comparing the representations between one clean graph and one perturbed graph, i.e., $||{H} - {H}_2^{i} ||_2$, as you have pointed out. As shown in the below table, these two manners of computing the alignment values are consistent as the intra-rankings are the same.
> >
> > | dataset (noise ratio)  | mean of  $\|\|H_1^i - H_2^i \|\|_2$ | mean of $\|\|H - H_2^i\|\|_2$  |
> > | ---------------------- | --------------------------------------- | --------------------------------- |
> > | Cora (clean)           | .616 (rank=1)                           | .562 (rank=1)                     |
> > | Cora ($\epsilon=20\%$) | .687 (rank=2)                           | .600 (rank=2)                     |
> > | Cora ($\epsilon=40\%$) | .695 (rank=3)                           | .632 (rank=3)                     |
> > | Cora ($\epsilon=60\%$) | .732 (rank=4)                           | .650 (rank=4)                     |
> >
> > **(4)** In addition, Table 1 shows the mean alignments values on different datasets (i.e., the x axis) and various noise settings (i.e., the y axis) from clean to extremely noisy. Besides, the x and y axes in Figure 3 are the projection dimensions like the common TSNE plots. Here, we keep the same ranges of x (y) axis from $0$ to $128$ in different plots to provide a fair and consistent overview of representation distributions under various noise scenarios.
> >
> > We have updated the corresponding description in the latest draft.
> >
> > [1] Understanding Contrastive Representation Learning through Alignment and Uniformity on the Hypersphere. ICML 2020.
> >
> > - **Q5.** *An inaccurate claim of the Dropedge. Dropedge does not actively select the informative edges, they do random sampling of the edges. Also this reference is incomplete and has a typo.*
> >
> > **Reply:** Thanks for pointing out this problem. We have carefully revised the draft following your suggestions.

---

> > > ### Author Response · Authors · 2022-11-15
> > > **Response to Reviewer m5JE (Part 3/3)**
> > >
> > > - **Q6.** *"Since existing benchmarks are usually well-annotated and clean, there is a need to simulate the inherent edge noise properly to investigate the impact of noise." Can you substantiate this claim?*
> > >
> > > **Reply:** Thanks for your comment. We would like to claim that the inherent edge noise does exist in real-world graphs.
> > >
> > > **(1)** A similar problem has been noticed in an early work [1], where robots tend to build connections with normal users to spread misinformation on social networks, yielding the degeneration of GNNs for robot detection.
> > >
> > > **(2)** In addition, as pointed out by a recent survey [2], the inherent noise can be produced in the data generation process that requires manually annotating the data. As introduced in [2], in general, the inherent noise here refers to irreducible noises in graph structures, attributes, and labels. In short, the studied edge noise in our work is in line with the generic concept of graph inherent noise in the literature, which is recognized as a common problem in practice but also an **underexplored** challenge in both academic and industrial scenes.
> > >
> > > **(3)** However, several existing benchmarks, e.g., Cora and Citeseer, are generally clean and without annotated noisy edges. Inevitably, there is a gap when one would like to study the influence of inherent noise on these common benchmarks. Thus, to fill this gap, it is necessary to simulate the inherent edge noise properly to investigate the impact of noise. If the inherent edge noise exists, it should be false positive samples, while the false negative samples are often intractable to be collected. Thus, we focus on the investigation of the false positive edges as the inherent noise and design the simulation approach in Definition 3.1 as it is more close to the real-world scenarios.
> > >
> > > **(4)** Besides, to our best knowledge, we are the **first** to study the robustness problem of link prediction under the inherent edge noise. One of our major contributions to this research problem is that we reveal that the inherent noise can bring a severe representation collapse and performance degradation, and such negative impacts are general to common datasets and GNNs.
> > >
> > > [1] Using ghost edges for classification in sparsely labeled networks. SIGKDD 2008.
> > >
> > > [2] A Survey of Trustworthy Graph Learning: Reliability, Explainability, and Privacy Protection. ArXiv 2022.
> > >
> > > - **Q7.** *It's not clear what additions are made to GIB to improve it's robustness.*
> > >
> > > **Reply:** Thanks for your feedback. In short, RGIB generalizes the GIB with improvements in both theories and methodologies to learn a robust representation that is more resistant to the inherent edge noise, which is in four folds.
> > >
> > > **(1)** Conceptually, we would clarify that GIB is intrinsically susceptive to label noise since it entirely preserves the label supervision with maximizing . Thus, GIB **cannot** provide an ideal solution to the inherent edge noise investigated in this work. Different from the GIB, the RGIB takes both noisy inputs and noisy labels into the design of its information bottleneck. RGIB decouples and balances the mutual dependence among graph topology, target labels, and representation, building new learning objectives toward robust representation, as illustrated in Figure 1(b).
> > >
> > > **(2)** Analytically, GIB only indirectly regularizes the MI term $I(\bf H;\tilde{A}|\tilde{Y})$, as we introduce section 4.1, which can only solve partial noisy information. By contrast, RGIB takes all the related MI terms, i.e., $H(\bf{H})$, $I(\bf H; \tilde{Y} | \tilde{A})$ and $I(\bf H; \tilde{A} | \tilde{Y})$, based on which to provide a solution that balances these MI terms as a more strict information bottleneck.
> > >
> > > **(3)** From the view of methodology, we also provide two instantiations, RGIB-SSL and RGIB-REP, to approximate the aforementioned MI terms. These two instantiations benefit from different methodologies, i.e., self-supervised learning and data reparametrization, for implicit and explicit data denoising, respectively. Note that these two methodologies are not considered in the original GIB. However, we show that they can be utilized to implement the specific RGIB instantiation and build effective information bottlenecks for handling the inherent noise.
> > >
> > > **(4)** Empirically, we justify the effectiveness of the two instantiations of RGIB that outperform GIB in several noisy scenarios. Besides, the GIB is highly coupled with the GAT. By contrast, RGIB does not require any modifications to GNN architecture. It can be seamlessly integrated with various existing GNNs and promote their robustness against inherent noise.

---

> > > > ### Comment · Reviewer_m5JE · 2022-11-29
> > > > **Post Review Discussion**
> > > >
> > > > Q6 -> As an example I do not belive that the CORA dataset is hand labelled, but is the results of the construction of an academic portal using machine learning [1] and therefore could contain false positive or false negative edges; I believe the other datasets are also not hand labelled and therefore can have noisy edges. Furthermore, your results on the clean graphs show that the performance goes up significantly, indicating that these public datasets do have noisy edges. I have not seen a paper that claims the public datasets are free of noise, although I agree with your statement that real-world graph data could have more noise than these.
> > > >
> > > > 1. McCallum, A.; Nigam, K.; Rennie, J.; and Seymore, K. 2000. Automating
> > > > the construction of internet portals with machine learning. Information Retrieval
> > > > Journal.
> > > >
> > > > Q7 -> Thank you for the clear intuitive explanation of the differences of GIB and RGIB, the benefits sound clear.

---

> > > ### Comment · Reviewer_m5JE · 2022-11-29
> > > **Post Review Discussion**
> > >
> > > Thanks to the authors for the response.
> > >
> > > Q4 -> For clarity [Wang & Isola 2020] could be cited w.r.t. Table 1.
> > >
> > > Q5 -> I see the text is now corrected for DropEdge. It does raise some concern that the main point of a referenced paper was wrongly described here.

---

> > ### Comment · Reviewer_m5JE · 2022-11-29
> > **Post Review Discussion**
> >
> > Thanks to the authors for responding.
> >
> > Q1 -> In my comments I refer to how you choose a location for a generated noisy edge which is not described in Definition 3.1. Which two nodes (i and j) get the new edge? Is it uniform random between all pair-wise combinations of all nodes, excluding pair-wise combinations that already have an edge, or do you have a strategy to find pairs of nodes to connect based on their labels, their features, their degree to better guarantee an edge is a noisy one, not that you place an edge that should really be there if the label and features are the same for the nodes in a homogenuos graph for example?
> >
> > Q2 -> It's good to see that this approach can also improve the performance on the clean graph.
> >
> > Q3 -> This is clear now, thank you.

---

> ### Author Response · Authors · 2022-11-17
> **Would you mind confirming if you have further questions? Thanks!**
>
> Dear Reviewer m5JE,
>
> We would like to thank you again for your efforts and time in providing constructive feedback and comments. We have carefully considered your advice/questions and provided as much refinement and experiments to address your concerns regarding the submission.
>
> Since there are now only a few days remaining that we are allowed to further modify the submission. Would you mind checking the currently updated submission and confirming whether we have solved your concerns? If there are any further questions or suggestions, we very would like to conduct the experiments or refine the description to improve our submission.
>
> Sincerely,
>
> Authors

---

> ### Author Response · Authors · 2022-11-18
> **Discussion stage 1 is coming to the end. Would you mind confirming if you have further questions? Thanks!**
>
> Dear Reviewer m5JE,
>
> Thanks very much for your time and valuable comments. As discussion stage 1 is coming to the end, would you mind checking our response and confirming whether you have any further questions?
>
> Sincerely,
>
> Authors

---

> ### Author Response · Authors · 2022-11-24
> **Would you mind confirming if you have further questions? Thanks!**
>
> Dear Reviewer m5JE,
>
> We appreciate your comments and time! We have revised the paper following your suggestions. Would you mind checking our response and confirming if you have further questions?
>
> Best Regards,
>
> Authors

---

> ### Author Response · Authors · 2022-11-27
> **We are anticipating your feedback!**
>
> Dear Reviewer m5JE,
>
> The conclusion of discussion period is closing, and we eagerly await your response. The authors greatly appreciate your time and effort in reviewing this paper and helping us improve it.
>
> We have provided detailed responses to every point of your concerns. Please help us to know whether they fully or partially address your concerns and if our explanations are in the right direction.
>
> Best Regards,
>
> Authors

---

> ### Author Response · Authors · 2022-11-29
> **We are anticipating your post-rebuttal feedback!**
>
> Dear Reviewer m5JE,
>
> We would like to sincerely thank you again for your time in reviewing our work!
>
> We understand you might be quite busy. However, as the discussion deadline is approaching, would you mind checking our response and confirming whether you have any further questions? Any further comments are discussions are welcomed!
>
> Best Regards,
>
> Authors

---

> ### Author Response · Authors · 2022-11-30
> **Response to Reviewer m5JE for the Post Review Discussion (Part 1/2)**
>
> We appreciate the reviewer for the detailed feedback about our previous response and for increasing the score. About the remaining concerns about Q1, Q4, and Q6, we would kindly give more clarification as follows.
>
> - Q1. *How are the noisy edges added?*
>
> **Reply:** Regarding the way to add noisy edges, we would like to kindly point out that as explained in the 5th and 6th lines of Definition 3.1, the noisy edges are randomly generated with the zero elements in $A$ as candidates. This is truly the first case you have mentioned, i.e., uniform random between all pair-wise combinations of all nodes while excluding pair-wise combinations that already have an edge. The reason that we choose the random generation is to avoid manipulating biases in the simulation process. Regarding the corner case where we may place an edge that should really be there, we would like to clarify it requires pretty ideal conditions, e.g., two nodes have absolutely equal contents. Actually, it is not very common in Cora and Citeseer, etc., where most unobserved links should not exist indeed. On the other hand, while utilizing node features or labels under careful control might generate noisy edges that are more difficult to be learned from, we choose to utilize random generation for its **simplicity** and **generality**.
>
> Nevertheless, we also attempt an ablation study to generate noisy edges w.r.t. the node feature that is aligned with the second case you have mentioned. Specifically, we use the **edge homophily** metric to quantify the distribution of edges. The homophily value $h_{ij}^{homo}$ of the edge $e_{ij}$ is computed as the cosine similarity of the node feature $x_i$ and $x_j$, i.e., $h_{ij}^{homo} = cos(x_i, x_j)$. As **Figure 8(c)** shown in Appendix D.3, the distributions of edge homophily are nearly the same for label noise and structure noise, where the envelope of the two distributions are almost overlapping. Here, we show that the randomly split label noise and structure noise are indeed coupled together.
>
> Based on the measurement of edge homophily, we then decouple these two kinds of noise in the simulation. Here, we consider two cases that separate these two kinds of noise apart, i.e., (case 1) input noise with high homophily and label noise with low homophily shown in **Figure 9** and (case 2) input noise with low homophily and label noise with high homophily shown in **Figure 10**. Next, we perform standard training and two RGIB instantiations on these two decoupled cases and compare the case of coupled noise. As the empirical results are shown below, we justify that the mutual information between structure noise and label noise is not necessarily high. More importantly, the RGIB also works effectively in different decoupled noise settings that significantly outperform the standard training by a large margin.
>
>
>
> | Cora dataset ($\epsilon=40\\%$) | Coupled | Decoupled case1 | Decoupled case2 |
> | ------------------------------ | ------- | --------------- | --------------- |
> | standard training              | .7419   | .7904           | .7284           |
> | RGIB-SSL                       | .8554   | .9032           | .8324           |
> | RGIB-REP                       | .7966   | .8323           | .8152           |
>
> | Citeseer dataset ($\epsilon=40\\%$) | Coupled | Decoupled case1 | Decoupled case2 |
> | ---------------------------------- | ------- | --------------- | --------------- |
> | standard training                  | .7380   | .8171           | .7870           |
> | RGIB-SSL                           | .8427   | .9226           | .8500           |
> | RGIB-REP                           | .7519   | .8222           | .7981           |
>
> - Q4. *For clarity [Wang & Isola 2020] could be cited w.r.t. Table 1.*
>
> **Reply:** Thanks for your kind suggestion. We will revise the draft in the final version.

---

> > ### Author Response · Authors · 2022-11-30
> > **Response to Reviewer m5JE for the Post Review Discussion (Part 2/2)**
> >
> > - Q6. *The original datasets can also be intrinsic with edge noise.*
> >
> > **Reply:** Regarding the previous Q6, we thank the reviewer’s argument for understanding further your critical concern about the potentially intrinsic edge noise that may exist in the applied datasets. We do agree with the reviewer’s point and will revise the first sentence of Section 3.1 to avoid controversy.
> >
> > Here, we would kindly clarify that we assume the original datasets that are widely used in classical GNN study cannot be directly used to edge noise study. The reason is that we cannot either control different noise ratios or have an absolutely clean test set available. Note that, here, we mean the original test set is not clean anymore if we seriously consider the graph datasets are intrinsic with edge noise. Therefore, we follow the popular way in the machine learning community of label-noise learning [1,2,3,4], where they assume the original data are clean (e.g., CIFAR-10 and CIFAR-100) and add simulated noise for study. Actually, even CIFAR-10 and CIFAR-100 are not clean. They require expensive re-identification to form the ideal clean dataset and ideal noisy dataset, which we kindly refer to the recently proposed CIFAR-10N and CIFAR-100N [5]. A similar case is also found in ImageNet [6]. However, sticking to the datasets is beyond the purpose of this submission, and obtaining the **exact** answer to this question can require expensive manpower in checking the correctness of each annotated edge, which might contradict our target of automated denoising via the designed principle of robust graph information bottleneck.
> >
> > Therefore, in our work, we assume all the datasets used in our work are approximately clean, and we follow the common data split and evaluation manner with popular GNN models. Here, we make a further discussion on the intrinsic edge noise in two folds.
> >
> > **(1)** The intrinsic false negative edges. We would claim that the false negative samples are often intractable to be collected and annotated in practice. This kind of noise cannot be added to the dataset to act as an explicit perturbation, and it is not the focus of our work. Actually, investigating the influences of the false negative samples is another line of research, such as [7,8], which is **orthogonal** and **complementary** to our work. Learning from a clean graph can also encounter the problem of false negative samples, which is usually due to the random sampling of negative edges. Besides, we have cited these related works and made a further discussion in Appendix D.4 of our latest draft.
> >
> > **(2)** The intrinsic false positive edges. We focus on the false positive edges in our work as it is more practical and common in real-world scenarios since the data annotating procedure can produce such a kind of noise. Thus, if the inherent edge noise exists, it is more likely to be false positive samples. We do **agree** with your statement that the original dataset can also intrinsically with noisy edges that can be wrongly annotated or task-irrelevant. And empirically, the promotion brought by RGIB with the original dataset might shed some insights on this point. Besides, as the learning curves of RGIB shown in **Figure 16/17/18/19** of appendix F.3, two RGIB instantiations with information constraints can indeed adapt to noisy scenarios with different noise ratios, working as an effective information bottleneck to filter out the noisy signals. Thus, the RGIB can potentially act as a data inspector to estimate the intrinsic edge noise. It might be an exciting **extension** of the RGIB principle, and we will add a further discussion in the final version of our draft.
> >
> > We thank the reviewer’s argument again, and we will (1) revise the first sentence of Section 3.1 to avoid controversy; (2) add these discussions to the appendix with one section to explain our intuition in the final version. Any new comments and discussions are welcomed!
> >
> > [1] A Closer Look at Memorization in Deep Networks. ICML 2017.
> >
> > [2] MentorNet: Learning Data-Driven Curriculum for Very Deep Neural Networks on Corrupted Labels. ICML 2018.
> >
> > [3] Co-teaching: Robust Training of Deep Neural Networks with Extremely Noisy Labels. NIPS 2018.
> >
> > [4] DivideMix: Learning with Noisy Labels as Semi-Supervised Learning. ICLR 2020.
> >
> > [5] Learning with Noisy Labels Revisited: A Study Using Real-World Human Annotations. ICLR 2022.
> >
> > [6] Re-labeling ImageNet: from Single to Multi-Labels, from Global to Localized Labels. CVPR 2021.
> >
> > [7] Understanding Negative Sampling in Graph Representation Learning. SIGKDD 2020.
> >
> > [8] Comprehensive Analysis of Negative Sampling in Knowledge Graph Representation Learning. ICML 2022.

---

### Author Response · Authors · 2022-11-15
**General Response by Authors**

We would like to thank all the reviewers for their thoughtful suggestions on our paper. We are glad that the reviewers have some positive impressions of our work, including **(1)** an interesting well-defined research problem (m5JE, sr13, gCKh, zAFv); **(2)** a novel and non-trivial solution (sr13, gCKh); **(3)** convincing experiments (sr13, zAFv); **(4)** with theoretical supports (gCKh, zAFv); and **(5)** well written and easy to follow (gCKh, zAFv).

We have provided detailed responses to all the comments/questions point-by-point and also added new empirical evaluations. The summary of our updates is as follows:

- We further clarify the problem settings in Section 3.2, the method design in Section 4.1, and the corresponding technical details.
- We conduct more ablation studies to justify the effectiveness of our method, including (1) results on the original clean data in Appendix F.1, (2) trade-off of the hyper-parameter in Appendix F.2, (3) alignment and uniformity of other denoising methods in Appendix F.3, (4) visualizations of the distribution of information terms in Appendix F.3, and (5) decoupled noise distributions in Appendix D.3.
- We add two up-to-date baselines for comparisons and analyses, and justify the generalization ability of our method to adversarial scenarios, large-scale datasets, and other graph learning tasks. Please refer to our detailed responses to reviewers.

The above updates in the revised draft (the regular pages and the appendix of 26 pages) are highlighted in blue color.

We appreciate all reviewers’ time again. We are looking forward to your reply!

---

### Decision · Program_Chairs · 2023-01-20

**Decision:**

Reject

**Justification For Why Not Higher Score:**

The reviewers were not extremely enthusiastic about the submission. The paper needs probably a second pass and to be polished to be accepted. While the authors have significantly improved their submission over the last months, this probably needs a second round of reviews.

**Justification For Why Not Lower Score:**

N/A

**Metareview: Summary, Strengths And Weaknesses:**

__Summary.__ This paper focuses on the characterization of the effect of noisy edges on learned GNN node embeddings, and proposes a new technique --- RGIB --- to mitigate it. More specifically, the authors propose investigating the effect of edge noise (more specifically, edge additions) on two metrics ( as per Wang and Isola, 2020): uniformity and alignment. They show that these decrease in the presence of edge noise. Consequently, the authors propose a new learning loss that is supposed to mitigate the effect of edge deletions, which they call Robust Graph Information Bottleneck. Essentially, the objective amounts to maximizing the mutual information between H (the graph representation) and the labels, with constraints on the value of the entropy of H (it cannot be too big or too little), and upper bounds on the conditional information H, A|Y and H,Y|A. To compute these entropies and optimize their loss, the authors propose two variants: one based on self supervised learning (in which case the loss writes as a weighted sum of a loss on the imputed labels Y, a loss on uniformity and one on alignment); and one using a reparameterization trick.  The authors then evaluate their method on 6 different benchmarks.


__Feedback and Recommendation.__ The topic has been deemed interesting and of importance by all four reviewers, but the reviews are lukewarm at best. While interesting, the paper has also raised significant concerns during the rebuttal phase. At submission time, several reviewers deemed the paper unfinished --- lacking in clarity, missing figure labels and with factual mistakes in the description of other methods (e.g. DropEdge).While these have been corrected during the review process, the presentation of an unfinished paper impairs its evaluation.  Reviewers also urged the authors to perform more experiments, and in particular to benchmark their approach compared to newer SOTA methods. The discussion with the reviewers have consequently led the authors to provide more plots and insights into the trade-off in the information bottleneck principle, as well as to include newer benchmark methods.  The method does indeed seem like it has some amount of merit. However, the discussion of the robustness of the method to noise on the edges and labels could have benefited from a suit of synthetic experiments (e.g. using a contextual SBM) to make their point more salient.

Consequently, our recommendation is unfortunately a rejection of the paper. The main reason for the rejection is the "unfinished" quality of the paper at the initial stage. This might have put a dent in the ability of the reviewers to appropriately review the paper, and explain their lukewarm enthusiasm for the submission.

**Summary Of Ac-Reviewer Meeting:**

Two reviewers explained in particular their reservation on the paper: during the initial reviewing stage, the paper was highly unfinished. While the authors have considerably added to the paper, the evaluation of the paper was highly impaired. Both reviewers seem to find the method interesting, and potentially relevant.